# OVERSHOOT AND SHRINKAGE IN CLASSIFIER-FREE GUIDANCE: FROM THEORY TO PRACTICE

**Krunoslav Lehman Pavasovic**
FAIR at Meta, École Normale Supériure, Paris
{krunolp}@meta.com

**Jakob Verbeek**[*]
FAIR at Meta, Paris

**Giulio Biroli**[*]
École Normale Supériure, Paris

**Marc Mézard**[*]
Bocconi University, Milan

## ABSTRACT

Classifier-Free Guidance (CFG) is widely used in diffusion and flow-based generative models for high-quality conditional generation, yet its theoretical properties remain incompletely understood. By connecting CFG to the high-dimensional framework of diffusion regimes, we show that in sufficiently high dimensions it reproduces the correct target distribution—a "blessing-of-dimensionality" result. Leveraging this theoretical framework, we analyze how the well-known artifacts of mean overshoot and variance shrinkage emerge in lower dimensions, characterizing how they become more pronounced as dimensionality decreases. Building on these insights, we propose a simple nonlinear extension of CFG, proving that it mitigates both effects while preserving CFG's practical benefits. Finally, we validate our approach through numerical simulations on Gaussian mixtures and real-world experiments on diffusion and flow-matching state-of-the-art class-conditional and text-to-image models, demonstrating continuous improvements in sample quality, diversity, and consistency.

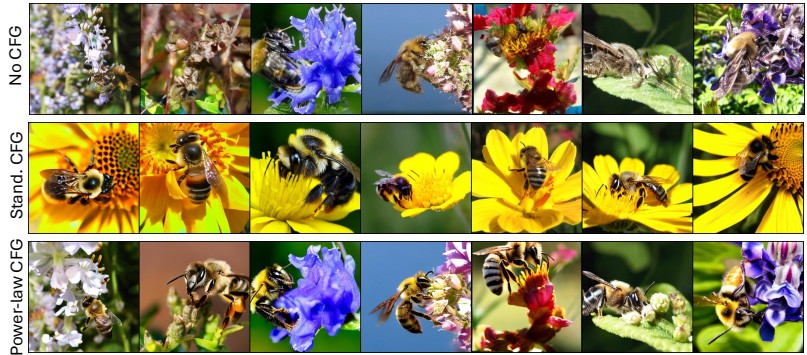

Figure 1: **Qualitative comparison of unguided sampling, standard Classifier-Free Guidance (CFG), and our proposed non-linear power-law CFG** (DiT/XL-2 on ImageNet-1K $256 \times 256$). Standard CFG increases fidelity at a substantial expense to diversity and semantic meaning compared to unguided CFG. Our power-law guidance improves fidelity at no cost to semantics or diversity. Each column sample starts from the same seed.

## 1 INTRODUCTION

Diffusion (Sohl-Dickstein et al., 2015; Song & Ermon, 2020; Ho et al., 2020) and flow-based methods (Lipman et al., 2022; Albergo et al., 2023; Liu et al., 2022) have emerged as the de facto state-of-the-art for generating high-dimensional signals. Diffusion relies on Orstein-Uhlenbeck Langevin dynamics, where noise is progressively added to the data until it becomes completely random. New samples are generated by reversing this process through a time-reversed Langevin

---

[*]Joint last author.

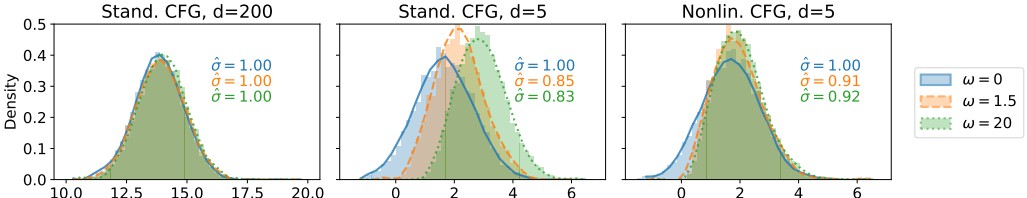

Figure 2: **CFG accurately generates the target distribution in high dimensions, causes mean overshoot and variance shrinkage in low dimensions, which are mitigated by nonlinear CFG.** We simulate the backward process using a two-Gaussian mixture with target mean $\vec{m} = (1, 1, \ldots)^d$ and variance $\sigma^2 = 1$, and project the generated samples onto the target mean, $q(t = 0) = \vec{x} \cdot \vec{m}/|\vec{m}|$. **Left:** in high dimensions ($d = 200$), CFG accurately recovers the target distribution. **Center:** in low dimensions ($d = 5$), CFG exhibits mean overshoot and variance shrinkage. **Right:** Our proposed nonlinear extension of CFG reduces these artifacts, partially restoring the target distribution.

equation. This backward evolution is steered by a force, the *score*, estimated from the data. In contrast, flow matching circumvents the diffusion construction by directly specifying the probability paths between noise and data. This is done by regressing onto a target vector field which in turn generates the desired probability paths. An important task for both paradigms is generating data conditioned on a class label or textual description of the image content. This can be achieved through conditioning mechanisms in the model architecture, as well as guidance techniques (Dhariwal & Nichol, 2021; Ho & Salimans, 2022) that steer the generation process towards samples aligned with user intentions or desired properties.

The notion of guidance was first introduced in classifier guidance (Song et al., 2020; Dhariwal & Nichol, 2021), where a pre-trained classifier is leveraged to induce class conditioning in the sampling of unconditional models. Relying on a pre-trained classifier is, however, computationally expensive and requires classifiers robust to noisy samples. Classifier-free guidance (CFG) (Ho & Salimans, 2022) was developed as an alternative, and was quickly adopted as a standard technique (Nichol et al., 2022; Betker et al., 2023; Saharia et al., 2022; Esser et al., 2024). CFG does not rely on an auxiliary classifier, instead, the model is trained to generate unconditional and conditional samples, and at inference extrapolates the denoising path towards the conditional one. Using CFG, however, the process *is no longer guaranteed* to sample the original conditional distribution.

CFG affects the generation process mainly in two ways; theoretically, this has been characterized in case of Gaussian mixtures in one and finite dimensions, where CFG causes ***mean overshoot***, **causing samples more shifted towards the boundary of the class**, and *variance shrinkage*, **resulting in a sharper distribution than the target one** (Chidambaram et al., 2024; Xia et al., 2024; Wu et al., 2024; Bradley & Nakkiran, 2024). These two effects are linked closely with the effects that the practicioners have observed, with CFG steering the samples towards the "mode" of high-quality and input-consistent samples corresponding directly to the well-observed increased **saturation/contrast**, while also **reducing sample diversity** in the process (Astolfi et al., 2024; Saharia et al., 2022).

Theoretically, it is unknown whether the unwanted effects can be dampened and whether in fact CFG can ever generate the correct distribution. In practice, as CFG has indeed shown beneficial regardless of these effects, it would be useful if similar guidances existed that reduce these effects, while keeping the practical benefits of increased quality of the generated samples. In this work, we provide theoretical analysis of the properties of the distribution generated by CFG, and how they compare to the target one. We examine how these effects arise, what influences them, and whether they can be dampened. Our results are backed by numerical simulations, and finally we test how do our findings extend to real world settings.

In summary, our contributions are the following:

**(1)** We provide blessing-of-dimensionality result showing that, in infinite and sufficiently high dimensions, CFG-guided trajectories generate the correct distribution, one that is generated by unguided conditional trajectories. This is established by relating CFG to the emergence of dynamical regimes (Biroli et al., 2024). In this setting, we show that CFG accelerates convergence of samples toward the target class.

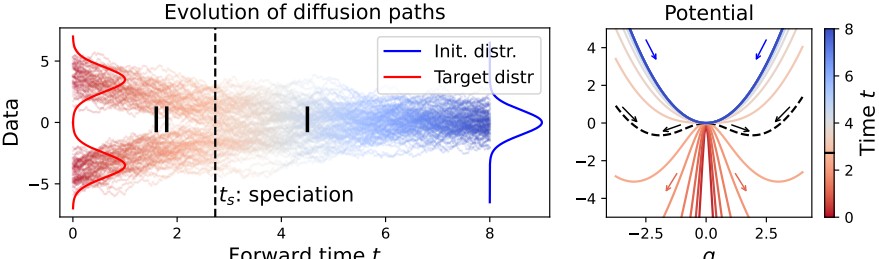

Figure 3: **Dynamical regimes in diffusion. Left:** Illustration of the speciation phenomenon using a one-dimensional Gaussian mixture. Starting from pure Gaussian noise at large time $t$, the backward diffusion begins in Regime I, where the class has not been decided yet. After speciation time $t_s$ (dashed line), the class membership is decided. **Right:** Evolution of the effective potential (conditional potential in Eq. (6)) over time for high-dimensional Gaussian mixture showcasing the symmetry breaking phenomenon.

**(2)** We demonstrate that, as dimensionality decreases, the commonly-observed artifacts of mean overshoot and variance shrinkage appear (illustrated on a toy Gaussian example in Fig. 2). We precisely characterize these effects and support our theoretical analysis with numerical simulations.

**(3)** Finally, we introduce a simple nonlinear generalization of CFG and prove theoretically that it mitigates both mean overshoot and variance shrinkage. We further show its superiority in numerical simulations (as in Fig. 2, right), and validate its practical benefits on state-of-the-art diffusion and flow-matching models, achieving improved sample quality, consistency, and diversity.

Our results can intuitively be described as follows: as the dimension grows, the "attraction force" for each class grows exponentially. One can visualize a well-separated class acting as a magnet that pulls diffusion trajectories towards itself (the attraction force depending on the mean and variance magnitude). During the backward process this force becomes very strong when the system starts to approach one class. Unless the magnitude of the CFG guidance scale also grows exponentially with the dimension (which is not the case in practice), these natural attraction forces completely dominate the CFG term. The backward dynamics and the attraction forces are portrayed in Figure 3 (left and right respectively).

## 2 RELATED WORK

Having introduced CFG, Ho & Salimans (2022) highlighted the trade-off between image quality, measured by Fréchet inception distance (FID, Heusel et al. (2017)), and diversity, measured by inception score (Salimans et al., 2016) when adjusting the guidance strength parameter $\omega$. Since then, a significant body of research has examined CFG from various perspectives.

**Theoretical works on CFG.** Several works employed Gaussian mixture models (GMMs) to analyze diffusion and guidance (Shah et al., 2023; Liang et al., 2024; Cui et al., 2023; Bai et al., 2024; Song et al., 2020). In contrast, Du et al. (2023) explored alternative conditioning, while Bradley & Nakkiran (2024) characterized CFG as a predictor-corrector (Song et al., 2020). Most relevant to this work, Chidambaram et al. (2024) demonstrated CFG's mean overshoot and variance shrinkage in one-dimensional settings, while Wu et al. (2024) extended the findings to multi-dimensional settings using GMMs. We expand on these by developing a high-dimensional statistical analysis and precisely characterizing how these effects diminish as dimensionality increases, ultimately demonstrating that the CFG-generated distribution in fact aligns with the target one for $d \to \infty$.

**CFG variants and experimental analyses.** Among experimental analyses of CFG, Karras et al. (2024a) propose guiding generation using a less-trained version of the model, Kynkäänniemi et al. (2024) apply CFG during a limited interval, and Wang et al. (2024) use weight schedulers for the classifier strength parameter. Several other CFG alternatives have been proposed, such as rectified guidance (Xia et al., 2024), projected score guidance (Kadkhodaie et al., 2024), characteristic guidance (Zheng & Lan, 2023), second-order CFG (Sun et al., 2023), CADS (Sadat et al., 2023), CFG++ (Chung et al., 2024), REG (Xia et al., 2024), APG (Sadat et al., 2024) and Feedback Guidance

(Koulischer et al., 2025). In later sections, we demonstrate our proposed nonlinear CFG generalizes to these methods, consistently enhancing their performance.

**Dynamical regimes, statistical physics and high-dimensional settings.** Statistical physics methods have shown particularly useful in analyzing high-dimensional generative models, *e.g.*, data from Curie-Weiss models (Biroli & Mézard, 2023), high-dimensional Gaussian mixtures (Biroli et al., 2024), and hierarchical models (Sclocchi et al., 2024). Furthermore, several recent works studied dynamical regimes diffusion models (Biroli & Mézard, 2023; Raya & Ambrogioni, 2024; Biroli et al., 2024; Sclocchi et al., 2024; Yu & Huang, 2024; Li & Chen, 2024; Aranguri et al., 2025), however none of them analyzed the effects brought by classifier-free guidance.

## 3 BACKGROUND AND HIGH-LEVEL DISCUSSION

We begin by providing an overview of the standard framework for generative diffusion, serving as the foundation for our analysis. Our exposition focuses on diffusion, though our findings directly extend to flow-matching with Gaussian paths (see, e.g., Lipman et al. (2024), Sec. 4.10.2).

### 3.1 GENERAL SETUP

Let $\{\vec{a}_i\}_{i=1}^n \in \mathbb{R}^d$ represent $n$ independent data points sampled from the *true* underlying data distribution $P_0(\vec{a})$. The forward diffusion process, starting from data points $\{\vec{a}_i\}_{i=1}^n$, is modeled by an Ornstein-Uhlenbeck process, described by the following stochastic differential equation (SDE):

$$d\vec{x}(t) = -\vec{x}(t)\,dt + \sqrt{2}\,d\vec{B}(t), \tag{1}$$

where $d\vec{B}(t)$ denotes the standard Brownian motion in $\mathbb{R}^d$. At any given time $t$, the state $\vec{x}(t)$ is distributed according to a Gaussian with mean $\vec{a}e^{-t}$ and variance $\Delta_t = 1 - e^{-2t}$. The forward process is terminated at time $t_f \gg 1$, when $\vec{x}(t_f)$ is effectively pure Gaussian noise, distributed as $\mathcal{N}(0, \mathcal{I}_d)$, with $\mathcal{I}_d$ being the identity matrix in $\mathbb{R}^d$.

The backward diffusion process operates in reverse time $\tau = t_f - t$, described with:

$$d\vec{x}(\tau) = \vec{x}(\tau)\,d\tau + 2\vec{S}(\vec{x}, \tau)\,d\tau + \sqrt{2}\,d\vec{B}(\tau), \tag{2}$$

where $\vec{S}(\vec{x}, t) = \vec{\nabla}\log P_t(\vec{x})$ denotes the score function. The backward diffusion process generates points $\vec{x}$ sampled from the distribution $P_t(\vec{x})$ for every time step $\tau$. At the end of the backward process, *i.e.*, when $\tau = t_f$, the process generates points drawn from the original distribution $P_0$.

In this work, we focus on generating data that can be categorized into distinct classes. We begin by assuming that the underlying data distribution is a $d$-dimensional probability distribution $P_0(\vec{x}, c)$, where $c$ represents a discrete class index and $\vec{x}$ a $d$-dimensional vector. The aim is to generate data conditioned on $c$, the class label. The procedure that is mathematically guaranteed to generate the exact conditional target distribution consists of using the true conditional score, $\vec{S}_t(\vec{x}, c) = \vec{\nabla}\log P_t(\vec{x}|c)$ in Eq. (2). CFG, however, does not do that; it instead further directs diffusion in a manner proportional to the difference between conditional and unconditional scores:

$$S_t^{\text{CFG}}(\vec{x}, c) = S_t(\vec{x}, c) + \Delta S_t^{\text{CFG}}, \qquad \Delta S_t^{\text{CFG}} := \omega\big(S_t(\vec{x}, c) - S_t(\vec{x})\big). \tag{3}$$

While CFG has shown practical benefits, such as improved fidelity and classification confidence (Wu et al., 2024), several key questions remain open: **(1)** Can one establish a theoretical framework proving that CFG can indeed generate the target distribution? **(2)** If so, can this framework also account for the well-documented artifacts of mean overshoot and variance shrinkage? **(3)** If these artifacts are indeed inherent to standard CFG, can one design alternative guidance schemes that provably mitigate them—and, crucially, do such schemes also deliver practical improvements beyond the theoretical setting? In this paper, we draw on the statistical-physics framework of Biroli & Mézard (2023); Biroli et al. (2024) to provide affirmative answers to all of the above.

### 3.2 CONNECTING DYNAMICAL REGIMES OF DIFFUSION TO CLASSIFIER-FREE GUIDANCE

To analyze CFG, we build on the statistical-physics framework of Biroli & Mézard (2023); Biroli et al. (2024), which identifies three dynamical regimes in diffusion models. Our exposition focuses

on the first two, since we show that CFG has the same effect in the second and third regime. These regimes are distinguished by the presence or absence of symmetry breaking, characterized via the leading eigenvalue of the data covariance matrix.

**Previous findings: dynamical regimes in diffusion.** Using this framework, Biroli et al. (2024) analyze the dynamical regimes of the backward process in Eq. (2) for two classes in the $d \to \infty$ limit. They identify the *speciation time* $t_\mathrm{s}$ as the transition between Regime I and Regime II. In Regime I, the backward trajectories have not yet committed to a particular data class, while in Regime II, they have committed and begin generating the class-specific features necessary to produce samples. The core structure of these regimes persists (in the infinite-dimensional limit) well beyond Gaussian mixtures, extending to models such as data lying on manifolds (Ventura et al., 2024; Bae et al., 2024; George et al., 2025).

**Our findings: connecting dynamical regimes to CFG.** A central insight of our work is that the dynamical-regime framework provides a principled perspective for understanding CFG, from which three main results emerge:

**Result I.** *In sufficiently large and infinite dimensions, CFG generates the correct target distribution.* Specifically: *(i)* Before speciation time $t_\mathrm{s}$, CFG accelerates convergence toward the target class. *(ii)* Just before $t_\mathrm{s}$, CFG-guided paths realign with the unguided trajectory that produces the correct distribution. *(iii)* After $t_\mathrm{s}$, CFG has no effect on the generation process.

**Result II.** *In finite dimensions, mean overshoot and variance shrinkage arise.* Using Gaussian mixtures, we characterize how the overshoot amplitude increases as dimensionality decreases and how CFG modifies the dynamics' potential, resulting in reduced variance of the generated distribution.

**Result III.** *There exist many simple guidances that effectively mitigate these effects.* Specifically, we introduce a simple nonlinear extension of CFG that reduces overshoot and variance shrinkage by multiplying the score difference $\Delta S_t^{\mathrm{CFG}}$ in (3) with $|\Delta S_t^{\mathrm{CFG}}|^\alpha$, $\alpha > 0$. This modification is theoretically justified and effective in simulations on Gaussian mixtures. As shown in Sec. 5, it is also beneficial in real-world applications.

While this framework provides valuable insights, it has a key limitation: it relies on access to the exact score. As a result, although it explains how to mitigate overshoot and variance shrinkage and supports nonlinear guidance schemes that improve practical performance, it does not clarify why standard CFG—despite these artifacts—often achieves strong empirical results. Addressing this question is beyond the scope of our current work, as it would require tools or analyses not available within our framework. Nonetheless, our theoretical analysis and nonlinear guidance show how to control these effects, providing meaningful theoretical insights and practical improvements.

## 4 MAIN RESULTS

The framework of Biroli & Mézard (2023); Biroli et al. (2024) characterizes dynamical regimes using a simple two-GMM. Its strength lies in broad applicability: such regimes appear across diverse generative models and data modalities (Ventura et al., 2024; George et al., 2025; Bae et al., 2024), and the framework has been validated on both real data and manifold-supported distributions (Biroli & Mézard, 2023). We provide its generalizations to non-centered/multi-component mixtures (Appx. C.1, C.2), heterogeneous variances, and high-dimensional manifold-structured data (Appx. C.3).

We focus here on the simplest case—two Gaussians of equal weight and isotropic variance—which already captures the essential phenomena and serves as the foundation for our theoretical framework.

### THEORETICAL FRAMEWORK

We examine the case where $P_0(\vec{a})$ is a superposition of two Gaussians with equal weight, means $\pm\vec{m}$ and isotropic variance $\sigma^2$. To ensure the two Gaussians are well separated, we take the large $d$ limit with fixed values of $|\vec{m}|^2/d$ and $\sigma^1$. As mentioned above, we assume exact scores are available.

In this setting, the speciation transition between Reg. I and II occurs on timescales $t_\mathrm{s} = \frac{1}{2}\log(d)$. Biroli et al. (2024) showed that $t_\mathrm{s}$ corresponds to the time at which diffusion paths commit to a

---

[1]The choice $|m| = \sqrt{d}$ is standard for modeling data with well-defined classes and has been adopted in several previous works (Li & Chen, 2024; Wu et al., 2024; Shah et al., 2023; Biroli et al., 2024).

specific class, as determined by changes in the potential of the backward Langevin equation (see Fig. 3, Appx. A). This speciation time plays a central role in our first contribution: before $t_s$, CFG guides trajectories toward the target, accelerating convergence, while just prior to $t_s$, guided and unguided paths become aligned. After $t_s$, trajectories naturally follow unguided paths, ensuring that CFG produces the correct target distribution in infinite and sufficiently high dimensions.

Building on this foundation, our analysis also reveals two additional phenomena: (i) how finite-dimensional corrections give rise to mean overshoot and variance shrinkage, which we characterize exactly, and (ii) how a nonlinear extension of CFG can mitigate these effects.

In practice, our framework generalizes effectively to text-to-image models where one can view T2I learning as learning a mixture of zero-variance Gaussians corresponding to unique prompt-image pairs. In the presence of noise or non-unique mappings, this representation evolves into a standard Mixture of Gaussians (MoG). Given that MoGs are universal approximators, this formulation offers a strong theoretical basis for not only class-conditional, but also text-to-image training, which we later show by evaluating on various state-of-the-art models.

In the following, we present the theoretical arguments underlying these three results in a concise, conceptual manner, with the full derivations and proofs deferred to Appx. B through Appx. E.

## 4.1 KEY FINDING I: CFG GENERATES THE CORRECT DISTRIBUTION IN INFINITE AND LARGE DIMENSIONS

To establish our first result, we analyze the distribution of $\vec{x}$ at time $t$, which can be written as

$$P_t(\vec{x}) \propto \exp\Big[-\frac{(\vec{x} - \vec{m}e^{-t})^2}{2\Gamma_t}\Big] + \exp\Big[-\frac{(\vec{x} + \vec{m}e^{-t})^2}{2\Gamma_t}\Big],$$

with $\Gamma_t = 1 + (\sigma^2 - 1)e^{-2t}$. In this setting, the CFG score (Eq. (3)) simplifies to

$$S_t^{\text{CFG}}(\vec{x}, c) = -\frac{\vec{x}}{\Gamma_t} + \frac{c\vec{m}e^t}{\Gamma_t} + \omega\frac{\vec{m}e^{-t}}{\Gamma_t}\left\{c - \tanh\Big(\frac{\vec{x} \cdot \vec{m}e^{-t}}{\Gamma_t}\Big)\right\}, \tag{4}$$

with $c = \pm 1$ and $\omega > 0$. Our analysis develops the first result through three successive steps, all linked to the speciation time $t_s = \frac{1}{2}\log d$, the point at which trajectories commit to a specific class.

**Step I: CFG guides trajectories before speciation.** As we show in Appx. B.1, before $t_s$, CFG provides an extra push toward the target class, accelerating convergence. Importantly, as seen in the CFG score formula (4), CFG only affects the $\vec{m}$ directions; all directions orthogonal to $\vec{m}$ remain unaffected. Formally, projecting the backward dynamics onto a vector $\vec{v} \perp \vec{m}$ shows that the resulting dynamics are independent of $\omega$, confirming only the $\vec{m}$ direction is influenced.

Therefore, we project onto $\vec{m}$ and define $q(t) = \vec{x} \cdot \vec{m}/|\vec{m}|$ with $|\vec{m}| = \sqrt{d}$. Then the backward evolution guided toward class $c = 1$ satisfies

$$dq = \Big(q + 2\big[-q + e^{-(t_f - t_s - \tau)}((1+\omega) - \omega\tanh\big(qe^{-(t_f - t_s - \tau)}\big))\big]\Big)d\tau + d\eta(\tau), \tag{5}$$

where $\tau = t_f - t$ and $d\eta(\tau)$ denotes $\sqrt{2}$ times Brownian motion (specific theoretical arguments provided in Appx. B). This can be rewritten in terms of an effective potential $V^{\text{CFG}}(q, \tau)$:

$$V^{\text{CFG}} = \underbrace{\frac{1}{2}q^2 - 2e^{-(t-t_s)}q}_{\text{Conditional potential}} + \omega\underbrace{\left[-qe^{-(t-t_s)} + \ln\cosh\big(qe^{-(t-t_s)}\big)\right]}_{\text{CFG-induced potential}}. \tag{6}$$

Examining the effective potential in (6), we observe that the CFG-induced term provides an additional push along the $\vec{m}$ direction toward the target class $c = 1$. This effect is especially significant for trajectories that deviate from typical paths, correcting those that would otherwise move toward the wrong class (see Fig. 7 in Section B). These observations allow us to conclude that, prior to the speciation $t_s$, CFG amplifies the push toward the desired class, establishing Step I.

**Step II: CFG paths align before exiting Regime I.** At late times in Regime I, $q$ becomes of order $\sqrt{d}$ (Biroli & Mézard, 2023), and the CFG-added term in Eq. (4) produces only exponentially small

corrections to the dynamics. Although different values of $\omega$ may have led to different positions for $q$ earlier in the backward process, these differences are quickly forgotten. The full theoretical argument is spelled out in Sec. B.2, i.e., we provide the proof of the dynamics naturally readjusting, and the trajectories converging toward the same distribution they would follow without CFG.

Step II is therefore obtained by analyzing the corresponding SDE starting from this late stage of Regime I: we show its solution no longer depends on the earlier CFG-induced deviations and coincides statistically with the backward evolution of a single Gaussian corresponding to the target class $c = +1$, implying the path alignment (see Fig. 8 in Section B for numerical simulations).

**Step III: After speciation time $t_s$, CFG paths follow the unguided path.** At the end of Regime I, as we show in Sec. B.3, the projection $q$ diverges, and one has to analyze the rescaled variable $\vec{x} \cdot \vec{m}/d$. By this stage, the CFG-guided paths have realigned with the unguided conditional paths.

Specifically, we use results from Biroli et al. (2024) to show that in Regime II the extra CFG term in Eq. (4) effectively vanishes, since $1 - \tanh{(\vec{x} \cdot \vec{m} e^{-t}/\Gamma_t)} \to 0$ as $d \to \infty$. As a result, the dynamics as well as the trajectories follow the unguided backward evolution, yielding Step III.

**Consistent conclusions in large, finite $d$.** Within Regime I, for large, yet finite dimension, the CFG-added-term in the score in Eq. (4) remains of the same order as the conditional score of the unguided path so CFG has the same effect as in the infinite limit. The full proof is provided in Appx. D; when exiting Regime I and during Regime II, the extra CFG term is exponentially small in $d$, so the results from all three steps carry over for large but finite $d$ (we discuss below the strength of the corrections for finite $d$).

In summary, during Regime I, CFG accelerates convergence toward the target class. Just before speciation, the paths realign with the unguided trajectory, subsequently following them throughout Regime II. This behavior is illustrated in Section B, Fig. 8. This result shows that, contrary to previous beliefs (Chidambaram et al., 2024; Wu et al., 2024), CFG can indeed generate the correct distribution, and serves as key guiding principle for the remainder of our work.

## 4.2 KEY FINDING II: CFG EFFECTS IN FINITE DIMENSIONAL SETTINGS

So far, we have shown that for any value of $\omega$ the target distribution is correctly reproduced in the infinite dimensional limit. We now show that the mean overshoot and variance shrinkage arise as the dimensionality of the system decreases. Full arguments can be found in Appx. D.

**Mean overshoot and variance shrinkage in finite $d$.** In finite dimensions, the paths do not realign when exiting Regime I. The additional push introduced by CFG within Regime I has an effect on Regime II, resulting in an overshoot of the target distribution of relative amplitude of order $1/\sqrt{d}$. The CFG-added-term also results in a larger second derivative of the potential $V^{\text{CFG}}(q, t)$. Thus, the CFG Langevin equation is associated to a more confining potential, ultimately shrinking the variance of the CFG-generated distribution. These are in line with previous empirical (Ho & Salimans, 2022) and theoretical findings (Chidambaram et al., 2024; Wu et al., 2024). In Appendix Figs. 8 and 12, we further analyze how CFG behavior changes with increasing dimensionality and number of classes both for numerical simulations and real-world scenarios. This concludes our second finding.

## 4.3 KEY FINDING III: NONLINEAR POWER-LAW CFG MITIGATES FINITE-DIM. EFFECTS

Building on the finite-dim. deviations identified in Key Finding II, we propose Power-Law CFG: a simple nonlin. extension of standard CFG. Our scheme raises the cond. score difference (3) along $\vec{m}$ to a power $\alpha > 0$, allowing guidance to scale with the local strength of the conditional signal:

$$\vec{S}_t^{\text{PL}}(\vec{x}, c) = \vec{S}_t(\vec{x}, c) + \omega \left[ \vec{S}_t(\vec{x}, c) - \vec{S}_t(\vec{x}) \right] \left| \vec{S}_t(\vec{x}, c) - \vec{S}_t(\vec{x}) \right|^\alpha. \tag{7}$$

Intuitively, this modification has two complementary effects: (i) When the score difference $\delta S_t = |\vec{S}_t(\vec{x}, c) - \vec{S}_t(\vec{x})|$ (vector norm) is small and potentially unreliable, guidance is naturally dampened. (ii) When the signal is strong, guidance is amplified, enhancing the push toward the correct class.

Importantly, as a direct consequence of our large-$d$ theoretical analysis, nonlinear power-law CFG retains the exact high-dim. guarantees of standard CFG: in the limit of large dimension, it recovers

the correct conditional distribution without distortion, ensuring that the benefits of nonlinear scaling are confined to the finite-dimensional regime. This is discussed in Appx. H.

**Reduced overshoot and shrinkage.** Our analysis in App. E identifiies the key term $B(q) := de^{-(t_f-t)}/\Gamma u(q)$ as the driver for mean overshoot and variance shrinkage. This term is intrinsically linked to the Hessian (curvature) of the potential, and the effect of nonlinear guidance is shown to directly modify $B(q)$, with $0 < B(q) < 1$ almost everywhere. **Mean Overshoot:** Overshoot is proportional to $B(q)$ and the parameter $\alpha$ controls this through $B(q)^{\alpha}$: $\alpha > 0$ suppresses the overshoot, while $\alpha < 0$ increases it. **Variance Shrinkage** is positively correlated with $B(q)$; $B(q)$ determines the ratio of curvature increments between nonlinear and linear CFG through $B(q)^{\alpha}$. Specifically, $\alpha > 0$ reduces the curvature increase (decreasing shrinkage), while $\alpha < 0$ amplifies it (increasing shrinkage). Simulations (see, e.g., Fig. 2 (right) or Fig. 35) confirm this.

**Application to flow matching.** The power-law formula can be directly applied to flow-matching as well; in order be fully consistent with our theoretical proposition, we expressed the Power-Law formula as $\phi_t(s) = (\frac{1-t}{t})^{\alpha} s^{\alpha}$, rather than $\phi_t(s) = s^{\alpha}$. However, even using $\phi_t(s) = s^{\alpha}$ (which is a valid guidance within our framework) yields very similar results and behavior - this is described in detail in appendix G.2.1.

**Magnitude vs. Directional Scaling.** Our Power-Law formulation uses the Euclidean norm $\|S_t(\vec{x}, c) - S_t(\vec{x})\|$ which captures both score *direction* and *magnitude*, unlike, e.g., cosine metrics that would isolate alignment. While the Euclidean norm is resolution-sensitive, luckily, the guidance parameter $\omega$ acts as a renormalization factor, which ultimately removes the need for an additional explicit resolution-based adjustments. As detailed in App. G.8, our analyses showed that this combined signal yields superior performance compared to purely directional metrics, though alternative cosine-based methods remain a promising avenue for future work.

## 4.4 DISCUSSION: OTHER NONLINEAR EXTENSIONS

Beyond the power-law modification, our analysis indicates a broad class of theoretically valid, potentially beneficial nonlinear guidances. Our framework and the "blessing of dimensionality", ensuring CFG recovers the correct target distribution in high dimensions, are not limited to the specific construction above: many nonlinear extensions share the same guarantees.

Formally, our results can be extended to nonlinear guidances of the form

$$S_t^{\text{CFG-NL}}(\vec{x}, c) = S_t(\vec{x}, c) + \left[ S_t(\vec{x}, c) - S_t(\vec{x}) \right] \phi_t \left( \left| \vec{S}_t(\vec{x}, c) - \vec{S}t(\vec{x}) \right| \right), \tag{8}$$

under the condition $\lim_{s\to 0} s\phi_t(s) = 0$, ensuring guidance vanishes smoothly when cond. and uncond. scores coincide. This condition essentially relies on the part of our proof showing that scores equalize at low noise levels (as often observed in practice), preventing pathological guidances that break this property). This framework recovers a number of existing methods as special cases: constant $\phi_t(s) = \omega$ yields standard CFG; $\phi_t(s) = \omega \cdot \mathbb{I}_{[t_1,t_2)}(t)$ gives limited-interval CFG (Kynkäänniemi et al., 2024); and time-varying $\phi_t(s) = \omega_t$ recovers weight schedulers (Wang et al., 2024; Gao et al., 2023). Other recent proposals (Chung et al., 2024; Xia et al., 2024; Ventura et al., 2024) can also be expressed as specific, simple choices of $\phi_t$. Importantly, all of these remain linear in the score difference $\delta S_t$, whereas our nonlinear power-law scheme $\vec{S}_t^{\text{PL}}$ demonstrates that altering the score difference in a nonlinear manner is both theoretically natural and practically beneficial.

This perspective opens a larger design space of alternative nonlinear functions $\phi_t$, which could as well be directly optimized, yielding theoretically sound guidance mechanisms, potentially outperforming existing approaches. Building on this, we next evaluate the nonlinear power-law guidance in both controlled GMM settings and large-scale generative models, validating theoretical predictions, assessing robustness, and comparing directly to linear CFG and alternative CFG schemes.

## 5 NUMERICAL SIMULATIONS AND REAL-WORLD EXPERIMENTS

We now turn to experiments, testing nonlinear CFG derived from our large-$d$ theoretical guidelines. Through num. simulations and real-world experiments, we evaluate if the theoretically motivated power-law CFG delivers tangible benefits, e.g., in mitigating overshoot and preserving variance.

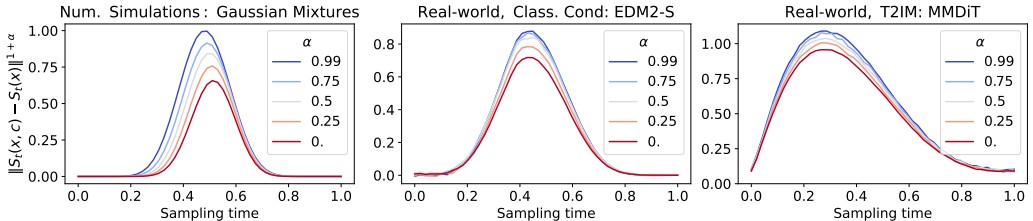

Figure 4: **The flexibility of nonlinear power-law CFG, key ingredient in our theoretical argument, appears consistently in num. simulations and real-world models. Left**: Num. simulations on GMMs. **Center**: Class-conditional EDM2-S trained on ImageNet-1K. Right: a text-to-image model with MMDiT architecture. All exhibit the same hump-shaped behavior of $\|S_t(\vec{x}, c) - S_t(\vec{x})\|^{1+\alpha}$, consistent with our theoretical analysis.

**Experimental details.** We examine power-law CFG in GMM simulations and four generative models: DiT (Peebles & Xie, 2023) and EDM2 (Karras et al., 2024b), trained and evaluated on ImageNet-1K (res. 256 and 512). We also consider two text-to-image (T2IM) models: first is trained on ImageNet-1k and CC12M (Changpinyo et al., 2021), evaluated on CC12M, using the diffusion DDPM training objective (Ho et al., 2020) with MMDiT architecture (Esser et al. (2024), similar to SD3. The second model, using MMDiT scaled to 1.6B parameters, is trained with flow matching on YFCC100M (Thomee et al., 2016), CC12M and a proprietary dataset of 320M Shutterstock images, evaluated on COCO dataset (Lin et al., 2014). Section G contains another T2IM model trained with DDPM objective with the MDTv2 (Gao et al., 2023) architecture scaled to 800M parameters.

**Comparing GMM simulations to real-world experiments.** Fig. 4 shows both GMM simulations and large-scale gen. models exhibit the same characteristic hump-shaped behavior of the amplitude of the guidance term $\|S_t(\vec{x}, c) - S_t(\vec{x})\|^{1+\alpha}$, suggesting our theoretical insights might be beneficial in practice. Crucially, the power-law param. $\alpha$ allows to alter the shape of these curves, offering the precise flexibility our analysis identifies as necessary for improving guidance. We further examine this in Fig. 35 in Appx. H, showing nonlin. CFG results in faster convergence, with paths of smaller Jensen-Shannon divergence to the target (across all time $\tau$), while also reducing the overshoot.

**Power-law CFG is robust.** We perform sensitivity analysis, showing that large values of $\alpha$ consistently yield improved performance, increasing robustness and stability when tuning for $\omega$.[2] This is shown in Fig. 5 for EDM2-S, in Section G for DiT/XL-2, two T2IM models, together with further ablation studies showing that non-lin. CFG consistently outperforms standard CFG when varying number of sampling steps.

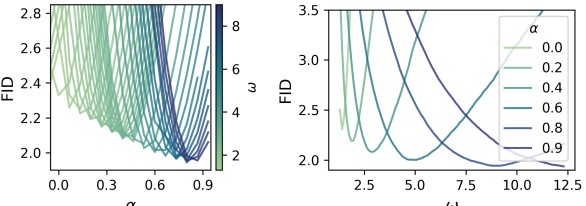

Figure 5: **Sensitivity analysis** (EDM2-S, ImageNet-1K $512 \times 512$). **Left:** Increasing parameter $\alpha$ consistently improves FID to standard CFG ($\alpha = 0.$). **Right:** Increasing $\alpha$ yields more stable FID values across a larger range of $\omega$.

**Power-law CFG improves image quality and diversity.** We quantitatively evaluate our method using FID (Heusel et al., 2017) measuring image quality, and precision and recall (Sajjadi et al., 2018) measuring diversity. In Table 1, we compare power-law CFG to state-of-the-art guidance methods. As power-law is easily combined with other guidances, we test combining it with CADS (Sadat et al., 2023) and limited-guidance (Kynkäänniemi et al., 2024), the strongest competitors. Power-law CFG improves over standard CFG in most cases and improves results of CADS and lim.-interval guidance. We provide qualitative results in Fig. 6, observing that power-law improves both quality and diversity, while being more robust to changing $\omega$.

**Latent-space vs Pixel-space.** The power-law method consistently showed robust improvements with the nonlinear parameter $\alpha = 0.9$ set across all experiments in latent space. For pixel space (see Table 11 in Appx.), the optimal $\alpha$ seemed to fluctuate slightly more and tuning $\alpha$ as well as $\omega$ resulted in stronger improvement in performance. While Power-Law demonstrates consistent

---

[2]Although power-law CFG introduces another hyerparameter, $\alpha$, we did not have to perform extensive hyperparameter search, and found large values, $e.g.$, $\alpha = 0.9$ to consistently perform well.

Table 1: **Power-law CFG often improves both fidelity and diversity metrics**. We applied power-law to standard CFG, limited and CADS variants, as the two were strongest competitors. Applying power-law improved their performance further, achieving competitive results. Best results are **bolded**, second best underlined. (↑) indicates power-law CFG improves the guidance method compared to its version with stand. CFG, while (↓) means the metric deteriorated. T2IM represents text-to-image models, CC class-conditional; FM is short for flow-matching objective and diff. stands diffusion. Experimental details are provided in Section G.

| Model | EDM2-S (CC, IM-1K 512) | | | DiT/XL-2 (CC, IM-1K 256) | | | Diff. MMDiT (T2IM, CC12m) | | | FM MMDiT (T2IM, COCO) | | |
|---|---|---|---|---|---|---|---|---|---|---|---|---|
| | FID | Precision | Recall | FID | Precision | Recall | FID | Precision | Recall | FID | Precision | Recall |
| Standard (Ho & Salimans, 2022) | 2.29 | 0.751 | 0.582 | 2.27 | 0.829 | 0.584 | 8.58 | 0.661 | 0.569 | 5.20 | 0.629 | 0.594 |
| Scheduler (Wang et al., 2024) | 2.03 | 0.762 | 0.591 | 2.14 | 0.840 | 0.614 | 8.30 | 0.681 | 0.559 | 5.00 | 0.606 | 0.623 |
| Limited (Kynkäänniemi et al., 2024) | 1.87 | 0.760 | 0.598 | 1.97 | 0.801 | 0.632 | 8.58 | 0.680 | 0.553 | 5.00 | 0.609 | 0.602 |
| Cosine (Gao et al., 2023) | 2.15 | 0.770 | 0.619 | 2.30 | **0.861** | 0.520 | 8.29 | 0.659 | 0.564 | 5.14 | 0.630 | 0.616 |
| CADS (Sadat et al., 2023) | 1.60 | **0.792** | 0.619 | 1.70 | 0.772 | 0.627 | 8.32 | **0.692** | 0.559 | 4.91 | 0.633 | 0.613 |
| APG (Sadat et al., 2024) | 2.13 | 0.756 | **0.640** | 2.11 | 0.815 | 0.628 | 8.49 | 0.661 | 0.571 | 5.23 | 0.614 | **0.631** |
| REG (Xia et al., 2024) | 1.99 | 0.761 | 0.608 | 1.76 | 0.799 | 0.601 | 8.10 | 0.673 | 0.540 | 5.06 | 0.619 | 0.619 |
| CFG++ (Chung et al., 2024) | N/A | N/A | N/A | N/A | N/A | N/A | 8.35 | 0.668 | 0.552 | 4.85 | 0.632 | 0.629 |
| Power-law CFG (Ours) | 1.93 (↓) | 0.780 (↑) | 0.631 (↑) | 2.05 (↓) | 0.831 (↓) | 0.595 (↑) | 8.11 (↑) | 0.670 (↑) | 0.553 (↓) | 4.81 (↓) | 0.621 (↓) | 0.619 (↑) |
| Power-law CFG + Limited (Ours) | 1.73 (↓) | 0.752 (↓) | 0.600 (↑) | 1.87 (↓) | 0.849 (↑) | **0.642** (↑) | 8.27 (↓) | **0.692** (↑) | 0.555 (↑) | 4.84 (↓) | 0.615 (↑) | 0.622 (↑) |
| Power-law CFG + CADS (Ours) | **1.52** (↓) | 0.770 (↓) | 0.622 (↑) | **1.63** (↓) | 0.754 (↓) | 0.639 (↑) | **7.98** (↓) | 0.690 (↓) | **0.573** (↑) | **4.71** (↓) | **0.640** (↑) | 0.624 (↓) |

Figure 6: **Qualitative comparison of Standard and Power-Law CFG** on DiT/XL-2 trained on ImageNet-1k ($256\times256$). **Left:** while standard CFG results in diversity decrease or mode collapse (first image for $\omega = 5$), power-law CFG ($\alpha = 0.9$) improves in diversity at no cost to fidelity, showing robustness to varying of $\omega$ (note very large $\omega = 10$). **Right:** Increasing non-linear parameter $\alpha$ yields larger diversity, while preserving image quality. Experimental details with further examples (as well as text-to-image) are provided in App. G.

benefits, its performance relative to other non-linear strategies remains unexplored. There may be more strategies (particularly for pixel-space), which are worth exploring.

# 6 CONCLUSION

By connecting CFG to the high-dimensional framework of diffusion regimes, we theoretically analyzed its behavior and showed that, in sufficiently high dimensions, CFG reproduces the correct target distribution—a "blessing-of-dimensionality" result. We further demonstrated that the well-known artifacts of mean overshoot and variance shrinkage emerge as dimensionality decreases. Finally, we proposed a simple nonlinear extension of CFG, proving that it mitigates both effects while preserving CFG's practical benefits, consistently improving sample quality and diversity across state-of-the-art text-to-image and class-conditional models.

**Limitations and future work.** Our theory demonstrates that in high-dimensional settings, CFG generates the correct target distribution, extending previous results showing CFG alters it in finite-dimensions. In practice, CFG improves fidelity while reducing diversity: although our theory allows discovery of guidances that maintain strong fidelity while significantly boosting diversity, the reason why CFG-modified distribution is more effective in practice is not explained by our theory which relies on perfect score estimation. We hypothesize, therefore, that the practical benefits of (non-linear) CFG might be tied to the imperfect score estimators used in practice. Investigating how score approximation errors impact guidance effectiveness is an important area for future research.

## ACKNOWLEDGEMENTS

This work has received funding from the French government, managed by the National Research Agency (ANR), under the France 2030 program with the reference ANR-23-IACL-0008. Furthermore, this paper is supported by PNRR-PE-AI FAIR project funded by the NextGeneration EU program. We would like to thank Mathurin Videau, João Maria Janeiro, Kunhao Zheng, Theophane Valleys, Wes Bouaziz and Tony Bonnaire for fruitful discussions regarding the numerical experiments. We would further like to thank Levent Sagun, David Lopez-Paz, Brian Karrer, Ricky Chen, Arnaud Doucet, Yaron Lipman and Luke Zettlemoyer for feedback and support.

## ETHICS STATEMENT

This study contributes to the growing body of research aimed at deepening our theoretical understanding of diffusion models and their broader implications for generative modeling. By bridging the gap between theory and practice, we strive to improve the performance and efficiency of these models, which have far-reaching applications in various fields.

However, as with any powerful technology, there are also potential risks associated with development and deployment of advanced generative models. The increasing sophistication of deepfakes raises concerns about misinformation, propaganda, and the erosion of trust in digital media. Moreover, the misuse of generative models for malicious purposes, such as creating fake identities or spreading disinformation, poses significant threats to society as a whole.

In light of these challenges, we hope that our paper, along with many others that aim to improve understanding of the models, will contribute to a deeper understanding of their strengths and limitations. We believe it is essential for developing effective strategies to mitigate the risks associated with generative models, and we hope that our work will be a step toward achieving this goal.

## REPRODUCIBILITY STATEMENT

We have clearly stated the main assumptions underlying our work, along with their limitations and how they influence our conclusions. To support our theoretical contributions, Appendices B-E contain the full proofs of our claims, while Appendix G.3 lists all hyperparameter configurations required to reproduce our experiments exactly. For real-world experiments, we specify the GPUs used as well as the Gflops of the models, offering transparency in the computational resources required. Finally, our experiments on class-conditional models rely on and reference publicly available checkpoints, enabling straightforward verification and further exploration by the community.

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

SUPPLEMENTARY MATERIAL

The supplementary material is structured as follows:

- In Section A, we give a brief introduction to related work, focusing on Biroli et al. (2024).
- In Section B, we give proofs for two equidistant, symmetric Gaussian mixtures.
- In Section C, we present arguments how to extend the proofs to non-centered Gaussian mixtures (subsec. C.1) and multiple Gaussian mixtures (subsec. C.2).
- In Section D, we present the theoretical and numerical findings for finite dimension (including low dimension $d$).
- In Section E, we provide the arguments showing that the nonlinear Power-Law CFG improves mean overshoot and variance shrinkage.
- In Section F, we present experimental details for Gaussian mixture numerical simulations.
- In Section G, we provide experimental details involving real-world experiments.
- In Section H, we propose another non-linear CFG alternative and provide num. experiments.

## A INTRODUCTION TO RELATED WORK: CLASSIFIER-FREE GUIDANCE (CFG) AND SPECIFICATION TIME IN THE HIGH-DIMENSIONAL LIMIT

We start by briefly introducing the calculation required for estimating the speciation time $t_{\mathrm{s}}$ for a case of two equally weighted Gaussians. This section is a direct adaptation of the framework introduced by Biroli et al. (2024). The diffusion process, consisting of $d$ independent Ornstein-Uhlenbeck Langevin equations, reads as follows (using $f(t) = -1$ and $g(t) = \sqrt{2}$ in Eq. (1)):

$$d\vec{x}(t) = -\vec{x}dt + d\vec{B}(t), \tag{9}$$

where $d\vec{B}(t)$ equals the square root of two times the standard Brownian motion in $\mathbb{R}^d$. At time $t = 0$, the process starts from the probability distribution $P_0(\vec{a})$, consisting of two Gaussian clusters that have means at $\pm\vec{m}$ and share the same variance $\sigma^2$. To guarantee that these Gaussians remain distinct in high-dimensional space, we assume that $|\vec{m}|^2 = d\tilde{\mu}^2$, where both $\sigma$ and $\tilde{\mu}$ are of order 1.

As the process evolves, the emergence of speciation resembles symmetry breaking observed during thermodynamic phase transitions. A common approach to analyzing this phenomenon is to construct a perturbative expansion of the free energy as a function of the field. Therefore, Biroli et al. (2024) derive an expression for $\log P_t(\vec{x})$ using a perturbative expansion in terms of $e^{-t}$, which is valid for large time values. This method is justified since speciation occurs at large times.

One can rewrite the probability to be at $\vec{x}$ at time $t$ as

$$P_t(\vec{x}) = \int d\vec{a} P_0(\vec{a}) \frac{1}{\sqrt{2\pi\Delta_t^d}} \exp\left(-\frac{1}{2}\frac{(\vec{x} - \vec{a}e^{-t})^2}{\Delta_t}\right)$$

$$= \frac{1}{\sqrt{2\pi\Delta_t}} \exp\left(-\frac{1}{2}\frac{\vec{x}^2}{\Delta_t} + g(\vec{x})\right),$$

where the function $g(\vec{x})$, defined as

$$g(\vec{x}) = \log \int d\vec{a} P_0(\vec{a}) \exp\left(-\frac{1}{2}\frac{\vec{a}^2 e^{-2t}}{\Delta_t}\right) \exp\left(\frac{e^{-t}\vec{x} \cdot \vec{a}}{\Delta_t}\right)$$

can be viewed through a field-theoretic (or equivalently, a probabilistic) approach, where it serves as a generative function for connected correlations among the variables $\vec{a}$ (Zinn-Justin, 2021). By

expanding this function at large times, one can show:

$$g(\vec{x}) = \frac{e^{-t}}{\Delta_t} \sum_{i=1}^{d} x_i \langle a_i \rangle + \frac{1}{2} \frac{e^{-2t}}{\Delta_t^2} \sum_{i,j=1}^{d} x_i x_j \left[ \langle a_i a_j \rangle - \langle a_i \rangle \langle a_j \rangle \right] + O\left( \left( x e^{-t} \right)^3 \right),$$

where we utilize the brackets $\langle \cdot \rangle$ to denote the expectation value with respect to the effective distribution $P_0(\vec{a}) e^{-\vec{a}^2 e^{-2t}/(2\Delta_t)}$. Therefore, the expansion can be used to show that at large times:

$$\log P_t(\vec{x}) = C + \frac{e^{-t}}{\Delta_t} \sum_{i=1}^{d} x_i \langle a_i \rangle - \frac{1}{2\Delta_t} \sum_{i,j=1}^{d} x_i M_{ij} x_j + O\left( \left( x e^{-t} \right)^3 \right),$$

where $C$ is an $\vec{x}$-independent term and

$$M_{ij} = \delta_{ij} - e^{-2t} \left[ \langle a_i a_j \rangle - \langle a_i \rangle \langle a_j \rangle \right].$$

The curvature of $\log P_t(\vec{x})$ is closely linked to the spectral properties of the matrix $M$. In the large time regime, $M$ approaches the identity matrix, and consequently, all its eigenvalues are positive. However, a qualitative shift in shape occurs at the maximum time $t_s$, where the largest eigenvalue of $M$ transitions through zero. This marks the onset of the **speciation time**, distinguished by a change in curvature of the effective potential $-\log P_t(\vec{x})$. In this case, it can be easily computed: the matrix $M$ is given by $M_{ij} = \left( 1 - \sigma^2 e^{-2t} \right) \delta_{ij} - e^{-2t} m_i m_j$ and its largest eigenvalue is ( $1 - \sigma^2 e^{-2t} - d\tilde{\mu}^2 e^{-2t}$ ). We get therefore in the large $d$ limit $t_s = \frac{1}{2} \log \left( d\tilde{\mu}^2 \right)$ which up to subleading corrections identifies the speciation timescale as

$$t_s = \frac{1}{2} \log(d).$$

## B  THEORETICAL PROOFS: TWO EQUIDISTANT, SYMMETRIC GAUSSIAN MIXTURES

### ASYMPTOTIC STOCHASTIC PROCESS IN REGIME I AND SYMMETRY BREAKING

In the limit of large dimensions, a comprehensive analytical examination of the dynamics in Regime I, taking place on time-scales $t_s + O(1) = (1/2) \log d + O(1)$, can be provided, specifically at the beginning of the backward process. Assuming no collapse (for further details, refer to Biroli et al. (2024)), an investigation into diffusion dynamics shows that the empirical distribution $P_t^e(\vec{x})$ at time $t$ can be approximated with high accuracy by $P_t(\vec{x})$. This approximation represents the convolution of the initial distribution $P_0$, comprising a mixture of Gaussians centered at $\pm \vec{m}$, and a diffusion kernel proportional to $e^{-\left( \vec{x} - \vec{a} e^{-t} \right)^2 / 2}$. Consequently, the explicit expression for this approximation is

$$P_0(\vec{x}) = \frac{1}{2 \left( \sqrt{2\pi\sigma^2} \right)^d} \left[ e^{-(\vec{x}-\vec{m})^2/\left( 2\sigma^2 \right)} + e^{-(\vec{x}+\vec{m})^2/\left( 2\sigma^2 \right)} \right], \text{ and} \qquad (10)$$

$$P_t(\vec{x}) = \frac{1}{2 \left( \sqrt{2\pi\Gamma_t} \right)^d} \left[ e^{-\left( \vec{x} - \vec{m} e^{-t} \right)^2 / (2\Gamma_t)} + e^{-\left( \vec{x} + \vec{m} e^{-t} \right)^2 / (2\Gamma_t)} \right]$$

where $\Gamma_t = \sigma^2 e^{-2t} + \Delta_t$ goes to 1 at large times. The log of this probability is

$$\log P_t(\vec{x}) = -\frac{\vec{x}^2}{2\Gamma_t} + \log \cosh \left( \vec{x} \cdot \vec{m} \frac{e^{-t}}{\Gamma_t} \right),$$

and hence the score reads

$$S_t^i(\vec{x}) = -\frac{x^i}{\Gamma_t} + m_i \frac{e^{-t}}{\Gamma_t} \tanh\left(\vec{x} \cdot \vec{m} \frac{e^{-t}}{\Gamma_t}\right). \tag{11}$$

As there are two classes: $+\vec{m}$ and $-\vec{m}$, the score conditioned to one class equals the score associated to a given Gaussian. Therefore, for the two classes we have:

$$
\begin{aligned}
+\vec{m} &: S_t^i(\vec{x}, +) = \frac{-x^i + m_i e^{-t}}{\Gamma_t}, \text{and} \\
-\vec{m} &: S_t^i(\vec{x}, -) = \frac{-x^i - m_i e^{-t}}{\Gamma_t}.
\end{aligned}
\tag{12}
$$

### B.1   RESULT I: WHAT IS THE ROLE OF CLASSIFIER-FREE GUIDANCE?

Let us first analyze the "transverse" directions $\vec{v} \perp \vec{m}$. For these directions, for all $\omega$, the score is the same and equals $\vec{S}_t^{\text{CFG}}(\vec{x}, c) \cdot \vec{v} = -\frac{\vec{x} \cdot \vec{v}}{\Gamma_t}$. Let us project the backward Eq. (2) on a unit vector $\vec{v} \perp \vec{m}$. We write $p = \vec{x} \cdot \vec{v}$, and the backward equation now reads $dp = p(1 - 2/\Gamma_{t_f - \tau})d\tau + \sqrt{2}dB$ which is the backward equation for a single Gaussian variable. When $\tau \to t_f$ the projection $p = \vec{x} \cdot \vec{v}$ is thus distributed as $\mathcal{N}(0, \sigma^2)$, for all values of $\omega$.

Therefore, as all the components except the one in the $\vec{m}$ direction are not affected, we can consider only the component along $\vec{m}$:

$$\vec{S}_{t_{CFG}}(\vec{x}, c) \cdot \frac{\vec{m}}{|\vec{m}|} = -\frac{\vec{x} \cdot \vec{m}/|\vec{m}|}{\Gamma_t} + \omega \frac{|\vec{m}|^2 e^{-t}}{|\vec{m}|\Gamma_t} \cdot \left\{c - \tanh\left(\frac{\vec{x} \cdot \vec{m} e^{-t}}{\Gamma_t}\right)\right\} + \frac{|\vec{m}|e^{-t}c}{\Gamma_t}.$$

By denoting $\frac{\vec{x} \cdot \vec{m}}{|\vec{m}|} = q(t)$, where $|\vec{m}| = \sqrt{d}$, we can obtain the backward equation:

$$dx^i = (x^i + 2S_{\tau_{CFG}}^i)d\tau + d\eta_i(\tau),$$

where $\tau = t_f - t$, i.e., the backward time. Therefore, we can obtain for Regime I and by projecting onto the $\frac{\vec{m}}{|\vec{m}|}$ direction, we have that:

$$dq = dx^i \cdot \frac{\vec{m}}{|\vec{m}|} = \left(q + 2\left[-q + e^{-(t_f - t_s - \tau)}\left((1 + \omega) - \omega \tanh\left(qe^{-(t_f - t_s - \tau)}\right)\right)\right]\right)d\tau + d\eta(\tau),$$

as, in Regime I, we have that $\Gamma_t \approx 1$, and also $\sqrt{d} = e^{-t_s}$.

Again, from this point onward by $t(\tau)$ we denote the backward time for ease of notation. This is like having an effective potential:

$$dq = -\frac{\partial V^{\text{CFG}}(q, \tau)}{\partial q}d\tau + d\eta(\tau),$$

where

$$
\begin{aligned}
V^{\text{CFG}} &= \frac{1}{2}q^2 + 2\left[-(1 + \omega)cqe^{-(t - t_s)} + \omega \ln\cosh\left(qe^{-(t - t_s)}\right)\right] \\
&= \underbrace{(\frac{1}{2}q^2 - 2e^{-(t - t_s)}cq)}_{\text{Classifier's potential}} + \omega \underbrace{\left[-cqe^{-(t - t_s)} + \ln\cosh\left(qe^{-(t - t_s)}\right)\right]}_{\text{Extra potential } V_{\text{extra}}}.
\end{aligned}
$$

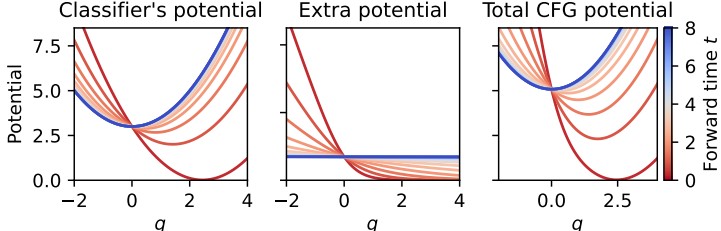

Figure 7: **Effect of CFG on the guiding potential of a Gaussian mixture.** The backward diffusion for the variable $q$ giving the projection of $\vec{x}$ on the center $\vec{m}$ of the Gaussian where one wants to guide the backward diffusion. From left to right: Potential within the class, CFG-added-potential $V_{\text{extra}}$ with $\omega = 2$, and their sum as in Eq. (6). CFG exhibits faster convergence to the target ($t = 0$), but results in narrower potential for small $t$ (with $t$ ranging from 0 to 8, as indicated on the right panel).

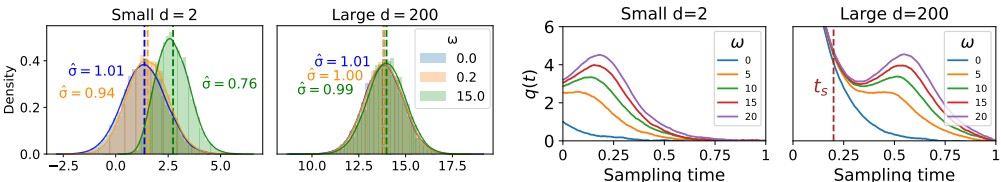

Figure 8: **Left: CFG produces the exact target distribution in high dimensions.** We simulate the backward process using a two Gaussian mixture. We project and plot the generated samples onto the target mean $+\vec{m}$: $q(t = 0) = \vec{x} \cdot \vec{m}/|\vec{m}|$. For small $d = 2$, CFG generates a distribution with larger magnitude mean (dashed line) and smaller variance than the target one (for $\omega = 0$.). This effect diminishes as the dimension increases: for $d = 200$ it is practically absent. **Right: High-dimensionality of the data allows CFG trajectories to align.** We plot the evolution of the mean of trajectories $q(t)$: starting at large forward times denoted with $t = 1$ (noise), for small $d = 2$, CFG trajectories do not align with the unconditional trajectories at $t = 0$ (data) causing the CFG overshoot. For large dimension $d = 200$, the high-dimensionality of the data allows trajectories to realign with the unguided one at speciation time $t_s$, resulting in the correct target distribution.

Therefore, for class $c = +1$ (equivalently for $c = -1$), there is little effect for $qe^{-(t-t_s)} \gg 1$, as then $-qe^{-(t-t_s)} + \ln \cosh \left( qe^{-(t-t_s)} \right) \approx 0$. Instead, for $qe^{-(t-t_s)} \sim O(1)$, we have that $-qe^{-(t-t_s)} + \ln \cosh \left( qe^{-(t-t_s)} \right) \sim O(1)$. Therefore, we can conclude our first result:

**Result I.** In Regime I, before speciation time $t_s$, CFG is effective in aiding class selection and speeds up the convergence towards the target class $c$.

The utility of CFG is therefore to "push" in the right direction in Regime I where arguably the class-based score/potential is likely not accurate in the rare region ($q > 0$ for $c = -1$ and $q < 0$ for $c = +1$). The behavior of the two potentials is displayed in Figure 7.

## B.2 RESULT II: PATH ALIGNMENT

The role of CFG in Regime I is to push the trajectories more in the direction of the selected class. We recall that the SDE verified by $q$ when pushed towards class $c = +1$ reads:

$$dq = \left( q + 2 \left[ -q + e^{-(t_f - t_s - \tau)} \left( (1 + \omega) - \omega \tanh \left( qe^{-(t_f - t_s - \tau)} \right) \right) \right] \right) d\tau + d\eta(\tau), \quad (13)$$

For large times but still during Regime I, i.e. $t_f - t_s \ll \tau \ll 1/2 \log d$, $q$ is very large (positive or negative). In this regime the CFG term can be neglected as it leads to exponentially small corrections to the SDE (of order $e^{-2qe^{-(t_f - t_s - \tau)}}$) with $t_f - t_s - \tau \gg 1$. In consequence, in Regime I at large times, the SDE just reads:

$$dq = -q + 2e^{-(t_f - t_s - \tau)} + d\eta(\tau),$$

The effect of CFG is to lead to different values of $q$ when entering this late regime of Regime I. We call these values $q(\tau_i)$ and denote $\tau_i$ the fixed time at which the CFG contribution can be neglected. The value $q(\tau_i)$ is quickly (exponentially) forgotten when $\tau$ departs from $\tau_i$, i.e., the evolution readjust to the correct value without CFG. This can be shown by solving the SDE starting from a given $\tau_i$:

$$q(\tau) = q(\tau_i)e^{-(\tau-\tau_i)} + e^{-(t_f-t_s)}\left(e^{\tau} - e^{-\tau+2\tau_i}\right) + \sqrt{1 - e^{-(2(\tau-\tau_i))}}z_\tau$$

where $z_\tau$ is a Gaussian variable with mean zero and unit variance. When $\tau \gg \tau_i$ but still in Regime I the solution of the SDE does not depend any longer on $q(\tau_i)$ and it coincides statistically with the one of the backward process of the single Gaussian corresponding to the class $c = +1$. This allows to conclude the second result:

**Result II.** Just before speciation time $t_s$, CFG-guided paths realign with the unguided path that generates the correct, unmodified target distribution.

### B.3 RESULT III: WHEN DOES CLASSIFIER-FREE GUIDANCE TAKE EFFECT?

We can proceed to answer this question by examining the classifier-free guidance score, as defined in Ho & Salimans (2022):

$$S^i_{t_{CFG}}(\vec{x}, c) = (1 + \omega)S^i_t(\vec{x}, c) - \omega S^i_t(\vec{x}), \tag{14}$$

where $c = \pm 1$ and $\omega > 0$. By plugging in the cond. (12) and uncond. scores (11), we can obtain:

$$\begin{aligned}
S^i_{t_{CFG}}(\vec{x}, c) &= -\frac{x^i}{\Gamma_t} + (1 + \omega)\frac{cm_ie^{-t}}{\Gamma_t} - \omega\frac{m_ie^{-t}}{\Gamma_t}\tanh\left(\frac{\vec{x}\cdot\vec{m}e^{-t}}{\Gamma_t}\right) \\
&= -\frac{x^i}{\Gamma_t} + \omega\frac{m_ie^{-t}}{\Gamma_t}\left\{c - \tanh\left(\frac{\vec{x}\cdot\vec{m}e^{-t}}{\Gamma_t}\right)\right\} + \frac{cm_ie^{-t}}{\Gamma_t}.
\end{aligned} \tag{15}$$

Now, in Regime II, when the trajectory has committed to a given class, $\vec{x}\cdot\vec{m} \sim O(d)$ and $\text{sign}(\vec{x}\cdot\vec{m}) = c$. Therefore, $c - \tanh\left(\frac{\vec{x}\cdot\vec{m}e^{-t}}{\Gamma_t}\right) \approx 0$, and one finds from (15), that $S^i_{t_{CFG}}(\vec{x}, c) = S^i_t(\vec{x})$. This implies that, within this regime, classifier-free guidance equals the conditional score. Therefore, Classifier free-guidance only affects Regime I, as $S^i_{t_{CFG}}(\vec{x}, c) = S^i_t(\vec{x})$ for $t > t_s = \frac{1}{2}\log(d)$. This allows us to conclude the third result:

**Result III.** In Regime II, after speciation time $t_s$, CFG has no effect on the generation process.

## C  GENERALIZATIONS OF THE PROOF

In this section, we present arguments for extending our proofs to more general cases. We start by discussing proof generalization for non-centered Gaussian mixtures (Section C.1) and then move on to a mixture of four Gaussians (C.2). Finally, we conclude with some remarks on how to further extend these results to more complex scenarios.

### C.1  GENERALIZATION TO NON-CENTERED GAUSSIAN MIXTURES

#### ASYMPTOTIC STOCHASTIC PROCESS IN REGIME I AND SYMMETRY BREAKING

Here we provide an example on how to generalize the study of Gaussian mixtures to the case where the two Gaussians are centered in $\vec{m}_1$ and $\vec{m}_2$. We take $\vec{m}_1, \vec{m}_2$ as two arbitrary vectors in $d$ dimensions, on the sphere $|\vec{m}_c|^2 = d$ the case where they have different norms, both scaling proportionally to $d$, could be studied as well with the same formalism.

The initial probability density is

$$P_0(\vec{x}) = \frac{1}{2\left(\sqrt{2\pi\sigma^2}\right)^d} \left[ e^{-(\vec{x}-\vec{m}_1)^2/(2\sigma^2)} + e^{-(\vec{x}-\vec{m}_2)^2/(2\sigma^2)} \right], \text{ and} \tag{16}$$

$$P_t(\vec{x}) = \frac{1}{2\left(\sqrt{2\pi\Gamma_t}\right)^d} \left[ e^{-\left(\vec{x}-\vec{m}_1 e^{-t}\right)^2/(2\Gamma_t)} + e^{-\left(\vec{x}-\vec{m}_2 e^{-t}\right)^2/(2\Gamma_t)} \right]$$

where $\Gamma_t = \sigma^2 e^{-2t} + \Delta_t$ goes to 1 at large times. The log of this probability is

$$\log P_t(\vec{x}) = -\frac{\vec{x}^2}{2\Gamma_t} + \log\left( e^{\vec{x}\cdot\vec{m}_1 \frac{e^{-t}}{\Gamma_t}} + e^{\vec{x}\cdot\vec{m}_2 \frac{e^{-t}}{\Gamma_t}} \right) + C,$$

where $C$ is a constant, and hence the score reads

$$S_t^i(\vec{x}) = -\frac{x^i}{\Gamma_t} + \frac{e^{-t}}{\Gamma_t} \frac{m_1^i\, e^{\vec{x}\cdot\vec{m}_1 \frac{e^{-t}}{\Gamma_t}} + m_2^i\, e^{\vec{x}\cdot\vec{m}_2 \frac{e^{-t}}{\Gamma_t}}}{e^{\vec{x}\cdot\vec{m}_1 \frac{e^{-t}}{\Gamma_t}} + e^{\vec{x}\cdot\vec{m}_2 \frac{e^{-t}}{\Gamma_t}}} \tag{17}$$

As there are two classes: $\vec{m}_1$ and $\vec{m}_2$, the score conditioned to one class equals the score associated to a given Gaussian. Therefore, for the two classes we have:

$$\begin{aligned}
\vec{m}_1 &: S_t^i(\vec{x},+) = \frac{-x^i + m_1^i e^{-t}}{\Gamma_t}, \text{ and} \\
\vec{m}_2 &: S_t^i(\vec{x},-) = \frac{-x^i - m_2^i e^{-t}}{\Gamma_t}.
\end{aligned} \tag{18}$$

WHAT IS THE ROLE OF CLASSIFIER-FREE GUIDANCE?

We shall use as basis the vectors $\vec{m}_+ = (\vec{m}_1 + \vec{m}_2)/2$, $\vec{m}_- = (\vec{m}_1 - \vec{m}_2)/2$, and we shall denote by $\vec{v}$ the vectors orthogonal to the place generated by $\vec{m}_1, \vec{m}_2$.

For these "transverse" directions $\vec{v} \perp (\vec{m}_1, \vec{m}_2)$. for all $\omega$, the score is the same and equals $\vec{S}_t^{\text{CFG}}(\vec{x}, c) \cdot \vec{v} = -\frac{\vec{x}\cdot\vec{v}}{\Gamma_t}$. Let us project the backward equation on a unit vector $\vec{v}$ in the transverse space. We write $p = \vec{x} \cdot \vec{v}$, and the backward equation now reads $dp = p(1 - 2/\Gamma_{t_f-\tau})d\tau + \sqrt{2}dB$ which is the backward equation for a single Gaussian variable. When $\tau \to t_f$ the projection $p = \vec{x}\cdot\vec{v}$ is thus distributed as $\mathcal{N}(0, \sigma^2)$, for all values of $\omega$.

Therefore, as all the components except the ones in the $\vec{m}_+$ and $\vec{m}_-$ directions are not affected.

We now project the score on $\vec{m}_+$ and $\vec{m}_-$, using $\vec{m}_+.\vec{m}_- = 0$, $\vec{m}_+.\vec{m}_1 = \vec{m}_+.\vec{m}_2 = d^2/2$ and $\vec{m}_-.\vec{m}_1 = -\vec{m}_-.\vec{m}_2 = d^2/2$:

$$\vec{S}_{t_{CFG}}(\vec{x}, c) \cdot \vec{m}_+ = \frac{(\vec{m}_+ e^{-t} - \vec{x}) \cdot \vec{m}_+}{\Gamma_t}$$

$$\vec{S}_{t_{CFG}}(\vec{x}, c) \cdot \vec{m}_- = \frac{(\vec{m}_- e^{-t} - \vec{x}) \cdot \vec{m}_-}{\Gamma_t} + \omega \frac{|\vec{m}_-|^2 e^{-t}}{\Gamma_t} \cdot \left\{ 1 - \tanh\left( \frac{\vec{x} \cdot \vec{m}_- e^{-t}}{\Gamma_t} \right) \right\}$$

Inserting these scores into the backward diffusion equation, one finds that:

- $\vec{x}.\vec{m}_+/|\vec{m}_+|$ evolves as a Gaussian variable. At time $\tau \to t_f$ the distribution of this variable is $\mathcal{N}(|\vec{m}_+|, \sigma^2)$.

- The variable $q_-(t) = \frac{\vec{x}\cdot\vec{m}_-}{|\vec{m}_-|}$ satisfies the same equation as the variable $q(t)$ which we analyzed in the 'centered' case where $\vec{m}_1 = -\vec{m}_2 = \vec{m}$

Therefore, we can conclude that in this case, CFG has the same effect: it is effective in aiding class selection, speeding up the convergence toward the correct target class $c$.

WHEN DOES CLASSIFIER-FREE GUIDANCE TAKE EFFECT?

We can proceed to answer this question by examining the classifier-free guidance score:

$$S^i_{t_{CFG}}(\vec{x}, c) = (1 + \omega)S^i_t(\vec{x}, c) - \omega S^i_t(\vec{x}), \tag{19}$$

where $c \in \{1, 2\}$ and $\omega > 0$. The CFG score guiding to class $c = 1$ is thus:

$$S^i_{t_{CFG}}(\vec{x}, c) = -\frac{x^i}{\Gamma_t} + (1 + \omega)\frac{m^i_1 e^{-t}}{\Gamma_t} - \omega\frac{e^{-t}}{\Gamma_t} m^i_1 \frac{e^{\vec{x} \cdot \vec{m}_1 \frac{e^{-t}}{\Gamma_t}} + m^i_2 e^{\vec{x} \cdot \vec{m}_2 \frac{e^{-t}}{\Gamma_t}}}{e^{\vec{x} \cdot \vec{m}_1 \frac{e^{-t}}{\Gamma_t}} + e^{\vec{x} \cdot \vec{m}_2 \frac{e^{-t}}{\Gamma_t}}} \tag{20}$$

Now, in Regime II, when the trajectory has committed to a given class say class $1*$, $\vec{x} \cdot \vec{m}_1 - \vec{x} \cdot \vec{m}_2$ is positive and of order $O(d)$. Therefore $S^i_{t_{CFG}}(\vec{x}, c) = S^i_t(\vec{x}, c)$. This implies that, within this regime, classifier-free guidance equals the conditional score. Therefore, Classifier free-guidance only affects Regime I, as $S^i_{t_{CFG}}(\vec{x}, c) = S^i_t(\vec{x})$ for $t > t_s = \frac{1}{2}\log(d)$. This allows us to conclude that in Regime II, CFG is innocuous.

Therefore all the results obtained for the centered case $\vec{m}_1 = -\vec{m}_2 = \vec{m}$ also hold for the more general case when the two Gaussians are centered in $\vec{m}_1$ and $\vec{m}_2$.

## C.2 EXTENSION TO THE MIXTURE OF FOUR GAUSSIANS

Here we present the computation for a mixture of four Gaussians, in order to analyze the behavior of the system for an increasing number of classes and emphasize the extendability of our framework. As before, assuming no collapse, we can approximate the empirical distribution $P^e_t(\vec{x})$ at time $t$ by $P_t(\vec{x})$ with high accuracy. In this case, the approximation represents the convolution of the initial distribution $P_0$, being a mixture of 4 Gaussians centered at $\pm\vec{\mu}_1 \pm \vec{\mu}_2$, s.t. $\vec{\mu}_1 \cdot \vec{\mu}_2 = 0$, and a diffusion kernel proportional to $e^{-\left(\vec{x} - \vec{a}e^{-t}\right)^2/2}$. The explicit expression for the distribution is:

$$P_0(\vec{x}) = \frac{1}{4\left(\sqrt{2\pi\sigma^2}\right)^d}\left[e^{-\left(\vec{x} - (\vec{\mu}_1 - \vec{\mu}_2)\right)^2/\left(2\sigma^2\right)} + e^{-\left(\vec{x} - (\vec{\mu}_1 + \vec{\mu}_2)\right)^2/\left(2\sigma^2\right)}\right.$$

$$\left. + e^{-\left(\vec{x} + (\vec{\mu}_1 - \vec{\mu}_2)\right)^2/\left(2\sigma^2\right)} + e^{-\left(\vec{x} + (\vec{\mu}_1 + \vec{\mu}_2)\right)^2/\left(2\sigma^2\right)}\right]$$

and

$$P_t(\vec{x}) = \frac{1}{4\left(\sqrt{2\pi\Gamma_t}\right)^d}\left[e^{-\left(\vec{x} - (\vec{\mu}_1 - \vec{\mu}_2)e^{-t}\right)^2/(2\Gamma_t)} + e^{-\left(\vec{x} - (\vec{\mu}_1 + \vec{\mu}_2)e^{-t}\right)^2/(2\Gamma_t)}\right.$$

$$\left. + e^{-\left(\vec{x} + (\vec{\mu}_1 - \vec{\mu}_2)e^{-t}\right)^2/(2\Gamma_t)} + e^{-\left(\vec{x} + (\vec{\mu}_1 + \vec{\mu}_2)e^{-t}\right)^2/(2\Gamma_t)}\right]$$

where $\Gamma_t = \sigma^2 e^{-2t} + \Delta_t$ goes to 1 at large times. This can be rewritten as:

$$P_t(\vec{x}) = \frac{1}{2\left(\sqrt{2\pi\Gamma_t}\right)^d}e^{-(\vec{x}^2 + \vec{\mu}^2_1 e^{-2t} + \vec{\mu}^2_2 e^{-2t})/(2\Gamma_t)}\left[e^{-\vec{\mu}_1 \cdot \vec{\mu}_2 e^{-2t}/\Gamma_t}\cosh\left(\vec{x} \cdot (\vec{\mu}_1 + \vec{\mu}_2)\frac{e^{-t}}{\Gamma_t}\right)\right.$$

$$\left. + e^{\vec{\mu}_1 \cdot \vec{\mu}_2 e^{-2t}/\Gamma_t}\cosh\left(\vec{x} \cdot (\vec{\mu}_1 - \vec{\mu}_2)\frac{e^{-t}}{\Gamma_t}\right)\right]$$

The log of this probability is:

$$\log P_t(\vec{x}) = \frac{-\vec{x}^2}{2\Gamma_t} + \log\left(e^{-\vec{\mu}_1 \cdot \vec{\mu}_2 e^{-2t}/\Gamma_t}\cosh\left(\vec{x} \cdot (\vec{\mu}_1 + \vec{\mu}_2)\frac{e^{-t}}{\Gamma_t}\right) + e^{\vec{\mu}_1 \cdot \vec{\mu}_2 e^{-2t}/\Gamma_t}\cosh\left(\vec{x} \cdot (\vec{\mu}_1 - \vec{\mu}_2)\frac{e^{-t}}{\Gamma_t}\right)\right)$$

And the score reads:

$$S_t^i(\vec{x}) = \frac{-x^i}{\Gamma_t}$$

$$+ \frac{e^{-t}}{\Gamma_t} \frac{(\vec{\mu}_1 + \vec{\mu}_2)_i e^{-\vec{\mu}_1 \cdot \vec{\mu}_2 e^{-2t}/\Gamma_t} \sinh\left(\vec{x} \cdot (\vec{\mu}_1 + \vec{\mu}_2)\frac{e^{-t}}{\Gamma_t}\right) + (\vec{\mu}_1 - \vec{\mu}_2)_i e^{\vec{\mu}_1 \cdot \vec{\mu}_2 e^{-2t}/\Gamma_t} \sinh\left(\vec{x} \cdot (\vec{\mu}_1 - \vec{\mu}_2)\frac{e^{-t}}{\Gamma_t}\right)}{e^{-\vec{\mu}_1 \cdot \vec{\mu}_2 e^{-2t}/\Gamma_t} \cosh\left(\vec{x} \cdot (\vec{\mu}_1 + \vec{\mu}_2)\frac{e^{-t}}{\Gamma_t}\right) + e^{\vec{\mu}_1 \cdot \vec{\mu}_2 e^{-2t}/\Gamma_t} \cosh\left(\vec{x} \cdot (\vec{\mu}_1 - \vec{\mu}_2)\frac{e^{-t}}{\Gamma_t}\right)} e$$

As $\vec{x}$ approaches one of the means $\pm\vec{\mu}_1 \pm \vec{\mu}_2$, the second summand reduces to $(\vec{\mu}_1 \pm \vec{\mu}_2)\tanh\left(x \cdot (\vec{\mu}_1 \pm \vec{\mu}_2)\frac{e^{-t}}{\Gamma_t}\right)$[3], resulting in an expression akin to the one for mixture of 2 Gaussians in (11).

### C.3 CONCLUSION AND FURTHER EXTENSIONS

The results above can be generalized to any finite number of Gaussians, centered around $\vec{m}_i$ where $\vec{m}_i$ is a vector of norm $\mu_i \sqrt{d}$. CFG will only have effect on space spanned by vectors $\vec{m}_i$ and only in regime I. One can also consider non-isotropic Gaussians. As long as the covariance has eigenvalues not scaling with $d$, the backward process displays the two distinct regimes I and II, which is examined in detail for the mixture of two Gaussians. This result can be obtained by analyzing the forward process. The key point is that on all times of order one the noised Gaussian mixture still consists in non-overlapping Gaussian (regime II). On times of order one close to the speciation time $1/2 \log d$ the Gaussians overlap and the center are of the same order of the noise (regime I). Because of the existence of these two regimes, the general arguments presented at the beginning of the paper hold and CFG does reproduce the correct distribution in the large $d$ limit.

The results derived for the Gaussian mixture model provide a foundation for broader application. We also note two key points regarding the robustness and scope of this analysis:

**Robustness and Generalization Beyond Gaussian Mixtures**: The core features of regimes I and II persist even in more complex, high-dimensional settings, extending beyond the specific case of Gaussian mixtures. Several studies Ventura et al. (2024); Achilli et al. (2025); Bae et al. (2024) have demonstrated that these regimes apply more broadly to models where data is distributed on manifolds, among other configurations. This robustness is a direct consequence of the infinite-dimensional limit, a phenomenon also observed in other domains like supervised learning.

**Scope and Future Work**: The spirit of our approach is twofold: (i) To analyze CFG in the tractable yet rich setting of infinite dimensions. (ii) To start with the simplest nontrivial case—a mixture of two Gaussians—to characterize the fundamental mechanisms that can guide future work.

As noted, a natural and straightforward extension is to consider mixtures involving any finite number of Gaussians. Further generalizations—such as data supported on hidden manifolds—can be pursued by following the methodologies established in studies like Bae et al. (2024) and George et al. (2025).

### C.4 RELATION TO OTHER MODELS

In this section we briefly note how our theoretical results connect to a broader range of methods related to diffusion and flow matching, specifically how nonlinear CFG applies to 1-step flow models (Chen et al., 2025a) and methods that learn optimal source distributions (Lee et al., 2023).

**Regarding one-step models.** Since the main distinction between regimes lies in whether class membership is decided, a 1-step flow model effectively jumps from the initial point (part of Regime I) directly into Regime II (in fact, directly into the third regime described in Chen et al. (2025a)). However, this does not pose a problem as theoretically the difference between the conditional and unconditional score equals zero (under the correct score assumption and in sufficiently high dimension); thus, regardless whether CFG is applied or not these models obtain the correct target distribution. In practice, we believe that the models could still benefit from non-linear guidances.

---

[3]For large values of $x \cdot (\vec{\mu}_1 \pm \vec{\mu}_2)e^{-t}/\Gamma_t$, we utilized the *log-sum-exp trick* to calculate the value of the fraction.

**Regarding methods that learn an optimal source distribution for flows** Methods such as FastODE (Lee et al., 2023) that successfully map similar inputs to the same class indeed shorten, or even eliminate Regime I—for example, if the initial distribution is a Gaussian mixture (GM) where each mixture has successfully been mapped to a distinct class, e.g., by using GM-Flow matching (Lee et al., 2023), Regime I would be eliminated altogether. One should note that in these cases it is likely that CFG would not prove as beneficial. In fact, this is exactly the finding of the authors in Lee et al. (2023), in which an alternative guidance method is proposed as CFG was not found to perform as well. However, our nonlinear guidance retains an advantage: the flexibility of $\alpha > 0$ vs. $\alpha < 0$. In our experiments (starting from standard Gaussian noise with significant class overlap in the initial distribution), the former proved beneficial by amplifying guidance when score differences are large and suppressing CFG when small. It would be interesting future work to test whether indeed $\alpha < 0$ would be beneficial in settings with prominent separations in the initial distributions, as might occur in FastODE-like mappings or GM-Flows (Chen et al., 2025a).

## D FINITE DIMENSION

In this section, we give exact analyses describing the effect of CFG in finite- (possibly low-) dimensional settings, outlined in Section 4.2 in the main manuscript. We start the backward equation at a time $t_f$ large enough that the distribution of $x$ is a isotropic Gaussian with variance one. The backward equation for $x(t)$ with the CFG score reads:

$$
\begin{aligned}
\frac{dx^i}{d\tau} =& x^i \left( 1 - \frac{2}{\Gamma(t_f - \tau)} \right) + \frac{2m_i}{\Gamma(t_f - \tau)} e^{-(t_f - \tau)} \\
& + 2\omega m_i \frac{e^{-(t_f - \tau)}}{\Gamma_{t_f - \tau}} \left\{ 1 - \tanh\left( \frac{\vec{x} \cdot \vec{m} e^{-(t_f - \tau)}}{\Gamma_{t_f - \tau}} \right) \right\} + \eta_i(\tau)
\end{aligned}
\tag{21}
$$

where $\tau = 0$ at the beginning of the backward process and $\tau = t_f (\gg 1)$ at the end.

This can be projected on the evolution of the single parameter $q(\tau) = \vec{x} \cdot \vec{m}/\sqrt{d}$. We obtain

$$
\begin{aligned}
\frac{dq}{d\tau} =& q \left( 1 - \frac{2}{\Gamma(t_f - \tau)} \right) + \frac{2\sqrt{d}}{\Gamma(t_f - \tau)} e^{-(t_f - \tau)} \\
& + 2\omega\sqrt{d} \frac{e^{-(t_f - \tau)}}{\Gamma_{t_f - \tau}} \left\{ 1 - \tanh\left( \frac{q\sqrt{d} e^{-(t_f - \tau)}}{\Gamma_{t_f - \tau}} \right) \right\} + \eta(\tau).
\end{aligned}
\tag{22}
$$

Considering the right-hand side as a force due to a moving external potential $-\partial_q V(q, t)$, the effect of CFG is to add an extra term which has two main effects: (1) it adds a positive term to the force and, in consequence, it pushes $q$ faster away from zero, (2) it increases the value of the Hessian at any point in $q$ with respect to its $\omega = 0$ counterpart, thus making the potential more confining.

The initial condition is $q(\tau = 0) \sim \mathcal{N}(0, \sigma^2)$ and

$$
\Gamma(t_f - \tau) = \sigma^2 e^{-2(t_f - \tau)} + 1 - e^{-2(t_f - \tau)}.
\tag{23}
$$

CASE: $\omega = 0$

The solution of the backward equation is:

$$
q(\tau) = q(0) e^{\tau - 2 \int_0^\tau \frac{1}{\Gamma(t_f - \tau'')} d\tau''} + \int_0^\tau \left[ \frac{2\sqrt{d} e^{-(t_f - \tau')}}{\Gamma(t_f - \tau')} + \eta_i(\tau') \right] e^{(\tau - \tau') - 2 \int_{\tau'}^\tau \frac{1}{\Gamma(t_f - \tau'')} d\tau''} d\tau'.
\tag{24}
$$

Its probability distribution must coincide with the one of the solutions of the forward equation, which reads:

$$
q(t) = \sqrt{d} e^{-t} + \sqrt{1 - e^{-2t}} z_i + e^{-t} \sigma \tilde{z}_i,
$$

where $z_i, \tilde{z}_i \sim \mathcal{N}(0,1)$ and $t = t_f - \tau$. Let us now focus on the mean of $q$. When we consider

$$\int_0^\tau \left[ \frac{2\sqrt{d}e^{-(t_f-\tau')}}{\Gamma(t_f-\tau')} \right] e^{(\tau-\tau')-2\int_{\tau'}^\tau \frac{1}{\Gamma(t_f-\tau'')}d\tau''} \, d\tau',$$

using that

$$\frac{d}{d\tau'} \exp\left[ -2\int_{\tau'}^\tau \frac{1}{\Gamma(t_f-\tau'')}d\tau'' \right] = \frac{2}{\Gamma(t_f-\tau')} \exp\left[ -2\int_{\tau'}^\tau \frac{1}{\Gamma(t_f-\tau'')}d\tau'' \right],$$

one finds that the mean of $q$ for the evolution with $\omega = 0$, starting from any value $q(0)$ at any time $t_f$, is

$$q(\tau) = q(0)e^{\tau - 2\int_0^\tau \frac{1}{\Gamma(t_f-\tau')}d\tau'} + \sqrt{d}e^{-(t_f-\tau)}\left( 1 - \exp\left( -2\int_0^\tau \frac{1}{\Gamma(t_f-\tau')}d\tau' \right) \right). \tag{25}$$

Using

$$\int_0^\tau \frac{1}{\Gamma(t_f-\tau')}d\tau' = -\frac{1}{2}\log\frac{e^{-2\tau}+(\sigma^2-1)e^{-2t_f}}{1+(\sigma^2-1)e^{-2t_f}},$$

we find that

$$q(\tau) = q(0)\, e^\tau\, \frac{e^{-2\tau}+(\sigma^2-1)e^{-2t_f}}{1+(\sigma^2-1)e^{-2t_f}} + \sqrt{d}\, e^{-(t_f-\tau)}\, \frac{1-e^{-2\tau}}{1+(\sigma^2-1)e^{-2t_f}}. \tag{26}$$

One can check that, when $q(0)$ is obtained by the equilibrium process with $\omega = 0$, namely $q(0) = \sqrt{d}e^{-t_f}$, then at all times $q(\tau) = \sqrt{d}e^{-(t_f-\tau)}$.

CASE: INTERRUPTED GUIDANCE

Now let us consider a protocol of interrupted guidance. We start the backward process at $t_f \gg 1$ with a CFG coefficient $\omega > 0$. Then at time backward time $\tau_1$ (forward time $t_1 = t_f - \tau_1$) we switch to $\omega = 0$. At time $t_1$ the mean of $q$ obtained from the backward process with $\omega > 0$ is larger than the value $\sqrt{d}e^{-t_1}$ which would be obtained with the $\omega = 0$ dynamics (the reason is that the extra force in (22) is positive). Let us write this mean as

$$q(t_1, \omega) = \sqrt{d}\, e^{-t_1} + \delta q(t_1, \omega).$$

Let us measure the backward time starting from $t = t_1$. We thus write $t = t_1 - \tilde{\tau}$. We can use formula (26) with $t_f \to t_1$, $\tau \to \tilde{\tau}$ and $q(0) \to q(t_1, \omega)$. This gives for the mean value of $q$:

$$\tilde{q}(\tilde{\tau}, \omega) = \sqrt{d}e^{-(t_1-\tilde{\tau})} + \delta q(t_1, \omega)\, \frac{e^{-\tilde{\tau}}+(\sigma^2-1)e^{\tilde{\tau}-2t_1}}{1+(\sigma^2-1)e^{-2t_1}},$$

which, translated in terms of the forward time $t = t_1 - \tilde{\tau}$, gives:

$$q(t) = \sqrt{d}e^{-t} + \delta q(t_1, \omega)\, e^{t-t_1}\, \frac{1+(\sigma^2-1)e^{-2t}}{1+(\sigma^2-1)e^{-2t_1}}. \tag{27}$$

In particular at the end of the backward process, for $\tilde{\tau} = t_1$ we get

$$q(t=0) = \sqrt{d} + \delta q(t_1, \omega)\, e^{-t_1}\, \frac{\sigma^2}{1+(\sigma^2-1)e^{-2t_1}}$$

If we choose $t_1 = t_s = (1/2)\log d$, and assuming that the dynamics at $t > t_1$ has produced an average $q(t_1) = \sqrt{d}e^{-t_1} + \delta q$, we find that

$$q(t=0) = \sqrt{d}\left( 1 + \delta q\, \frac{\sigma^2/d}{1+(\sigma^2-1)/d} \right).$$

This shows that the guidance interrupted at $t_s$ gives a good result only in the limit $\sigma^2/d \ll 1$. Figs. 9 and 10 illustrate the effect of the choice of $t_1$.

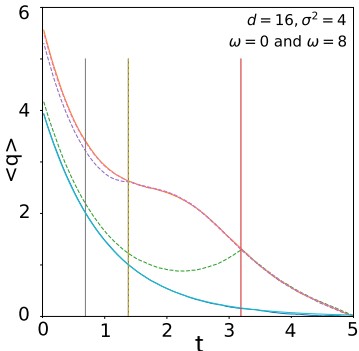

Figure 9: Mean value of $q$ obtained from the backward diffusion in a Gaussian mixture model with $d = 16, \sigma^2 = 4$ (speciation time $t_s = 1.38$). The CFG is run with $\omega = 8$ from $t = 5$ to $t = t_1$, then one switches to the class guidance $\omega = 0$. The top curve is when CFG is kept all the time ($t_1 = 0$). The bottom curve is the case without CFG ($\omega = 0$). Three values of $t_1$ are studied $t_1 = 0.69, 1.38, 3.19$ (vertical lines). The dashed curves give the mean value of $q$ for each of these three cases. They are in perfect agreement with the theoretical prediction (27).

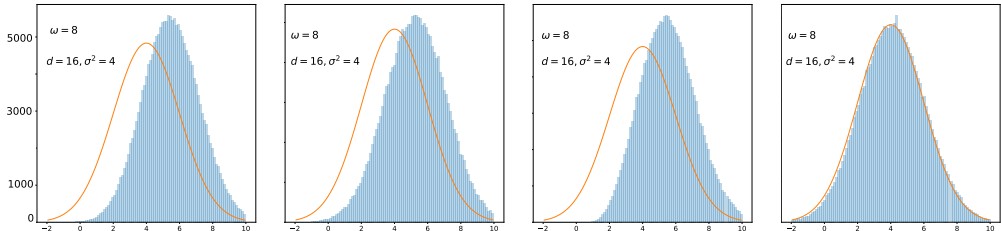

Figure 10: Histograms of $q(t = 0)$ obtained from the backward diffusion in a Gaussian mixture model with $d = 16, \sigma^2 = 4$ (the speciation time is 1.38), run with $200,000$ trajectories. Left: CFG with $\omega = 8$ is applied at all times. The final distribution has a larger mean and a smaller variance than the desired class distribution (full line). Next three figures: The CFG is run with $\omega = 8$ from $t = 5$ to $t = t_1$, then one switches to standard CFG $\omega = 0$. From left to right, $t_1 = 0.69, 1.38, 3.19$. The mean values of $q$ in the four cases are respectively $5.56, 5.51, 5.29, 4.12$ and the standard deviations $1.68, 1.74, 1.87, 1.98$, with targets $\mu = 4, \sigma = 2$. In order to minimize the bias due to CFG one must interrupt it before the speciation takes place in the background diffusion, hence at $t_1 > t_s$.

CFG CONTRIBUTION TO THE MAGNETIZATION IN REGIME I

Using Equation (14), one can derive the equation for the average $\langle q(\tau) \rangle_\omega$:

$$
\frac{d\langle q(\tau) \rangle_\omega}{d\tau} = \langle q(\tau) \rangle_\omega \left( 1 - \frac{2}{\Gamma(t_f - \tau)} \right) + \frac{2\sqrt{d}}{\Gamma(t_f - \tau)} e^{-(t_f - \tau)}
$$

$$
+ 2\omega\sqrt{d} \frac{e^{-(t_f - \tau)}}{\Gamma(t_f - \tau)} \left\langle 1 - \tanh\left( \frac{q\sqrt{d}e^{-(t_f - \tau)}}{\Gamma(t_f - \tau)} \right) \right\rangle_\omega . \tag{28}
$$

The extra $\omega$ term is strictly positive. Therefore, we have:

$$
\langle q(\tau) \rangle_\omega \geq \langle q(\tau) \rangle_{\omega=0}, \quad \forall \tau.
$$

Moreover, using that the right-hand side is less than or equal to:

$$\langle q(\tau)\rangle_\omega \left(1 - \frac{2}{\Gamma(t_f - \tau)}\right) + \frac{2(1+\omega)\sqrt{d}}{\Gamma(t_f - \tau)} e^{-(t_f - \tau)},$$

which corresponds to the backward equation one would obtain if $\|\vec{m}\|^2 = (1 + \omega)d$. We then find:

$$\langle q(\tau)\rangle_{\omega=0} < \langle q(\tau)\rangle_\omega < \sqrt{d}e^{-t}(1+\omega).$$

We conclude that $\langle q(\tau)\rangle_\omega$ gets an extra contribution due to CFG of the order $\sqrt{d}e^{-t}$.

CFG indeed shifts the mean value. The amount of shift is of order $\sqrt{d}e^{-t}$ in Regime I. However, as we shall see next the CFG has almost no effect in Regime II, so we can use the result of the previous section to argue that the total shift due to CFG is the one of CFG in Regime I followed by a switch at $\omega = 0$ in Regime II, i.e., it is of order one.

CFG CONTRIBUTION TO THE SCORE IN REGIME I VS IN REGIME II

Another interesting inequality can be derived for the difference between the CFG and the standard, non-guided score, $S_{\text{CFG}} - S_C$, evaluated on trajectories corresponding to CFG:

$$S_{\text{CFG}} - S_C = \omega \frac{\sqrt{d}e^{-(t_f - \tau)}}{\Gamma(t_f - \tau)} \left(1 - \tanh\left(\frac{q\sqrt{d}e^{-(t_f - \tau)}}{\Gamma(t_f - \tau)}\right)\right). \tag{29}$$

We use the fact that for the same thermal noise, we have $q_\omega(\tau) \geq q_{\omega=0}(\tau)$ because the CFG force is always equal or larger than the $\omega = 0$ one. Therefore for a given (the same) thermal history we have:

$$-\tanh\left(\frac{q_\omega(\tau)\sqrt{d}e^{-(t_f - \tau)}}{\Gamma(t_f - \tau)}\right) \leq -\tanh\left(\frac{q_{\omega=0}(\tau)\sqrt{d}e^{-(t_f - \tau)}}{\Gamma(t_f - \tau)}\right), \tag{30}$$

and we can obtain:

$$S_{\text{CFG}} - S_C \leq \frac{\sqrt{d}e^{-(t_f - \tau)}}{\Gamma(t_f - \tau)} \left(1 - \tanh\left(\frac{q_{\omega=0}(\tau)\sqrt{d}e^{-(t_f - \tau)}}{\Gamma(t_f - \tau)}\right)\right). \tag{31}$$

This inequality tells us, as expected, that the extra CFG contribution to the score is very small at the beginning of the backward process. Its mean increases, and is of the order of one during the backward process in Regime I. However, after the speciation time $q_{\omega=0}(\tau)$ is a Gaussian variable with a mean $\sqrt{d}e^{-(t_f - \tau)}$ much larger than the square root of the variance. Therefore, replacing the fluctuating variable by its mean we obtain

$$S_{\text{CFG}} - S_C \leq \frac{\sqrt{d}e^{-(t_f - \tau)}}{\Gamma(t_f - \tau)} \left(1 - \tanh\left(\frac{de^{-2(t_f - \tau)}}{\Gamma(t_f - \tau)}\right)\right). \tag{32}$$

In Regime II, $t_f - \tau$ is of order one, and using the asymptotic form of the hyperbolic tangent one finds that

$$S_{\text{CFG}} - S_C \leq \frac{\sqrt{d}e^{-(t_f - \tau)}}{\Gamma(t_f - \tau)} \exp\left(-2\frac{de^{-2(t_f - \tau)}}{\Gamma(t_f - \tau)}\right). \tag{33}$$

Therefore in Regime II the extra contribution to the score is exponentially small in $d$ and its effect is completely negligible with respect to the one in Regime I.

ANALYSIS OF THE CFG EFFECT ON THE VARIANCE

Let us derive the equation for $\langle q^2(\tau) \rangle_\omega - \langle q(\tau) \rangle_\omega^2$.

Using Itô calculus, we have (multiplying by $q(\tau)$ in the equation for $\frac{dq(\tau)}{d\tau}$):

$$
\begin{aligned}
\frac{dq^2(\tau)}{d\tau} = {} & 2 + 2q^2(\tau) \left( 1 - \frac{2}{\Gamma(t_f - \tau)} \right) + 2q(\tau) \frac{2\sqrt{d}}{\Gamma(t_f - \tau)} e^{-(t_f - \tau)} \\
& + 2 \frac{2\omega\sqrt{d}}{\Gamma(t_f - \tau)} e^{-(t_f - \tau)} \left( q(\tau) - q(\tau) \tanh \left( \frac{q(\tau)\sqrt{d}e^{-(t_f - \tau)}}{\Gamma(t_f - \tau)} \right) \right) \\
& + 2q(\tau)\eta(\tau).
\end{aligned}
\tag{34}
$$

Taking the average and subtracting $2\langle q(\tau) \rangle_\omega \frac{d\langle q(\tau) \rangle_\omega}{d\tau}$, we find the equation for $\langle q^2(\tau) \rangle_\omega - \langle q(\tau) \rangle_\omega^2$:

$$
\begin{aligned}
\frac{d\langle q^2(\tau) \rangle_\omega - \langle q(\tau) \rangle_\omega^2}{d\tau} = {} & 2 + 2 \left( \langle q^2(\tau) \rangle_\omega - \langle q(\tau) \rangle_\omega^2 \right) \left( 1 - \frac{2}{\Gamma(t_f - \tau)} \right) \\
& + \omega \frac{4\sqrt{d}e^{-(t_f - \tau)}}{\Gamma(t_f - \tau)} \left( \langle q(\tau) \rangle_\omega \left\langle \tanh \left( \frac{q\sqrt{d}e^{-(t_f - \tau)}}{\Gamma(t_f - \tau)} \right) \right\rangle_\omega \right. \\
& \left. - \langle q(\tau) \tanh \left( \frac{q\sqrt{d}e^{-(t_f - \tau)}}{\Gamma(t_f - \tau)} \right) \rangle_\omega \right).
\end{aligned}
\tag{35}
$$

At the beginning of the backward process, one can expand $\tanh(x)$ and observe that the term in the parentheses is proportional to:

$$
- \left( \langle q(\tau)^2 \rangle_\omega - \langle q(\tau) \rangle_\omega^2 \right),
\tag{36}
$$

which is negative. Therefore, we can conclude that the classifier-free-guidance-added term will result in shrinkage of the variance.

As for the mean, the main CFG effect on the variance is produced in Regime I, since the CFG score term is exponentially small in Regime II.

# E  EFFECT OF NONLINEAR CLASSIFIER-FREE GUIDANCE IN FINITE DIMENSIONS

## E.1  NONLINEAR CFG ALONG M

Recall the useful quantities:

$$
\theta \equiv \frac{\vec{x} \cdot \vec{m} \, e^{-t}}{\Gamma_t}, \qquad\qquad \hat{\vec{m}} \equiv \frac{\vec{m}}{\|\vec{m}\|}.
\tag{37}
$$

From Eqs. equation 11-equation 12, the score difference is aligned with $\vec{m}$:

$$
\vec{S}_t(\vec{x}, c) - \vec{S}_t(\vec{x}) = \frac{\|\vec{m}\|e^{-t}}{\Gamma_t} \left( c - \tanh \theta \right) \hat{\vec{m}}.
\tag{38}
$$

Denote $A \equiv \frac{\|\vec{m}\|e^{-t}}{\Gamma_t} \left( c - \tanh \theta \right)$, so that $\vec{S}_t(\vec{x}, c) - \vec{S}_t(\vec{x}) = A \hat{\vec{m}}$. With the power-law guidance $\phi_t(s) = \omega s^\alpha$, the nonlinear CFG score reads

$$
\vec{S}_t^{\text{PL}}(\vec{x}, c) = \vec{S}_t(\vec{x}, c) + \omega \left( \vec{S}_t(\vec{x}, c) - \vec{S}_t(\vec{x}) \right) \left| \vec{S}_t(\vec{x}, c) - \vec{S}_t(\vec{x}) \right|^\alpha,
\tag{39}
$$

Hence the projection onto $\hat{\vec{m}}$ becomes

$$\vec{S}_t^{\mathrm{PL}}(\vec{x}, c) \cdot \hat{\vec{m}} = \vec{S}_t(\vec{x}, c) \cdot \hat{\vec{m}} + \omega \, A \, |A|^\alpha \tag{40}$$

$$= -\frac{\vec{x} \cdot \hat{\vec{m}}}{\Gamma_t} + \frac{\|\vec{m}\| e^{-t}}{\Gamma_t} c + \omega \left( \frac{\|\vec{m}\| e^{-t}}{\Gamma_t} \right)^{1+\alpha} (c - \tanh\theta) \, |c - \tanh\theta|^\alpha. \tag{41}$$

Therefore, the SDE for $q$ becomes with $\|\vec{m}\| = d$ and selecting $c = +1$:

$$\frac{dq}{d\tau} = q \left( 1 - \frac{2}{\Gamma} \right) + \frac{2d}{\Gamma} e^{-(t_f - \tau)} + 2\omega \frac{d^{1+\alpha}}{2^{1+\alpha}} \left( \frac{1}{\Gamma} e^{-(t_f - \tau)} \right)^{1+\alpha} (1 - \tanh(\beta q))^{1+\alpha} + \eta(\tau). \tag{42}$$

Again, setting $\alpha = 0$ recovers the earlier linear-CFG equation (Eq. 22) exactly.

NONLINEAR VS LINEAR EXTRA-DRIFT

We use the same notation as before. Define

$$u(q) = 1 - \tanh(\beta q), \qquad \beta = \frac{1}{\Gamma} d \, e^{-(t_f - \tau)}.$$

(Note that for the "$+\vec{m}$" class one has $u(q) > 0$, hence $u|u|^\alpha = u^{1+\alpha}$.)

The nonlinear extra drift (coming from the power-law CFG, for the $+\vec{m}$ class) is

$$\Delta F(q) = 2\omega \frac{d^{1+\alpha}}{2^{1+\alpha}} \left( \frac{1}{\Gamma} e^{-(t_f - \tau)} \right)^{1+\alpha} u(q)^{1+\alpha},$$

while the linear-CFG ($\alpha = 0$) extra drift is

$$\Delta F_0(q) = 2\omega \frac{d}{2\Gamma} e^{-(t_f - \tau)} u(q).$$

Therefore we look at the following identity:

**Pointwise ratio (nonlinear / linear):**

$$\frac{\Delta F(q)}{\Delta F_0(q)} = \underbrace{\left[ \frac{d \, e^{-(t_f - \tau)}}{\Gamma} u(q) \right]}_{=:B(q)}^\alpha = B(q)^\alpha, \qquad B(q) := \frac{d \, e^{-(t_f - \tau)}}{\Gamma} u(q).$$

Hence whether the nonlinear guidance amplifies (ratio $> 1$) or suppresses (ratio $< 1$) the linear-CFG push at a given $q$ depends only on the sign of $\alpha$ and whether the base $B(q)$ is greater or smaller than 1:

- If $B(q) > 1$:  $\alpha > 0$ amplifies the push,  $\alpha < 0$ suppresses it.
- If $B(q) < 1$:  $\alpha > 0$ suppresses the push,  $\alpha < 0$ amplifies it.

**Hessian / curvature effect.**  Recall that the drift (deterministic part) is written $F(q, \tau)$ and the potential $V$ satisfies $F = -\partial_q V$. The curvature (Hessian) is $\partial_q^2 V = -\partial_q F$. The nonlinear term contributes an extra piece to $\partial_q F$ through the $q$-derivative of $u^{1+\alpha}$. Since $u(q) > 0$ we have

$$\frac{d}{dq} \left[ u(q)^{1+\alpha} \right] = (1 + \alpha) \, u(q)^\alpha \, u'(q).$$

Using $u'(q) = -\beta \, \mathrm{sech}^2(\beta q)$ we obtain

$$\frac{d}{dq} \left[ u(q)^{1+\alpha} \right] = -(1 + \alpha) \, \beta \, \mathrm{sech}^2(\beta q) \, u(q)^\alpha.$$

Therefore the additional contribution to $\partial_q F$ coming from the nonlinear term is (up to the multiplicative prefactors shown above)

$$\Delta(\partial_q F) \; = \; -2\omega\, \frac{d}{1+\alpha} \Big(\frac{1}{\Gamma} e^{-(t_f-\tau)}\Big)^{1+\alpha} (1+\alpha)\,\beta\, \mathrm{sech}^2(\beta q)\, u(q)^{\alpha},$$

which simplifies to

$$\Delta(\partial_q F) \; = \; -2\omega\, d \Big(\frac{1}{\Gamma} e^{-(t_f-\tau)}\Big)^{1+\alpha} \beta\, \mathrm{sech}^2(\beta q)\, u(q)^{\alpha}.$$

Since $\partial_q^2 V = -\partial_q F$, the corresponding increment in curvature is

$$\Delta(\partial_q^2 V) \; = \; +2\omega\,(1+\alpha)\, \frac{d}{1+\alpha} \Big(\frac{1}{\Gamma} e^{-(t_f-\tau)}\Big)^{1+\alpha} \beta\, \mathrm{sech}^2(\beta q)\, u(q)^{\alpha},$$

or, more compactly (omitting positive constants),

$$\Delta(\partial_q^2 V) \; \propto \; d \Big(\frac{1}{\Gamma} e^{-(t_f-\tau)}\Big)^{1+\alpha} \beta\, \mathrm{sech}^2(\beta q)\, u(q)^{\alpha}.$$

**Comparison with the linear ($\alpha = 0$) curvature increment.**  The ratio of nonlinear vs linear curvature increments is essentially

$$\frac{\Delta(\partial_q^2 V)_\alpha}{\Delta(\partial_q^2 V)_0} \; \approx \; B(q)^\alpha,$$

so (for the same reasons as for the drift) whether the nonlinear term *increases* or *decreases* the curvature relative to $\alpha = 0$ depends on the sign of $\alpha$ and on whether $B(q) > 1$ or $B(q) < 1$:

- If $B(q) > 1$:  $\alpha > 0$ amplifies the curvature increase,  $\alpha < 0$ reduces it.
- If $B(q) < 1$:  $\alpha > 0$ reduces the curvature increase,  $\alpha < 0$ amplifies it.

SELECTING $\alpha$ IN THE NONLINEAR CFG SCHEME

Consider the projected SDE for the component $q = \vec{x} \cdot \vec{m}/\sqrt{d}$ under the nonlinear power-law CFG scheme equation 42. We claim that, given a fixed dimension $d$, mean vector $\vec{m}$, variance $\sigma^2$, and a linear CFG scheme with weight $\omega$, one can always construct a simple nonlinear alternative with $\alpha \in \{\alpha', 0\}$, where $\alpha' > -1$. We first develop the argument for $\alpha' > 0$. The reasoning extends directly to $-1 < \alpha' < 0$, with the roles of the cases $B(q) < 1$ and $B(q) > 1$ reversed. The essential conclusion is that, in very low dimensions, effective use of nonlinear CFG requires avoiding an overly strong push, even in the correct direction, since the system is unstable and highly sensitive. For case $\alpha' > 0$ our claims are as follows:

1. The nonlinear term either suppresses the mean push along the conditioning direction (when the score difference is small, $B(q) < 1$) or preserves it (when the score difference is large, $B(q) > 1$).
2. The confinement of the effective potential is either reduced (when the score difference is small, $B(q) < 1$) or kept the same relative to the linear CFG case (when the score difference is large, $B(q) > 1$).

Recall that

$$B(q) \; = \; \frac{\|\vec{m}\| e^{-t}}{\Gamma_t} \big(1 - \tanh \Theta\big), \qquad \Theta \; = \; \frac{\vec{x} \cdot \vec{m}\, e^{-t}}{\Gamma_t},$$

which quantifies the conditional–unconditional score difference projected along $\vec{m}$.

In small dimensions, the generative dynamics of diffusion or flow-matching are highly sensitive to large guidance weights $\omega$ when $\alpha = 0$. In this regime, the system is not "self-correcting" via high-dimensional averaging, so a large push along $\vec{m}$ can cause overshooting or instability.

- When $B(q) > 1$, the score difference $\Delta S = S_t(\vec{x}, c) - S_t(\vec{x})$ is already large, meaning that the conditional signal is strong. In this case, setting $\alpha = 0$ preserves the linear push exactly, ensuring that the system is guided correctly without amplification. Choosing $\alpha > 0$ here would amplify the already strong signal, risking overshoot.

- When $B(q) < 1$, the score difference is small, and the conditional signal is weak. Setting $\alpha = \alpha' > 0$ *dampens* the nonlinear CFG term, preventing small and unreliable differences from producing disproportionate guidance. The nonlinear term is therefore suppressed where it is least reliable, and preserved where the signal is sufficiently strong but not too large.

Thus, by selecting $\alpha = \alpha'$ for $B(q) < 1$ and $\alpha = 0$ otherwise, the dynamics retain a stable, interpretable push along $\vec{m}$, while ensuring that curvature is never increased beyond that of the linear CFG case. In low dimensions, this prevents both over-confinement of the potential and excessive push, providing a natural safeguard against instability.

**Existence of regimes with $B(q) < 1$.** The base factor introduced above was

$$B(q) = \frac{d\, e^{-(t_f - \tau)}}{\Gamma}\, u(q), \qquad u(q) = 1 - \tanh(\beta q), \qquad \Gamma = \Gamma_{t_f - \tau}.$$

Two simple observations guarantee that the nonlinear scheme will *sometimes* suppress the linear-CFG effects (i.e. produce $B(q) < 1$) and therefore provide a benefit relative to standard CFG.

First, in the long-time limit $t \equiv t_f - \tau \to \infty$ we have $e^{-t} \to 0$ and $\Gamma \to 1$, hence

$$\lim_{t \to \infty} B(q) = 0 < 1,$$

for every fixed $q$. Thus there always exist times during the backward process where $B(q) < 1$ and the nonlinear scheme suppresses the linear-CFG push and curvature increase.

Second, at the other endpoint $t = 0$ (the earliest time in the backward process) one has $\Gamma(0) = 1 + (\sigma^2 - 1)e^{-0\cdot 2} = \sigma^2$, so

$$B(q)\Big|_{t=0} = \frac{d}{\sigma^2}\, u(q) \leq \frac{2d}{\sigma^2}.$$

Consequently, in very small dimension, if the problem parameters satisfy the condition

$$\frac{2d}{\sigma^2} < 1 \quad \implies \quad B(q)\big|_{t=0} < 1 \ \text{ for all } q,$$

then $B(q) < 1$ already at $t = 0$ (and hence in a neighbourhood of $t = 0$). Even when $2d/\sigma^2 \geq 1$, the inequality $B(q) < 1$ may still hold for most $q$ if $u(q)$ is typically small (i.e. when $\tanh(\beta q) \approx 1$).

Combining these remarks we conclude:

- There always exist times (in particular sufficiently large $t$) for which $B(q) < 1$, so the nonlinear scheme will suppress linear-CFG effects at those times.

- For small $d$ or sufficiently large noise variance $\sigma^2$ (more precisely when $2d/\sigma^2 < 1$), one also has $B(q) < 1$ near $t = 0$ for all $q$, so the nonlinear scheme suppresses CFG uniformly at early times as well.

- In practice, therefore, the nonlinear power-law CFG will often reduce the unwanted CFG side-effects (mean overshoot and excessive confinement) across substantial portions of the trajectory; this is the regime exploited in our experiments.

This means that the proposed simple nonlinear version of the CFG will always be more beneficial than standard CFG by reducing its unwanted effects. We now proceed to argue about the frequency of the event $B(q) > 1$.

In practice, we found that setting $\alpha > 0$ throughout generation performed well, without explicitly disabling nonlinear guidance when $B(q) > 1$. For simplicity, we adopted this approach across all our experiments. In Fig. 11, we show specifically that for a wide range of data parameters, numerical simulations indicate that the efent $B(q) > 1$ happens very rarely. Nonetheless, we anticipate that adaptively switching may provide further improvements in very low-dimensional settings, with minor gains in higher dimensions.

## F  EXPERIMENTAL DETAILS: GAUSSIAN MIXTURES

In this section, we present experimental details for the numerical simulations involving Gaussian mixtures, describing the procedures and the hyperparameter configurations.

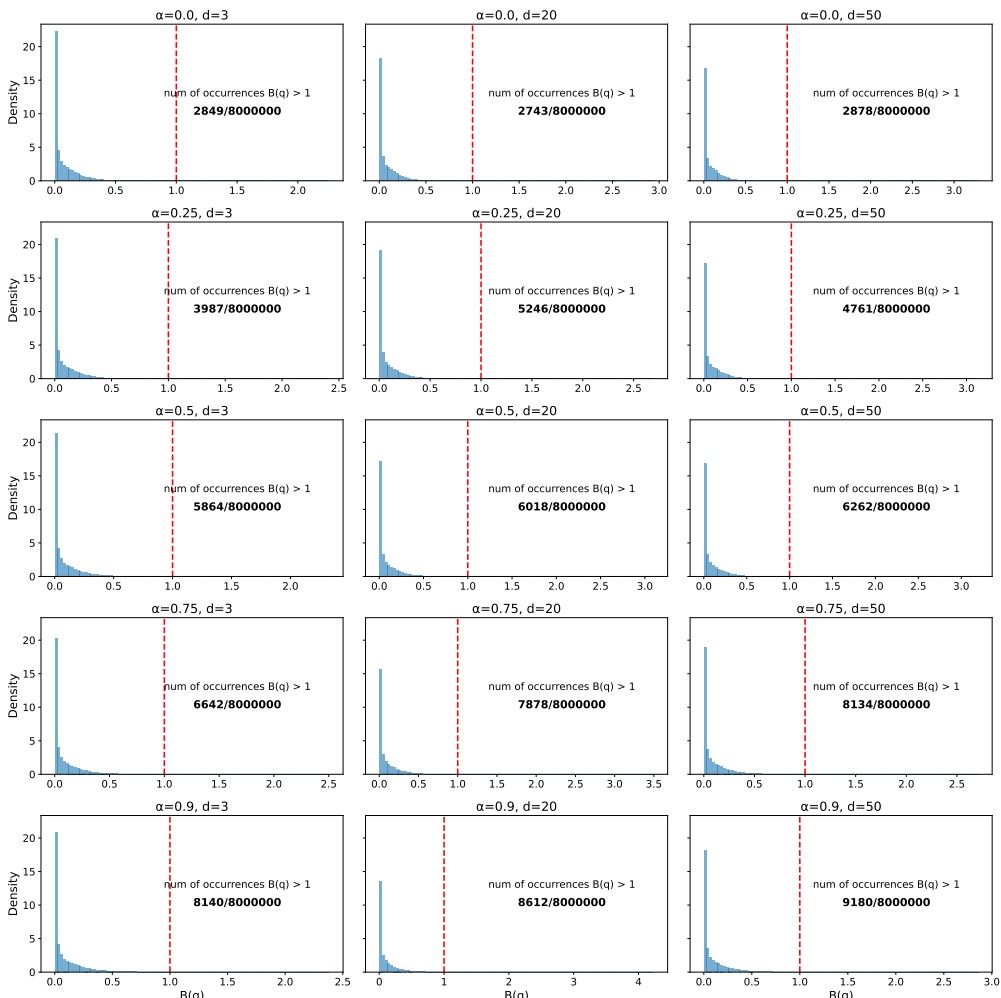

Figure 11: Histograms of the event $B(q) > 1$ for different values of $\alpha \in \{0.0, 0.25, 0.5, 0.75, 0.9\}$ and dimensions $d = 3, 20, 50$, with $\omega$ fixed at 2. The red line marks the cutoff $B(q) = 1$. Across all settings, the event $B(q) > 1$ is seen to be rare, confirming that the condition under which $\alpha$ would be set to zero occurs infrequently in practice.

**Numerical simulations.**  In the case of a mixture of two Gaussian clusters centered on $\pm \vec{m} \in \mathbb{R}^d$ with variance $\sigma^2$, the score function reads as

$$S_{t_{CFG}}(\vec{x}(t), c) = -\frac{x(t)}{\Gamma_t} + \omega \frac{\vec{m} e^{-t}}{\Gamma_t} \left\{ c - \tanh \left( \frac{\vec{x}(t) \cdot \vec{m} e^{-t}}{\Gamma_t} \right) \right\} + \frac{c \vec{m} e^t}{\Gamma_t},$$

where $\Gamma_t = \Delta_t + \sigma^2 e^{-2t}$, with $\Delta_t = 1 - e^{-2t}$. We can then discretize the stochastic differential equation associated to the backward process as

$$\vec{x}(t + 1) = \vec{x}(t) + \eta \left[ \vec{x}(t) + 2 S_{t_{CFG}}(\vec{x}(t), c) \right] + \vec{\eta} \sqrt{2\tau/L},$$

where $\vec{\eta} \sim \mathcal{N}(0, I)$, with $t_f = 8$ the time horizon and $t_f / L = 0.01$. We use $\vec{m} = [1, \dots, 1], \sigma^2 = 1$, and each point is obtained by averaging over 100 initial conditions. The speciation time $t_s$ is calculated as $t_s = -\frac{1}{2} \log d$. Throughout the experiments, we plot the evolution of $q(t) = \frac{\vec{x} \cdot \vec{m}}{|\vec{m}|}$, conditioning the guidance on the positive class with $c = 1$.

## G EXPERIMENTAL DETAILS: REAL-WORLD ANALYSES

### G.1 ASSETS

In Table 2 we list the datasets and models used in our work along with their licensing.

Table 2: Assets used for our work.

| Name | License/Link |
|---|---|
| COCO'14 | https://www.cocodataset.org |
| ImageNet | https://www.image-net.org |
| CC12M | https://github.com/google-research-datasets/conceptual-12m |
| YFCC100M | https://www.multimediacommons.org |
| Florence-2 | https://huggingface.co/microsoft/Florence-2-large/blob/main/LICENSE |
| DiT | https://github.com/facebookresearch/DiT/blob/main/LICENSE.txt |
| EDM2 | https://github.com/NVlabs/edm2/blob/main/LICENSE.txt |
| MMDiT | https://github.com/lucidrains/mmdit/blob/main/LICENSE |
| MDTv2 | https://github.com/sail-sg/MDT/blob/main/LICENSE |

### G.2 PERFORMING THE TIME REPARAMETERIZATION

In the second part of the paper, we evaluate the score of DiT models, in discrete time, as introduced by Peebles & Xie (2023). In this context, the forward process has a linear variance schedule $\{\beta'_t\}_{t'=1}^L$, where $L$ is the time horizon given as a number of steps. Here, the variance evolves linearly from $\beta_1 = 10^{-4}$ to $\beta_{1000} = 2 \times 10^{-2}$. An unguided sample, at timestep $t'$, denoted $\vec{x}(t')$ can be expressed readily from its initial state, $\vec{x}(0) = \vec{a}$, as

$$\vec{x}(t') = \sqrt{\bar{\alpha}}(t')\,\vec{a} + \sqrt{1 - \bar{\alpha}(t')}\vec{\xi}(t')$$

where $\bar{\alpha}(t') = \prod_{s=1}^{t'}(1 - \beta_s)$ and $\vec{\xi}$ is standard Gaussian noise. This equation corresponds to the discretization of the Ornstein-Uhlenbeck Eq. (9) under the following timestep $t'$ reparameterization,

$$t = -\frac{1}{2}\log\left(\bar{\alpha}(t')\right),$$

where time $t$ is as defined in the main manuscript. This gives the map between our theoretical timescale used in Gaussian mixtures, and the one used in real-world settings. We note that, as the neural network predicts the noise, in order to calculate the score, one needs to normalize the output by the standard deviation (depending on the variance schedule). In this case, this corresponds to dividing the neural network output by $\sigma(t') = \sqrt{1 - \bar{\alpha}(t')}$. **In numerical experiments, we divide the CFG-added-term** by $\sigma(t') + 1$ to avoid numerical errors. This is theoretically justified due to the fact that, as discussed in main paper, the score difference $|S_{t'}(\vec{x}, c) - S_{t'}(\vec{x})|$ for large forward times decays exponentially (as $e^{-t'}$) to zero.

For completeness, we present the full comparison of numerical simulations to real-world using the time-reparameterization to plot the timesteps on the same time-scale. Our findings are portrayed in Figure 12. As each framework uses a separate time reparameterization, the x-axis needs to be recalculated accordingly. For the EDM2 framework (Karras et al., 2022), this can be done as follows: given a noise schedule $\sigma(t)$, the reparameterization can be calculated as $t'(t) = (1/2)\log(1 + \sigma^2(t))$, assuming that $s(t) = 1$. For the case $s(t)$, one needs to resort to equation Eq. (2).

### G.2.1 APPLYING THE FORMULA TO VELOCITY FLOW-MATCHING.

For velocity-based flow models (Achilli et al., 2025) it follows that: $\tilde{u}_t(x|y) = u_t(x|y) + \omega[u_t(x|y) - u_t(x)]$ and from their Lemma 1, $u_t(x|y) + b_t\nabla\log p_t(x|y)$. This implies that $|u_t(x|y) - u_t(x)|^\alpha = b_t^\alpha|\nabla\log p_t(x|y) - \nabla\log p_t(x)|^\alpha$. In the case of straight paths, where $\alpha_t = t$ and $\sigma_t = 1 - t$ (as used in our experiments), $b_t^\alpha = (\frac{1-t}{t})^\alpha$. Therefore, Power-Law CFG with $\phi_t(s) = \omega s^\alpha$ does indeed correspond to scaling the velocity difference according to the aforementioned equations. However, since the score difference $\delta S_t = |\nabla\log p_t(x|y) - \nabla\log p_t(x)|$

decays exponentially, both approaches - simply exponentiating $v^\alpha$ and scaling the velocity difference to determine the score- satisfy the nonlinear guidance condition $\lim_{s\to 0} s\phi_t(s) = 0$. Directly exponentiating $v^\alpha$ corresponds to the choice $\phi_t(s) = \omega b_t^\alpha s^\alpha$. In our experiments, to stay consistent with the definition of Power-Law CFG where $\phi_t(s) = \omega s^\alpha$, we scaled the velocity difference according to formulas above. We also examined directly exponentiating $v^\alpha$: although it did improve over standard CFG, it underperformed compared to Power-Law CFG.

## G.3 Hyperparameter configurations

Here, we give exact hyperparameters used for reproducing all our experiments. The real-world experiments are performed using NVIDIA H100 Tensor Core - 80GB HBM3. The EDM2-S model has a model size of 280 Mparams and 102 Gflops, whereas the DiT-XL/2 model has model size of 675 Mparams and 525 Gflops. Parameter $\alpha$ is tuned in $(0.3, 0.95)$ with an increment of $0.05$ and parameter $\omega$ is tuned in $(1., 12.)$ with an increment of $0.05$. To tune $\omega$, we first perform a small grid search of the increment of $1.$ and then do a further extensive search of the best performing $\omega_{prelim}$ in the range $(\omega_{prelim} - 2., \omega_{prelim} + 2.)$ with the $0.05$ increment. We begin with the hyerparameters used in our figures.

In **Figure 1**[4], we plot the generation of images starting from 7 initial seeds for the DiT/XL-2 model trained on ImageNet-1K $(256 \times 256)$ for (1) conditional model without using guidance, (2) standard CFG with $\omega = 4.$, and Power-Law CFG with $\alpha = 0.9, \omega = 8$.

In **Figure 2**, the plots correspond to the histograms of the samples generated using the backward process with dimensions $d \in \{200, 5, 5\}$ and guidance parameter $\omega \in \{0, 0.2, 15\}$, with $\sigma^2 = 1$, averaged over $10,000$ trajectories. The non-linear parameter has $\alpha = 10$.

In **Figure 4**, we use 20-dimensional Gaussian simulation with $\sigma^2 = 1$ and vary the nonlinear parameter as shown in the legend. For the real-world experiments we use the Class-conditional EDM2-S trained on ImageNet-1K 256x256 and an in-house text-to-image model with MMDiT architecture.

In **Figure 5**, we perform sensitivity analysis for EDM2-S trained on ImageNet-1K $(512 \times 512)$, taking $\alpha$ from $0.$ to $0.99$ with 20 evenly spaced values, and $\omega$ from $1.$ to $10.$ with 20 evenly spaced values as well. The right plot involves $\alpha$ values of $0.2, 0.4, 0.6, 0.8, 0.9$ with $\omega$ in the range of $1.$ to $12.5$ with evenly spaced 20 points.

In **Figure 6** we show generations of DiT/XL-2 trained on ImageNet-1K $(256 \times 256)$. The red panel contains generations from weak and strong standard CFG (corresponding to $\omega = 2.$ and $\omega = 5.$ respectively). The green panel corresponds to power-law CFG ($\alpha = 0.9$) with weak and strong guidance (corresponding to $\omega = 2.$ and $\omega = 10.$). The blue panel corresponds to combinations of $\alpha$ and $\omega$ $(0, 2.5)$, $(0.25, 4.)$, $(0.5, 6.)$ and $(0.9, 8.)$.

In **Figure 3**, we plot the evolution of the 1D backward dynamics with means at $\pm 4$ and unit variance. The potential plotted corresponds to equation $V(q, t) = \frac{1}{2}q^2 - 2\log\cosh\left(qe^{-(t-t_s)}\right)$. For the derivation of this potential, see Appendix B.2 in Biroli et al. (2024).

In **Figure 7**, we examine the following functions:

$$V_{\text{class}}(q, t; c) = \frac{1}{2}q^2 - ce^{-(t-t_s)}q + 2$$

$$V_{\text{extra}}(q, t; c) = -ce^{-(t-t_s)}q + \log\left(\cosh\left(qe^{-(t-t_s)}\right)\right) + \log(2),$$

where the plots correspond to $V_{\text{class}}$, $V_{\text{extr}}$ and $(V_{\text{class}} + \omega V_{\text{extr}})$ with $\omega = 3$ respectively. We select $c = 1$, and fix the speciation time to $t_s = .5$. The additive constants are added for clarity only.

In **Figure 8**, the first two plots correspond to the histograms of the samples generated using the backward process with dimensions $d \in \{2, 200\}$ and guidance parameter $\omega \in \{0, 0.2, 15\}$, with $\sigma^2 = 1$, averaged over $10,000$ trajectories. The last two plots correspond to the actual trajectories projected onto the target mean $+\vec{m}$ for values of $\omega \in \{0., 5., 10., 15., 20.\}$.

---

[4]We expect that the images of bees on yellow flowers correspond to a high-likelihood mode of the bee class distribution. A similar trend appears for jellyfish, where stronger guidance produces images with more extensive blue backgrounds (Figure 6, left panel). This likely reflects stronger guidance pushing samples toward higher-likelihood regions—a connection also noted in the Autoguidance paper [(Karras et al., 2024a), Sec. 3], linking optimal score matching to ML estimation.

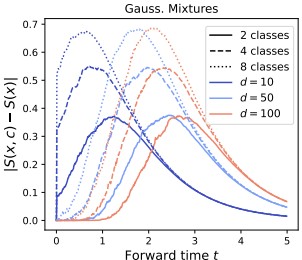 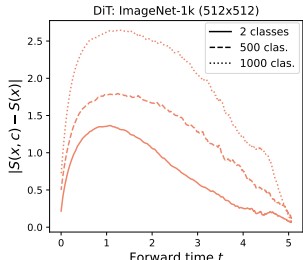 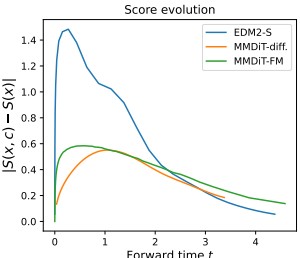

Figure 12: **Evolution of the score differences for numerical simulations and real-world experiments** projected onto the same time-scale for direct comparison. **First:** Numerically simulating mixture of two, four, and eight Gaussians with equidistant means on a sphere ($r = \sqrt{d}$), with varying dimension $d$, with $\omega = 4, \sigma^2 = 1$, averaged over 10,000 trajectories. As $d$ increases, the score difference starts to increase at an earlier backward time $\tau$. Additionally, as the number of classes increases, the magnitude of the score difference grows, as well as the duration of large difference between the scores. **Second:** Three DiT/XL-2 models trained on ImageNet-1K using 2, 500, and 1000 classes (image size $512 \times 512$). We observe a similar pattern: as $d$ increases, the score difference becomes larger at an earlier time. Furthermore, as the number of classes increases, the magnitude of the score difference increases, together with the duration for which the difference remains large. **Third:** evolution of the remaining models used in our experiments (EDM2-S, MMDiT and MDTv2). We observe a similar behavior to theory and the DiT/XL-2 models.

Table 3: Hyperparameter configurations used throughout the experiments.

| $(\alpha, \omega)$ | DiT/XL-2 | EDM2-S | Diff. MMDiT CC12m | Diff. MDTv2 IMN-1K | FM MMDiT COCO | FM MMDiT CC12M |
|---|---|---|---|---|---|---|
| Standard | $(0., 1.5)$ | $(0., 1.4)$ | $(0., 1.55)$ | $(0., 1, 2)$ | $(0., 2.)$ | $(0., 2.1)$ |
| Non-linear | $(0.75, 4.85)$ | $(0.85, 11.4)$ | $(0.6, 7.0)$ | $(0.8, 8.5)$ | $(0.7, 10.15)$ | $(0.6, 8.05)$ |
| Non-lin. + Limited | $(0.8, 4.95)$ | $(0.9, 12.05)$ | $(0.55, 8.25)$ | $(0.85, 8.25)$ | $(0.75, 10.05)$ | $(0.65, 7.85)$ |
| Non-lin. + CADS | $(0.7, 4.75)$ | $(0.80, 11.75)$ | $(0.75, 8.15)$ | $(0.80, 8.40)$ | $(0.75, 10.75)$ | $(0.55, 7.90)$ |

In **Figure 9**, we plot the backward diffusion in a Gaussian mixture model with $d = 16, \sigma^2 = 4, \omega = 8$. The CFG is either run at all times (top curve), stopped at times $t_1$ or not used at all (bottom curve).

In **Figure 10**, we perform linear CFG with $\omega = 8$ from $t = 5$ to $t = t_1$, after which we turn CFG off ($\omega = 0$) at times $t_1 = 0.69, 1.38, 3.19$.

In **Figure 12**, we use DiT/XL-2 model trained on 2, 500 and 1000 classes. For 2 classes, we have selected the same classes as in Biroli et al. (2024), and for the 500 classes we selected the first 500 classes in ImageNet-1K. The x-axis represents the Forward time $t$, where the parameterization is obtained as explained in Section G.2.

In **Figures 33-34**, we perform the same experimenet as in Figure 12 and use $d = 16$ and $\sigma^2 = 4$.

Finally, in the **first column** of **Figure 35**, we plot the estimated Jensen-Shannon Divergence between the target samples corresponding to a randomly selected class and the diffusion particles throughout the backward trajectory. Note that this is performed in latent space. For obtaining the **middle column**, we first take all data samples from one class, embed them into the latent space and calculate the centroid corresponding to this class. Then, we normalize the centroid (making it unit norm) and plot the dot product of the particles throughout the backward diffusion process with the calculated centroid. The **right column** corresponds to the score difference. Across all experiments, we selected $\omega = 4$, sampled using DDPM (Ho et al., 2020) using 250 sampling steps, averaged over 25 samples. All other hyperparameter configurations are set to the default.

**Table 3** displays the hyperparameters used to obtain the results given in Table 1. The evaluation code relied on EvalGIM library by Hall et al. (2024).

## G.4 FURTHER RESULTS

Here, we detail the remaining experiments conducted. We provide the following:

- Diversity and coverage metrics corresponding to experiments in Table 1 (see Table 4)

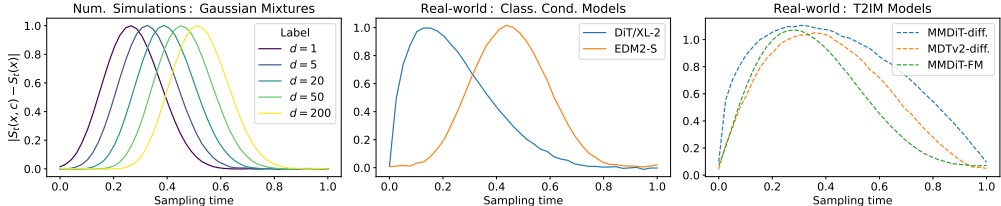

Figure 13: **Evolution of the CFG score difference, from noise ($t = 1$) to data ($t = 0$). Left (stand. CFG):** Numerically simulating mixture of two Gaussians: as $d$ increases, the score difference becomes substantial earlier (this happens during Regime I). **Middle and Right (stand. CFG):** Real-world experiments using advanced models show consistent behavior with theory: monotonically increasing score difference followed by decay after a certain point.

- Ablation studies showing that Power-Law CFG outperforms standard linear CFG when changing the number of steps (see Tables 5-10)

- Sensitivity analysis showing the FID benefit for increasing value of $\alpha$ (see Section G.4.1, Figures 14-16)

- Further qualitative analyses of power-law CFG for either fixed $\omega$ and varying $\alpha$ or varying $\omega$ and varying $\alpha$ (see Sections G.8.1 and G.8.2)

- Further generation examples of DiT/XL-2 and MMDiT diffusion model (see Sections G.8.3 and G.8.4)

**Diversity and coverage metrics.** In Table 4 present additional quantitative evaluations of our method, focusing on diversity and coverage metrics (as described in Hall et al. (2024)), which complement the results shown in Table 1. Our analysis compares power-law CFG to standard CFG and state-of-the-art guidance methods, including combinations with CADS (Sadat et al., 2023) and limited-guidance (Kynkäänniemi et al., 2024), which proved to be the most competitive approaches. As demonstrated in the main manuscript, power-law CFG generally outperforms standard CFG (indicated by arrows in the table). Moreover, when combined with CADS and limited-interval guidance, it yields improved results over existing methods in many cases.

**Ablation studies.** In Tables 5 through 10, we conduct ablation studies on two class-conditional and four text-to-image models, demonstrating that non-linear power-law CFG consistently surpasses standard CFG across varying sampling steps. The results show improved FID performance and enhanced outcomes across multiple metrics when using the non-linear approach compared to standard CFG.

**Sensitivity analysis.** In Section G.4.1, we present additional sensitivity analyses that build on Section 5 and Figure 5, demonstrating that high values of $\alpha$ consistently enhance performance, improving robustness and stability during $\omega$ tuning. As noted in the main manuscript, while power-law CFG introduces an additional hyperparameter, $\alpha$, extensive hyperparameter tuning was unnecessary, with large values like $\alpha = 0.9$ consistently performing well. This is evidenced in Section G.4.1, Figures 14 to 16, which show that higher $\alpha$ values reliably improve FID scores. Class-conditional models (Figure 14) exhibit greater benefits than text-to-image models (Figures 15 and 16), though both show improved performance with Power-Law CFG compared to standard CFG.

**Further qualitative analyses.** In Sections G.8.1 and G.8.2, we provide additional qualitative examples for DiT-XL/2. Specifically, we conduct two studies: one varying the guidance parameter $\omega$ with a fixed $\alpha$, and another varying $\alpha$ with a fixed $\omega$. When $\alpha$ is fixed, increasing $\omega$ can lead to issues such as complete mode collapse (e.g., for the *jellyfish* class), oversaturation (e.g., for the *bee* class), or a significant loss of diversity (e.g., for the *dung beetle* class), which are common artifacts of standard classifier-free guidance. These effects are mitigated when using a non-linear power-law guidance approach. The second study explores the impact of increasing $\alpha$ while keeping $\omega$ constant, demonstrating enhanced diversity as $\alpha$ strength increases. In Figure 13, we plot how the score difference evolves for other CC and T2IM models used in our analyses.

**Further generation examples.** In Sections G.8.3 and G.8.4, we present additional generation examples for class-conditional (DiT/XL-2) and text-to-image (MMDiT) models, demonstrating how power-law CFG enhances image details, thereby improving image quality and fidelity for individual images, and increases diversity when examining a set of images for a specific class.

Table 4: Comparison of EDM2-S on ImageNet-1K 512x512 data, Diffusion trained text-to-image MMDiT on CC12m data, and Flow-matching trained text-to-image MMDiT on COCO data. **Bolded** are the best results and underlined are the second best.

| Model | EDM2-S (CC, IM-1K 512) | | DiT/XL-2 (CC, IM-1K 256) | | Diff. MMDiT (T2IM, CC12m) | | FM MMDiT (T2IM, COCO) | |
|---|---|---|---|---|---|---|---|---|
| | Density | Coverage | Density | Coverage | Density | Coverage | Density | Coverage |
| Standard (Ho & Salimans, 2022) | 0.850 | 0.764 | 0.951 | 0.801 | 1.091 | 0.840 | 0.902 | 0.772 |
| Scheduler (Wang et al., 2024) | **0.867** | 0.780 | 1.117 | 0.790 | 1.266 | 0.860 | 0.908 | 0.795 |
| Limited (Kynkäänniemi et al., 2024) | 0.845 | 0.777 | **1.130** | 0.840 | 1.258 | 0.857 | 0.915 | **0.808** |
| Cosine (Gao et al., 2023) | 0.850 | 0.769 | 1.102 | 0.822 | 1.106 | 0.840 | 0.920 | 0.802 |
| CADS (Sadat et al., 2023) | 0.854 | 0.765 | 0.999 | 0.853 | 1.222 | 0.860 | 0.923 | 0.779 |
| APG (Sadat et al., 2024) | 0.845 | 0.760 | 1.033 | 0.867 | 1.095 | 0.858 | 0.915 | 0.797 |
| REG (Xia et al., 2024) | 0.850 | 0.771 | 1.112 | 0.833 | 1.091 | 0.855 | 0.903 | 0.783 |
| CFG++ (Chung et al., 2024) | N/A | N/A | N/A | N/A | 1.265 | 0.859 | 0.919 | 0.784 |
| Power-law CFG (Ours) | 0.845 (↓) | 0.760 (↑) | 0.986 (↑) | 0.844 (↑) | 1.128 (↑) | 0.850 (↑) | 0.918 (↑) | 0.778 (↑) |
| Power-law CFG + Limited (Ours) | 0.850 (↑) | 0.778 (↑) | 1.115 (↓) | 0.835 (↓) | **1.286** (↑) | 0.860 (↑) | 0.920 (↑) | 0.795 (↓) |
| Power-law CFG + CADS (Ours) | 0.862 (↑) | **0.782** (↑) | 1.071 (↑) | **0.876** (↑) | 1.279 (↑) | **0.862** (↑) | **0.924** (↑) | 0.804 (↑) |

Table 5: **Ablation study:** Changing the number of sampling steps for Class-conditional: DiT ImageNet-1K 256x256

| Version | Num. steps | $\alpha$ | $\omega$ | **FID** (↓) | **IS** (↑) | **Precision** (↑) | **Recall** (↑) | **sFID** (↓) |
|---|---|---|---|---|---|---|---|---|
| Stand. CFG | 50 | 0 | 1.5 | 3.33 | 259.88 | 0.8163 | 0.5474 | 7.406 |
| | 100 | 0 | 1.4 | 2.64 | 233.72 | 0.8027 | 0.5831 | 5.720 |
| | 150 | 0 | 1.3 | 2.38 | 233.52 | 0.8032 | 0.5936 | 5.462 |
| | 200 | 0 | 1.35 | 2.29 | 234.92 | 0.8031 | 0.5950 | 5.331 |
| | 250 | 0 | 1.5 | 2.27 | 278.30 | 0.8291 | 0.5840 | 4.601 |
| Non-lin. CFG | 50 | 0.6 | 4.35 | 3.03 | 284.55 | 0.8215 | 0.5757 | 7.110 |
| | 100 | 0.6 | 3.4 | 2.32 | 274.36 | 0.8199 | 0.6012 | 5.432 |
| | 150 | 0.6 | 3.4 | 2.19 | 274.39 | 0.8202 | 0.6071 | 5.512 |
| | 200 | 0.75 | 4.8 | 2.17 | 276.98 | 0.8204 | 0.5956 | 5.567 |
| | 250 | 0.75 | 4.85 | 2.05 | 279.90 | 0.8310 | 0.5950 | 4.670 |

Table 6: **Ablation study:** Changing the number of sampling steps for Class-conditional: EDM2-S ImageNet-1K 512x512

| Version | Num. Steps | $\alpha$ | $\omega$ | **FID** (↓) | $\alpha$ | $\omega$ | **FID**$_{\text{DINO}}$ (↓) |
|---|---|---|---|---|---|---|---|
| Stand. CFG | 8 | 0 | 1.95 | 4.78 | 0 | 2.3 | 103.33 |
| | 16 | 0 | 1.50 | 2.52 | 0 | 2.3 | 57.47 |
| | 32 | 0 | 1.40 | 2.29 | 0 | 2.3 | 54.78 |
| | 64 | 0 | 1.50 | 2.25 | 0 | 2.15 | 54.39 |
| Non-lin. CFG | 8 | 0.05 | 2.30 | 4.74 | -0.25 | 1.5 | 100.81 |
| | 16 | 0.25 | 2.30 | 2.32 | -0.05 | 2.15 | 56.92 |
| | 32 | 0.85 | 11.40 | 1.93 | 0.35 | 2.5 | 52.77 |
| | 64 | 0.85 | 11.30 | 1.89 | 0.35 | 2.1 | 52.56 |

Table 7: **Ablation study:** Changing the number of sampling steps for Diffusion text-to-image: MMDiT CC12m

| Version | Num. Steps | $\alpha$ | $\omega$ | FID ($\downarrow$) | Clip score ($\uparrow$) | Coverage ($\uparrow$) | Density ($\uparrow$) | Precision ($\uparrow$) | Recall ($\uparrow$) |
|---|---|---|---|---|---|---|---|---|---|
| Stand. CFG | 20 | 0 | 1.75 | 8.98 | 22.581 | 0.8392 | 1.104 | 0.6623 | 0.5545 |
| | 35 | 0 | 1.75 | 8.79 | 22.532 | 0.8450 | 1.124 | 0.6717 | 0.5590 |
| | 50 | 0 | 1.55 | 8.58 | 22.111 | 0.8401 | 1.109 | 0.6612 | 0.5692 |
| | 100 | 0 | 1.75 | 8.38 | 22.298 | 0.8462 | 1.117 | 0.6765 | 0.5698 |
| Non-lin. CFG | 20 | 0.25 | 3.05 | 8.94 | 22.773 | 0.8424 | 1.114 | 0.6619 | 0.5495 |
| | 35 | 0.65 | 7.5 | 8.40 | 22.590 | 0.8491 | 1.126 | 0.6638 | 0.5582 |
| | 50 | 0.60 | 7.0 | 8.11 | 22.415 | 0.8503 | 1.128 | 0.6703 | 0.5532 |
| | 100 | 0.75 | 9.5 | 8.02 | 22.563 | 0.8472 | 1.115 | 0.6747 | 0.5723 |

Table 8: **Ablation study:** Changing the number of sampling steps for Diffusion text-to-image: MDTv2 ImageNet-1K 512x512

| Version | Num. Steps | $\alpha$ | $\omega$ | FID ($\downarrow$) | Clip score ($\uparrow$) | Coverage ($\uparrow$) | Density ($\uparrow$) | Precision ($\uparrow$) | Recall ($\uparrow$) |
|---|---|---|---|---|---|---|---|---|---|
| Stand. CFG | 20 | 0 | 1.55 | 5.30 | 23.949 | 0.8218 | 1.167 | 0.7475 | 0.5133 |
| | 30 | 0 | 1.55 | 4.09 | 23.998 | 0.8292 | 1.233 | 0.7492 | 0.5264 |
| | 40 | 0 | 1.6 | 3.85 | 24.011 | 0.8311 | 1.178 | 0.7602 | 0.5294 |
| | 50 | 0 | 1.2 | 3.68 | 24.306 | 0.8318 | 1.150 | 0.7510 | 0.5989 |
| Non-lin. CFG | 20 | 0.6 | 6.0 | 4.88 | 24.154 | 0.8251 | 1.236 | 0.7503 | 0.4916 |
| | 30 | 0.6 | 6.0 | 4.03 | 24.033 | 0.8344 | 1.205 | 0.7583 | 0.5332 |
| | 40 | 0.7 | 7.0 | 3.73 | 23.367 | 0.8353 | 1.181 | 0.7557 | 0.5546 |
| | 50 | 0.8 | 8.5 | 3.57 | 25.339 | 0.8361 | 1.170 | 0.7513 | 0.5609 |

Table 9: **Ablation study:** Changing the number of sampling steps for Flow-Matching text-to-image: MMDiT on COCO

| Version | Num. Steps | $\alpha$ | $\omega$ | FID ($\downarrow$) | Clip score ($\uparrow$) | Coverage ($\uparrow$) | Density ($\uparrow$) | Precision ($\uparrow$) | Recall ($\uparrow$) |
|---|---|---|---|---|---|---|---|---|---|
| Stand. CFG | 20 | 0 | 2.85 | 6.84 | 26.373 | 0.7529 | 0.8820 | 0.6121 | 0.5604 |
| | 30 | 0 | 1.95 | 5.84 | 25.948 | 0.7581 | 0.8668 | 0.6051 | 0.5879 |
| | 40 | 0 | 2.05 | 5.62 | 25.817 | 0.7651 | 0.8798 | 0.6091 | 0.5978 |
| | 50 | 0 | 2.00 | 5.20 | 25.714 | 0.7726 | 0.9026 | 0.6299 | 0.5940 |
| Non-lin. CFG | 20 | 0.5 | 9.75 | 6.47 | 25.981 | 0.7241 | 0.7762 | 0.5719 | 0.5851 |
| | 30 | 0.5 | 9.45 | 5.62 | 26.003 | 0.7577 | 0.8457 | 0.6105 | 0.5874 |
| | 40 | 0.6 | 9.05 | 5.45 | 25.113 | 0.7633 | 0.8549 | 0.6149 | 0.6030 |
| | 50 | 0.7 | 10.15 | 4.81 | 25.848 | 0.7782 | 0.9183 | 0.6208 | 0.6191 |

Table 10: **Ablation study:** Changing the number of sampling steps for Flow-Matching text-to-image: MMDiT on CC12m

| Version | Num. Steps | $\alpha$ | $\omega$ | FID ($\downarrow$) | Clip score ($\uparrow$) | Coverage ($\uparrow$) | Density ($\uparrow$) | Precision ($\uparrow$) | Recall ($\uparrow$) |
|---|---|---|---|---|---|---|---|---|---|
| Stand. CFG | 20 | 0 | 2.75 | 10.75 | 25.224 | 0.8289 | 1.069 | 0.6396 | 0.5803 |
| | 30 | 0 | 1.95 | 9.85 | 24.935 | 0.8318 | 1.068 | 0.6946 | 0.6000 |
| | 40 | 0 | 2.0 | 9.50 | 25.018 | 0.8461 | 1.103 | 0.7064 | 0.5907 |
| | 50 | 0 | 2.1 | 9.46 | 25.133 | 0.8520 | 1.145 | 0.7159 | 0.5946 |
| Non-lin. CFG | 20 | 0.2 | 3.25 | 10.68 | 25.585 | 0.8301 | 1.075 | 0.7101 | 0.5815 |
| | 30 | 0.5 | 10.0 | 9.81 | 25.002 | 0.8338 | 1.085 | 0.6968 | 0.5909 |
| | 40 | 0.6 | 9.35 | 9.17 | 24.794 | 0.8352 | 1.087 | 0.6909 | 0.6030 |
| | 50 | 0.6 | 8.05 | 9.00 | 24.723 | 0.8397 | 1.087 | 0.6911 | 0.6023 |

## G.5 Low-dimensional experiments

In the following experiments, we have trained a U-ViT-like model (Bao et al., 2022) on lower-dimensional datasets to confirm that nonlinear CFG provides further improvemenets to standard CFG in lower-dimensional settings. For CIFAR10 the architecutre closely mimics that of U-ViT-S/2 and for ImageNet64x64 and Imagenet32x32 U-ViT-M/4. We have obtained ImageNet32x32 by using bilinear downsampling on ImageNet64x64.

Regarding sampling, for CIFAR10 we used *'euler_maruyama_sde'* sampler with a maximum of 1000 steps (we did not find further improvement by further increasing step size). For ImageNet32x32 and ImageNet64x64 we used *'dpm_solver'* with a maximum number of sampling steps equal to 100 (we did not find improvements by further increasing it from 100).

### G.4.1 SENSITIVITY ANALYSIS

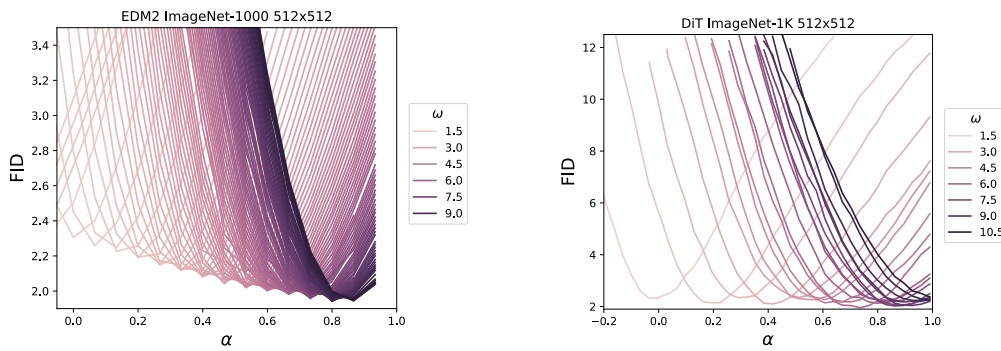

Figure 14: **Class-conditional diffusion**: image quality benefits from non-linear scheme, yielding lower FID for larger values of $\alpha$.

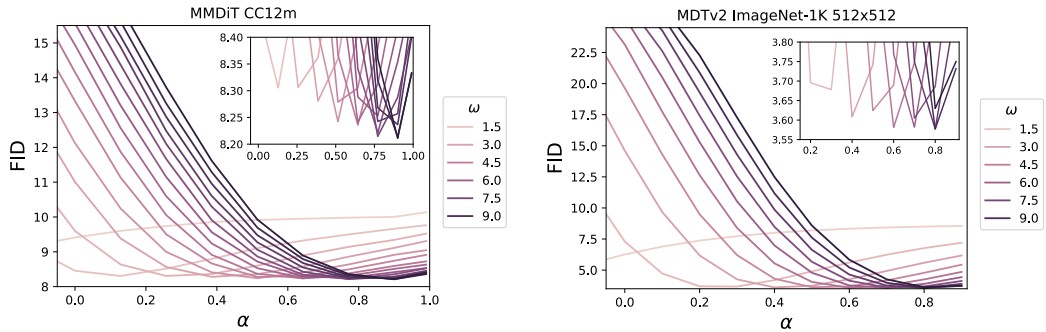

Figure 15: **Text-to-image diffusion models**: image quality benefits from non-linear scheme, yielding lower FID for larger values of $\alpha$.

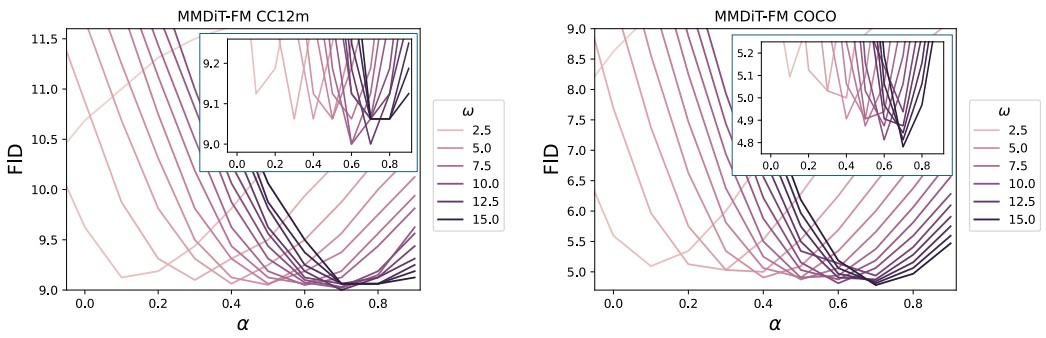

Figure 16: **Text-to-image flow matching**: image quality benefits from non-linear scheme, yielding lower FID for larger values of $\alpha$.

The results from the first experiment, showing the lower-dimensional improvements in FID can be seen in Tables 10-12, whereas the experiment of choosing very small guidance scale can be seen in Table 13. The findings are as follows: as the dimension decreases and the unwanted effects of CFG get stronger, the benefit from applying non-linear CFG further increases, as can be seen from a

Table 11: **Comparison of CFG variants on ImageNet-1K (pixel space)** for Matryoshka Diffusion and Pixelflow Flow Matching. Second row represents the performance with fixed $\alpha = 0.9$, last row with tuned $\alpha$.

| Version | Matryoshka Diffusion (Gu et al., 2023) | | | Pixelflow Flow Matching (Chen et al., 2025b) | | |
|---|---|---|---|---|---|---|
| | FID ↓ | Prec ↑ | Rec ↑ | FID ↓ | Prec ↑ | Rec ↑ |
| Standard CFG | 3.51 | 0.81 | 0.54 | 2.43 | 0.83 | 0.56 |
| Nonlinear CFG ($\alpha$=0.9) | 3.39 | 0.82 | 0.55 | 2.29 | 0.83 | 0.58 |
| Nonlinear CFG ($\alpha$=0.65/0.45) | 3.17 | 0.83 | 0.56 | 2.12 | 0.85 | 0.59 |

more pronounced improvemenet in FID. Secondly, for very small $\omega$, the performance of linear and non-linear CFG equals that of the conditional, unguided model.

### G.6 NUMERICAL SIMULATIONS: MIXTURES OF GAUSSIANS WITH DIFFERENT MEANS, VARIANCES AND MIXTURE PROBABILITIES

In this section, we present experimental results comparing standard Classifier-Free Guidance (CFG) and Nonlinear CFG on Gaussian mixtures with 3 and 4 components.

**Experiment 1: 3-Component Gaussian Mixture ($d = 2$)** We first investigate a two-dimensional ($d = 2$) setting using a Gaussian mixture model composed of 3 components with assignment probabilities of $30\%$, $40\%$, and $30\%$.

The results align with our two-mixture Gaussian analysis:

- Standard CFG, especially with strong guidance parameters, can induce **unwanted effects** (as shown in Figure 17).
- By selecting Nonlinear Guidance with $\alpha \neq 0$, these negative effects can be **dampened** (as shown in Figure 18).

**Experiment 2: 4-Component Gaussian Mixture ($d = 2$ vs. $d = 1000$)** The second experiment examines the improvements offered by Nonlinear Guidance over standard CFG, as well as the mitigating effect of high dimensionality on CFG's unwanted behaviors. We use a 4-component Gaussian mixture for a small-dimensional experiment ($d = 2$) and a large-dimensional experiment ($d = 1000$).

The findings are consistent with the two-Gaussian component analysis (and can be seen in Figures 19 and 20:

1. Nonlinear CFG effectively **dampens** the effects of **mean overshoot** and **variance shrinkage** induced by standard CFG.
2. As the dimensionality ($d$) increases, the **mean overshoot** and **variance shrinkage** effects associated with standard CFG **diminish**.

### G.7 RESULTS FOR PIXEL-SPACE VS LATENT SPACE

Here, we provide the table for the experiment conducted in Section 5.3. We show that the benefit of power-law CFG can also be observed when applying non-linear guidance in pixel space, and not just latent space.

### G.8 ROBUSTNESS ANALYSIS: RESOLUTION, SCHEDULING, AND ALTERNATIVE METRICS

In this section, we discuss the robustness of the Power-Law formulation across different vector norms (resolution), noise schedules, and distance metrics.

**Resolution and Vector Norms.** In high-dimensional spaces, the vector norm of the score difference is sensitive to the dimensionality of the data (e.g., a $512^2$ pixel-space image vs. a $32^2$ latent). While the intrinsic manifold dimensionality may remain similar, the Euclidean norm scales with resolution. Theoretically, the overall shape of the score difference curve remains consistent (hump-shaped behavior tending to zero at $t \to 0$ and $t \to 1$), but the absolute scale changes. In the context of Power-Law CFG, this resolution-based scaling is effectively managed by the renormalization provided by the hyperparameters; Specifically, choosing the optimal guidance scale $\omega$ renormalizes

Table 12: Comparison of Standard CFG, Cosine-based variants, and Power-Law (Nonlinear) CFG on EDM-2/S ImageNet-1K ($512 \times 512$).

| Version | FID $\downarrow$ | FID_DINO $\downarrow$ |
|---|---|---|
| Standard CFG | 2.29 | 54.76 |
| Cosine CFG | 2.26 | 54.28 |
| Cosine CFG w. $\alpha$ | 2.21 | 54.09 |
| **Nonlinear CFG (Ours)** | **1.93** | **52.77** |

the score difference for the given resolution regime. We validated this by evaluating Power-Law CFG on pixel-space methods (Matryoshka-diffusion (Gu et al., 2023) and PixelFlow (Chen et al., 2025b)), confirming that the method generalizes robustly to high-resolution pixel space without requiring explicit resolution-based formulation changes.

**Noise Schedule Scaling.** The magnitude of the score difference is naturally influenced by the diffusion time $t$ (and consequently $\sigma_t$). While one could theoretically define a schedule-specific nonlinear guidance of the form $\phi_t(s, \sigma_t) = s^\alpha f(\sigma_t)$ to decouple these effects, we find that the standard Power-Law formulation is sufficient to improve upon standard CFG. The exponential decay of the score difference at boundary times theoretically mitigates the need for complex, schedule-dependent scaling functions $f(\sigma_t)$ in high dimensions, but we suspect further empirical improvements might be observed by tuning for optimal $f(\sigma_t$.

**Euclidean Difference vs. Cosine Distance.** Finally, we investigated whether the performance gain of Power-Law CFG stems purely from directional alignment or if the magnitude of the score difference is essential. We compared our method against "Cosine CFG," where the guidance is scaled based on the cosine distance (purely directional), and "Cosine CFG w. $\alpha$," where the cosine distance is raised to a power $\alpha$.We evaluated these variants using EDM-2/S on ImageNet ($512 \times 512$ latent space). As shown in Table 12, while using Cosine distance improves over standard CFG, it does not match the performance of Power-Law CFG. This suggests that the "built-in" scaling provided by the Euclidean norm—which accounts for both the angular difference and the relative magnitude of the conditional and unconditional scores—is a critical component of effective nonlinear guidance.

| Version | Num. steps | $\alpha$ | $\omega$ | FID ($\downarrow$) | Precision ($\uparrow$) | Recall ($\uparrow$) |
|---|---|---|---|---|---|---|
| Lin. CFG | 100 | 0 | 1.55 | 3.50 | 0.6569 | 0.5932 |
| | 250 | 0 | 1.40 | 2.77 | 0.6938 | 0.6115 |
| | 500 | 0 | 1.35 | 2.03 | 0.6894 | 0.6262 |
| | 1000 | 0 | 1.45 | 1.80 | 0.7071 | 0.6186 |
| Nonlin. CFG | 100 | 0.9 | 3.90 | 2.94 | 0.6653 | 0.6111 |
| | 250 | 0.9 | 4.35 | 2.16 | 0.6688 | 0.6298 |
| | 500 | 0.9 | 4.35 | 1.62 | 0.6688 | 0.6298 |
| | 1000 | 0.9 | 4.75 | 1.48 | 0.7228 | 0.6506 |

Table 13: CIFAR-10 Results

| Version | Num. steps | $\alpha$ | $\omega$ | FID ($\downarrow$) | Precision ($\uparrow$) | Recall ($\uparrow$) |
|---|---|---|---|---|---|---|
| Lin. CFG | 25 | 0 | 1.55 | 5.81 | 0.3842 | 0.4632 |
| | 50 | 0 | 1.40 | 4.01 | 0.4161 | 0.4750 |
| | 75 | 0 | 1.60 | 3.81 | 0.4120 | 0.4724 |
| | 100 | 0 | 1.65 | 3.35 | 0.4306 | 0.4967 |
| Nonlin. CFG | 25 | 0.9 | 3.40 | 4.97 | 0.4022 | 0.4564 |
| | 50 | 0.9 | 3.35 | 3.90 | 0.4241 | 0.4672 |
| | 75 | 0.9 | 3.40 | 3.33 | 0.4431 | 0.5050 |
| | 100 | 0.9 | 3.85 | 2.88 | 0.4539 | 0.5263 |

Table 14: IMNET-32 Results

| Version | Num. steps | $\alpha$ | $\omega$ | FID ($\downarrow$) | Precision ($\uparrow$) | Recall ($\uparrow$) |
|---|---|---|---|---|---|---|
| Lin. CFG | 25 | 0 | 1.60 | 5.32 | 0.3996 | 0.4845 |
| | 50 | 0 | 1.80 | 4.48 | 0.4484 | 0.4962 |
| | 75 | 0 | 1.70 | 4.37 | 0.4576 | 0.5137 |
| | 100 | 0 | 1.75 | 4.34 | 0.4511 | 0.5267 |
| Nonlin. CFG | 25 | 0.9 | 3.80 | 4.76 | 0.4212 | 0.4718 |
| | 50 | 0.9 | 4.00 | 4.02 | 0.4656 | 0.5183 |
| | 75 | 0.9 | 4.20 | 3.75 | 0.4734 | 0.5512 |
| | 100 | 0.9 | 4.15 | 3.71 | 0.4883 | 0.5563 |

Table 15: IMNET-64 Results

| Method | CIFAR-10 FID ($\downarrow$) | IMNET-32 FID ($\downarrow$) | IMNET-64 FID ($\downarrow$) |
|---|---|---|---|
| Uncond. | 3.142 | 3.891 | 5.930 |
| Cond. | 2.401 | 3.167 | 4.277 |
| Lin. CFG w. $\omega + \epsilon$ | 2.400 | 3.176 | 4.299 |
| Nonlin. CFG w. $\omega + \epsilon$ | 2.399 | 3.194 | 4.285 |

Table 16: FID Comparison Across Datasets

Figure 17: Numerical simulation of Standard CFG in a mixture of 3 Gaussians

Figure 18: Numerical simulation of Nonlinear CFG in a mixture of 3 Gaussians

Figure 19: Numerical simulation of Standard CFG in a mixture of 4 Gaussians

Figure 20: Numerical simulation of Nonlinear CFG in a mixture of 3 Gaussians

### G.8.1 Qualitative analysis: varying $\omega$, fixed $\alpha$.

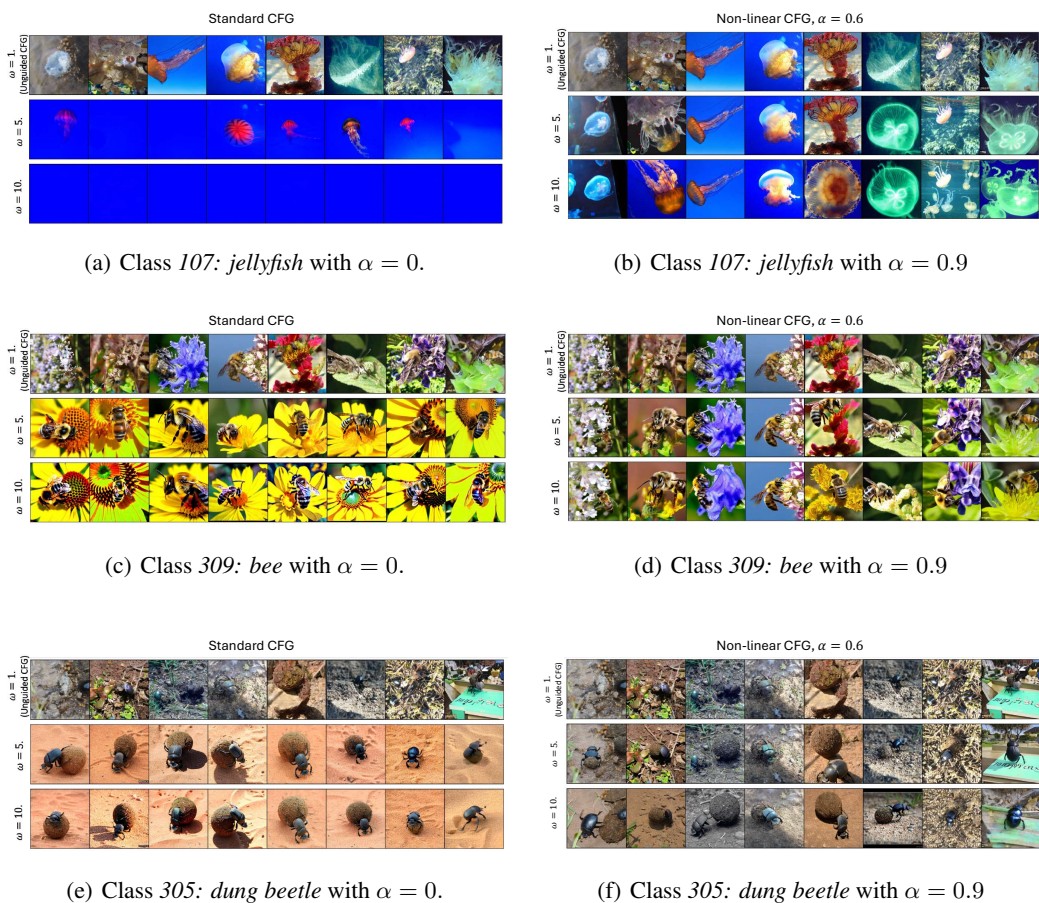

(a) Class *107: jellyfish* with $\alpha = 0$.

(b) Class *107: jellyfish* with $\alpha = 0.9$

(c) Class *309: bee* with $\alpha = 0$.

(d) Class *309: bee* with $\alpha = 0.9$

(e) Class *305: dung beetle* with $\alpha = 0$.

(f) Class *305: dung beetle* with $\alpha = 0.9$

Figure 21: Generated images for different classes for varying values of $\omega$ and $\alpha$. Each panel shows the effect of changing $\alpha$ from 0 to 0.9, demonstrating the impact on diversity and image quality.

### G.8.2 QUALITATIVE ANALYSIS: FIXED $\omega$, VARYING $\alpha$.

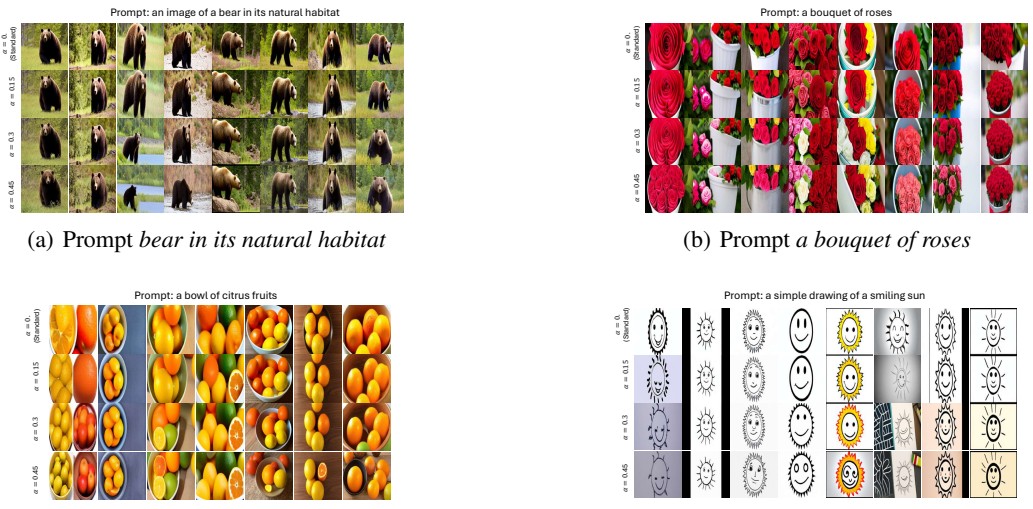

(a) Prompt *bear in its natural habitat*

(b) Prompt *a bouquet of roses*

(c) Prompt *a bowl of citrus fruit*

(d) Prompt *a simple drawing of a smiling sun*

Figure 22: T2IM generated images for different prompts with $\omega = 4$. and varying value of $\alpha$.

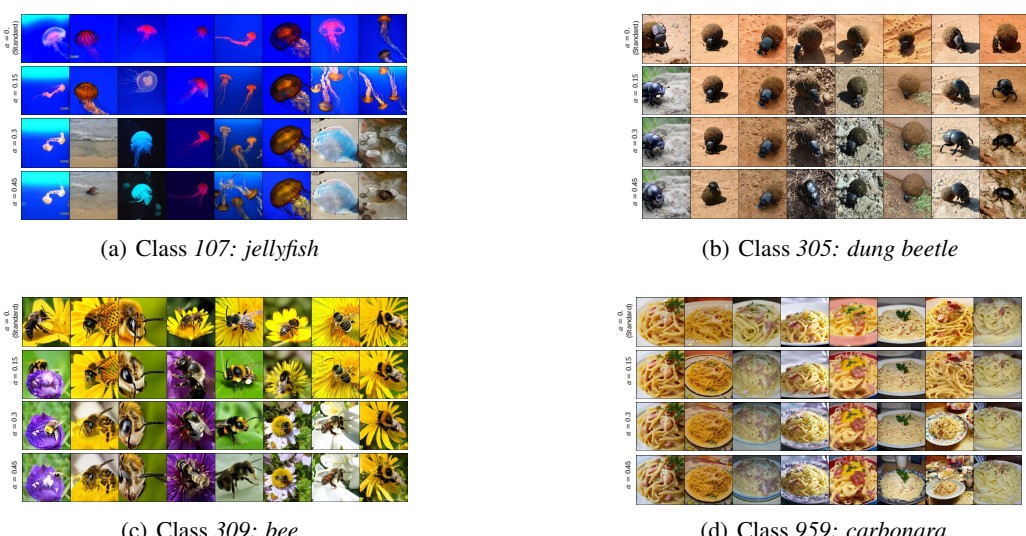

(a) Class *107: jellyfish*

(b) Class *305: dung beetle*

(c) Class *309: bee*

(d) Class *959: carbonara*

Figure 23: CC generated images for different classes with $\omega = 4$. and varying value of $\alpha$.

### G.8.3 GENERATED IMAGES BY DiT/XL-2 (256x256)

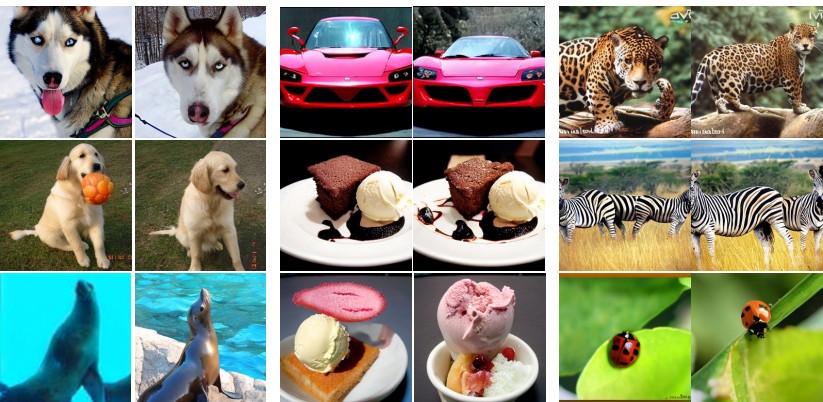

Figure 24: Additional examples generated by DiT/XL-2 using Standard CFG ($\omega = 4$) and Power-Law CFG ($\omega = 8., \alpha = 0.7$). Image pairs start from the same noise (same seed). The resulting pairs represent Standard CFG on the left and Power-Law CFG on the right.

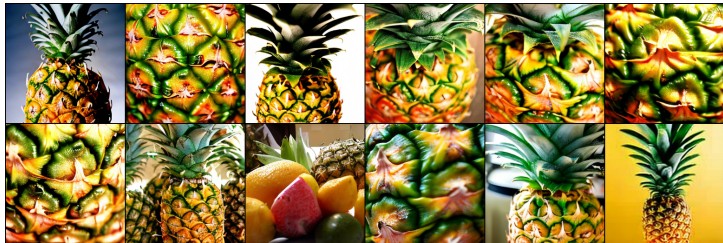

Figure 25: Gen. images conditioned on the class *pineapple* with Standard CFG ($\omega = 4$).

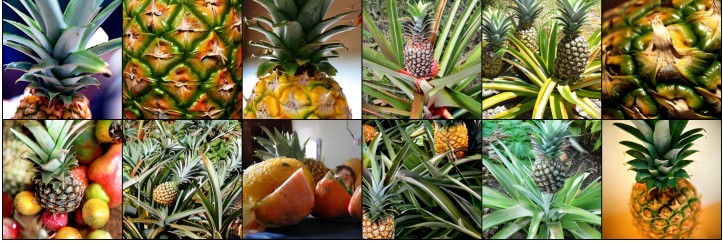

Figure 26: Gen. images conditioned on class *pineapple* with Power-Law CFG ($\omega = 8, \alpha = 0.7$).

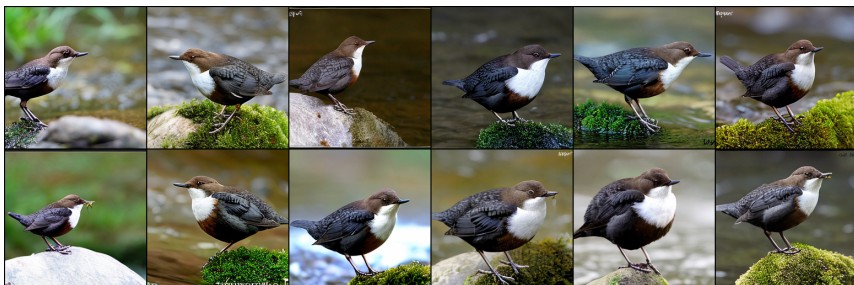

Figure 27: Gen. images conditioned on the class *water ouzel, dipper* using Standard CFG with $\omega = 4$.

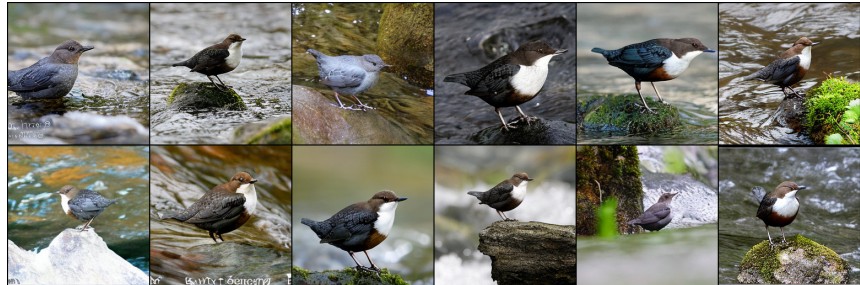

Figure 28: Gen. images conditioned on the class *water ouzel, dipper* using Power-Law CFG with $\omega = 8., \alpha = 0.7$.

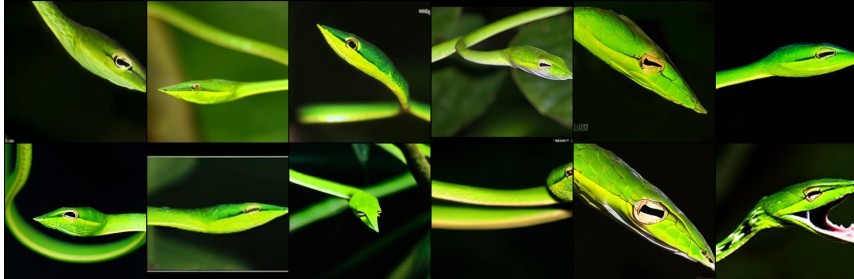

Figure 29: Gen. images conditioned on the class *vine snake* using Standard CFG with $\omega = 4.$.

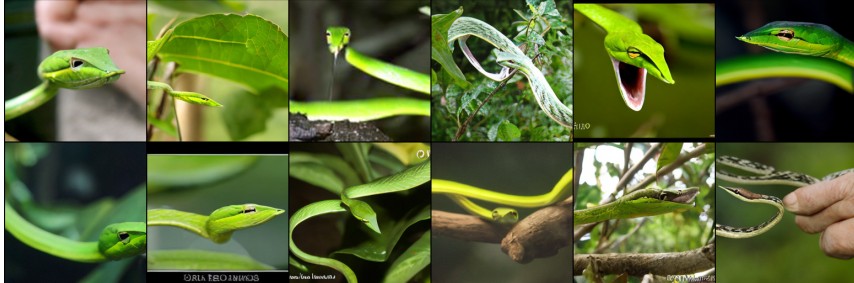

Figure 30: Gen. images conditioned on the class *vine snake* using Power-Law CFG with $\omega = 8., \alpha = 0.7$.

### G.8.4 GENERATED IMAGES BY MMDiT MODEL (DIFFUSION OBJECTIVE, RESOLUTION 512x512)

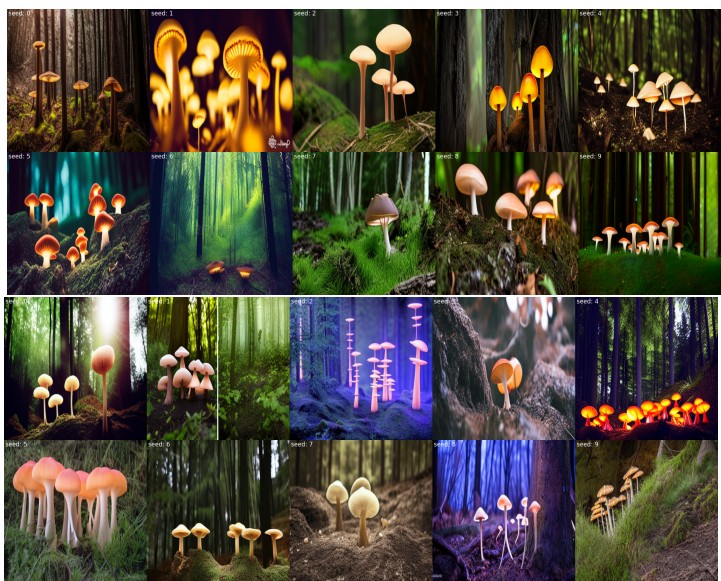

Figure 31: Images generated conditioned on the textual prompt *Glowing mushrooms in a dark forest.* using Standard CFG with $\omega = 3$ (top two rows) and Power-Law CFG with $\omega = 10, \alpha = 0.8$ (bottom two rows).

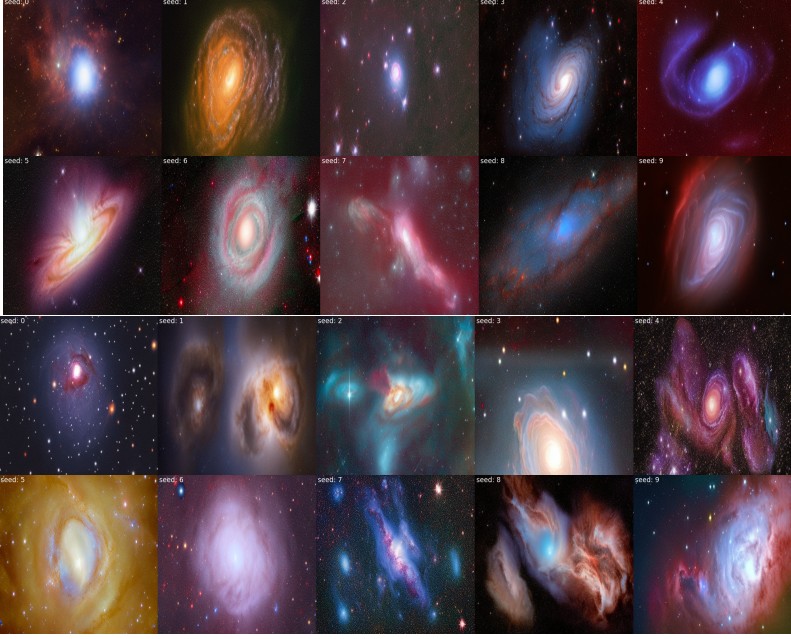

Figure 32: Images generated conditioned on the textual prompt *Stunning, breathtaking view of a galaxy or nebula* using Standard CFG with $\omega = 3$ (top two rows) and Power-Law CFG with $\omega = 10, \alpha = 0.8$ (bottom two rows).

# H FURTHER NOTES ON NON-LINEAR CFG

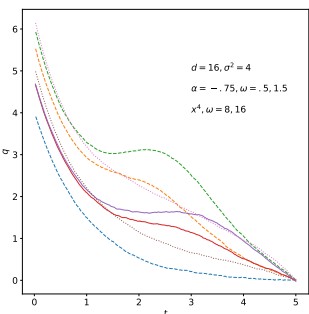

Figure 33: $\langle q \rangle$ versus time in the Gaussian binary mixture with $d = 16$ and $\sigma^2 = 4$. The dashed line are obtained by the standard linear CFG with $\omega = 0, 8, 16$ from bottom to top. The dotted line are obtained with the Power-Law non-linear scheme $f(x) = \omega x^{-.75}$ with $\omega = .5, 1.5$ from bottom to top. The full lines are obtained with the non-linear guidance of Eq.(44) with $\gamma = 4$ and $\omega = 8, 16$ from bottom to top. The Rescaled Power-law non-linear scheme departs from $q = 0$ at large time on a trajectory similar to the linear scheme and to the Power-Law non-linear scheme. But it gives a smaller bias at $t = 0$.

The first non-linear CFG proposal, the power-law CFG with $\phi_t(s) = \omega s^\alpha$ and $\alpha > -1$ (in main paper we focus on $\alpha > 0$ but the guidance can be applied in fact for any $\alpha > -1$) results in the following guidance scheme:

$$\vec{S}_t^{\text{PL}}(\vec{x}, c) = S_t(\vec{x}, c) + \omega \left[ S_t(\vec{x}, c) - S_t(\vec{x}) \right] \left| \vec{S}_t(\vec{x}, c) - \vec{S}_t(\vec{x}) \right|^\alpha. \tag{43}$$

As mentioned, the $\ell_2$ distance between scores $\delta S_t = |\vec{S}_t(\vec{x}, c) - \vec{S}_t(\vec{x})|$ is exponentially small both at the beginning of the backward process (as both conditional and unconditional distributions are standard Gaussian clouds) and before exiting Regime I (as shown in Section 4), after which it remains zero. This non-linear scheme automatically switches off in Regime II and has the following properties: choosing $\alpha < 0$ provides guidance which speeds up convergence to the target at early times, while $\alpha > 0$ dampens the guidance for small $\delta S_t$ and strengthens it for large $\delta S_t$. In practice, we found positive values for $\alpha$ to perform best. In numerical experiments for finite dimension it biases the distribution obtained at $t = 0$ (see Fig.35).

One would like to have different non-linearities applying to the regimes $t \gg t_s$ and $t < t_s$. One possibility is to use the following version, which extends to more general effective distributions $P_0(\vec{a}) e^{-\vec{a}^2 s(t)/(2s(t)^2 \sigma(t)^2)}$ with non-standard $s(t)$ and $\sigma(t)$.

**Rescaled Power-law CFG.** Here, by denoting with $\langle \cdot \rangle$ the expectation w.r.t. the effective distribution $P_0(\vec{a}) e^{-\vec{a}^2 s(t)/(2s(t)^2 \sigma(t)^2)}$, the score difference can be expressed as $|\vec{S}_t(\vec{x}, c) - \vec{S}_t(\vec{x})| = (1/(s(t)\sigma(t)^2) \, |\langle \vec{a} \rangle_{\vec{x},c} - \langle \vec{a} \rangle_{\vec{x}}|$, where $s(t)$ and $\sigma(t)$ are related to the functions $f(t)$ and $g(t)$ by $s(t) = \exp \int_0^t d\tau f(\tau)$ and $\sigma(t) = \int_0^t d\tau g(\tau)^2/s(\tau)^2$. Therefore the non-linear function depends on the difference between the estimators of the initial value $\vec{a}$, given $\vec{x}(t)$, in the class and in the full distribution. This difference is typically a function that decreases with the time of the backward process. This suggests to use a non-linear CFG of the form

$$\vec{S}_t^{\text{RPL}}(\vec{x}, c) = \vec{S}_t(\vec{x}, c) + \omega \left[ \vec{S}_t(\vec{x}, c) - \vec{S}_t(\vec{x}) \right] \left| \langle \vec{a} \rangle_{\vec{x},c} - \langle \vec{a} \rangle_{\vec{x}} \right|^\gamma$$

$$= \vec{S}_t(\vec{x}, c) + \omega \left[ \vec{S}_t(\vec{x}, c) - \vec{S}_t(\vec{x}) \right] \left| \vec{S}_t(\vec{x}, c) - \vec{S}_t(\vec{x}) \right|^\gamma s(t)^\gamma \sigma(t)^{2\gamma}, \tag{44}$$

with positive $\gamma$. As we will show in Figures 33-34, this non-linear guidance term has interesting performance in terms of combining a rapid drift toward the desired class $c$ at early stages of the backward process together with small bias in the finite distribution in finite dimensional problems.

The behavior of both versions is portrayed in Figure 35: both non-lin. versions yield smaller bias at $t = 0$. Furthermore, Figure 35 also displays additional experiments highlighting the benefits of non-linear versions.

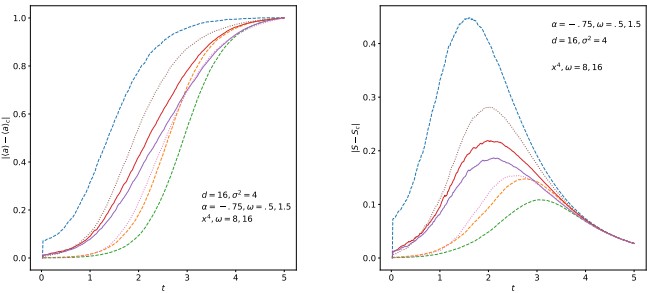

Figure 34: We perform the same experiment as in Fig. 33. Left: the value of $|\langle \vec{a} \rangle_{\vec{x}=\vec{0},c} - \langle \vec{a} \rangle_{\vec{x}=\vec{0}}|$. Right: the value of $|S_t(\vec{x}, a) - S_t(\vec{x})|$, with the same linestyle and color code as in Fig. 33.

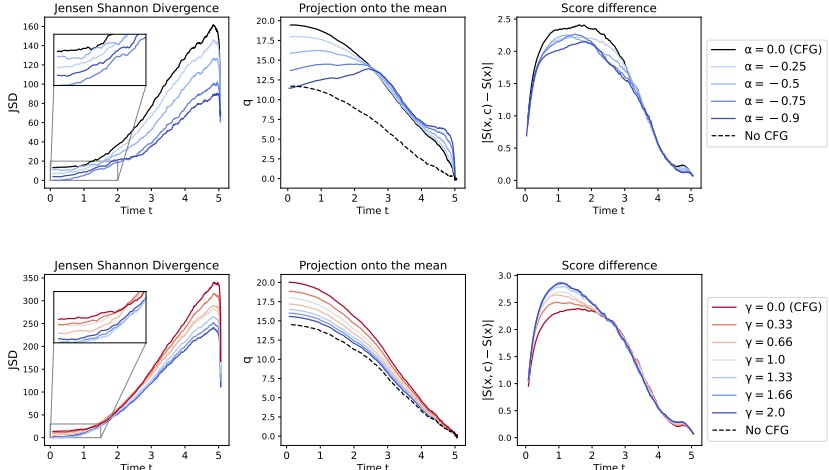

Figure 35: Real-world experiments using DiT/XL-2 (Peebles & Xie, 2023) trained on ImageNet-1000 (Deng et al., 2009): randomly selected class with $\omega = 4$, using DDPM (Ho et al., 2020) with 250 sampling steps, averaged over 25 samples. **First column:** Power-Law CFG. **Second column:** Rescaled Power-Law CFG (44). **Left column:** Jensen-Shannon Divergence between the embedded data points corresponding to randomly selected class and the generated samples as a function of reverse time $\tau$. **Middle column:** mean dot product of the normalized class centroid and the diffusion trajectories $\vec{x} \cdot \vec{c_i}/\|\vec{c_i}\|$ (both in the latent space) as a function of reverse time $\tau$. **Right column:** Evolution of the distance between conditional and unconditional scores. From all three plots, we can see that using first (second) version of non-linear CFG with $\alpha < 0$ ($\gamma > 0$) results in paths that have smaller JSD, estimated as in Wang et al. (2009), throughout the whole trajectory and smaller overshoot of the distribution's mean at $\tau = 0$. We can also see that the score difference $|S_\tau(x, c) - S_\tau(x)|$ has the same qualitative behavior as in numerical simulations of Gaussian mixtures.

