# OpenReview forum: "Overshoot and Shrinkage in Classifier-Free Guidance: From Theory to Practice"
_ICLR.cc/2026/Conference — ICLR 2026 Poster_

### Official Review · Reviewer_QJyR · 2025-10-21

**Soundness:** 3
**Presentation:** 3
**Contribution:** 3
**Rating:** 6
**Confidence:** 3

**Summary:**

This paper shows that the CFG artifacts of overshoot and variance shrinkage that have been observed in low dimensions vanish in high dimensions, via a theoretical analysis of dynamical regimes. They then propose a nonlinear extension of CFG that mitigates the low dimensional artifacts while preserving the high dimensional behavior.

**Strengths:**

The paper is clear and well-written. The problem is well-positioned within the literature (particularly wrt Chidambaram and Biroli). It is nice to see the question of why CFG 'works' despite the known issues in low dimensions being addressed. The dynamical regimes analysis is interesting. The improved diversity on ImageNet is promising.

**Weaknesses:**

Please see questions.

As the authors note in the limitations, although the theory can help improve diversity while maintaining fidelity/quality, it still cannot explain how CFG improves fidelity/quality.

**Questions:**

The Biroli dynamical regimes analysis applied to the CFG deals with a mixture of two Gaussians. How can we be sure the conclusions carry over to more complex multimodal distributions?

How does this analysis relate to the common empirical observation that conditional diffusion models often start to behave like unconditional denoisers at low noise levels? (Does the analysis actually rely on that behavior? Or can it help explain it?)

L328: Do you require $0 < \alpha \le 1$ or can $\alpha$ be greater than zero?

Clarification: (L332) When the difference between conditional and unconditional scores is zero, CFG automatically reproduces the conditional score. So the goal of the power-law is to further (nonlinearly) damp small (but nonzero) score differences?

Why is Power-Law CFG helpful in high-dimensional settings like ImageNet? I thought the analysis showed that CFG is already "correct" in high dimensions (i.e. recovers conditional distribution without distortion)?

Minor edits:
L 82-83 did you swap descriptions of mean overshoot and variance shrinkage?

---

> ### Author Response · Authors · 2025-11-19
> **Reply (1/2)**
>
> We thank the reviewer for their positive assessment and constructive feedback. We address the specific questions below.
>
> > The Biroli dynamical regimes analysis applied to the CFG deals with a mixture of two Gaussians. How can we be sure the conclusions carry over to more complex multimodal distributions?
>
> Thank you for this question.
>
> Our analysis (like that of Biroli et al. [1]) is carried out in the limit of infinite dimensions using a two-component Gaussian Mixture Model (GMM). While this is a simplification, it offers a powerful degree of generality. In this limit, the existence and nature of regimes I/II—and their dynamical properties—can be sharply characterized.
>
> The spirit of our approach is twofold: (1) to analyze CFG in the tractable yet rich setting of infinite dimensions, and (2) to begin with the simplest nontrivial case—a mixture of two Gaussians—to characterize fundamental mechanisms that can guide future work.
>
> However, we should have been clearer with regards to the generality of our framework in the main text.  Our results, as shown in Appendix C, **extend to mixtures involving any finite number of Gaussians with possibly different means and variances**. We have also added a section related to extending our framework to **data supported on hidden manifolds**, following the approaches in [3,5]. This aligns with several studies [2-5] demonstrating that the core features of regimes I/II persist beyond simple Gaussian mixtures and apply broadly to models with data distributed on manifolds. This robustness is a direct consequence of the infinite-dimensional limit, a phenomenon observed in other domains such as supervised learning [6].
>
> Mathematically, our framework can be viewed as a larger dimension concentration result which can be extended to more complex settings whenever each sample can be associated to a given class with probability one. If this property holds at time zero (zero noise) we expect that it will hold until a certain time $t_S$ (certain amount of noise) in the large dimension limit, as a 0-1 law. Below $t_S$ we therefore expect that one can predict with probability one what is the class at which a given backward trajectory ends given its position at (backward) time t>t_S (where t_S is the speciation time of Biroli 2024). In consequence for all times below t_S, conditioning has no effect on the score and the CFG term vanishes in the infinite dimensional limit.
>
> In summary, while our work focuses specifically on the two-Gaussian mixture, the methodology and insights we develop provide a solid and broadly applicable starting point for the theoretical analysis of CFG in more complex settings. We added this clarification in the manuscript.
>
> > How does this analysis relate to the common empirical observation that conditional diffusion models often start to behave like unconditional denoisers at low noise levels? (Does the analysis actually rely on that behavior? Or can it help explain it?)
>
> The reviewer raises a crucial point, which we should have mentioned in the main paper; **our theoretical results explain this empirical observation** which the reviewer mentions. Specifically, our analysis demonstrates that as the noise level decreases, the conditional model converges to the unconditional model behavior. (We also show this holds at very large noise levels, but this result is straightforward as both distributions approach standard Gaussians).
>
> We thank the reviewer for highlighting this connection. We have updated the manuscript to explicitly state that our theory provides a formal explanation for this low-noise behavior (see L 364-367 highlighted in blue), which strengthens the paper's contribution.
>
> > L328: Do you require $0\leq\alpha\leq1$ or can $\alpha$ be greater than zero?
>
> We only require that $\alpha>-1$, so it can be greater than zero. For simplicity of exposition we have found $\alpha=0.9$ to perform well across all experiments, although we believe additional exploration of $\alpha>1$ could yield further benefits.

---

> ### Author Response · Authors · 2025-11-19
> **Reply (2/2)**
>
> > Clarification: (L332) When the difference between conditional and unconditional scores is zero, CFG automatically reproduces the conditional score. So the goal of the power-law is to further (nonlinearly) damp small (but nonzero) score differences?
>
> The reviewer’s interpretation is correct. Theoretically, for very small times (low noise) and very large times, the score difference will be zero (as shown in our Appendix proofs). However, in practice, neural networks make approximation errors. Even small non-zero differences in these regimes can accumulate errors over the sampling trajectory.
>
> The goal of Power-Law CFG is to mitigate these errors. By using $\alpha > 0$, we non-linearly dampen these small, erroneous differences that occur where the theoretical difference should be zero, while further empowering guidance when the differences are large.
>
> > Why is Power-Law CFG helpful in high-dimensional settings like ImageNet? I thought the analysis showed that CFG is already "correct" in high dimensions (i.e. recovers conditional distribution without distortion)?
>
> While datasets like ImageNet are high-dimensional in terms of pixel count, their intrinsic data dimension is not necessarily large enough to fully satisfy the infinite-dimensional limit required for distortion-free recovery.
>
> Now, studies show that this dimension of the hidden data manifold can still be significantly greater than one (e.g., [7,8]), which implies that our perturbative correction of order $1/d$ will be relatively small but not zero. Crucially, because even a large $d$ does not guarantee distortion-free recovery, nonlinear CFG remains beneficial. Furthermore, we note that increasing the number of classes increases class overlap, which our theoretical findings suggest demands an even larger dimension $d$ for proper resolution. The large class overlap is one of the main factors explaining why non-linear CFG performs well in practice.
>
> Additionally, please note that supplementary material contains experiments on ImageNet $256 \times 256$ and $512 \times 512$ in the appendix indicating that the intrinsic dimension is not sufficiently large to recover the conditional distribution without distortion using standard CFG (see Figures 12-13).
>
> However, the reviewer’s intuition is correct: we observe that for lower dimensions, the benefit of Power-Law CFG is even more pronounced. To demonstrate this, we performed three additional experiments on ImageNet $32 \times 32$, ImageNet $64 \times 64$, and CIFAR-10 using the U-ViT architecture. The results (one presented below, the remaining two in our response to reviewer canA), confirm that Power-Law CFG significantly outperforms Linear CFG in these settings (full details are in the Appendix section G.5, outlined in blue):
>
> ### IMNET-64 Results
>
> | Version       | Num. steps | **α** | **ω** | **FID ↓** | **Precision ↑** | **Recall ↑** |
> |---------------|------------|-------|-------|-----------|------------------|--------------|
> | **Lin. CFG**  | 25  | 0   | 1.60 | 5.32 | 0.3996 | 0.4845 |
> |               | 50  | 0   | 1.80 | 4.48 | 0.4484 | 0.4962 |
> |               | 75  | 0   | 1.70 | 4.37 | 0.4576 | 0.5137 |
> |               | 100 | 0   | 1.75 | 4.34 | 0.4511 | 0.5267 |
> | **Nonlin. CFG** | 25  | 0.9 | 3.80 | 4.76 | 0.4212 | 0.4718 |
> |                 | 50  | 0.9 | 4.00 | 4.02 | 0.4656 | 0.5183 |
> |                 | 75  | 0.9 | 4.20 | 3.75 | 0.4734 | 0.5512 |
> |                 | 100 | 0.9 | 4.15 | 3.71 | 0.4883 | 0.5563 |
>
> We thank the reviewer for their critique, as we believe these additional experiments enabled us to strengthen the motivation and our contributions.
>
> > Minor edits: L 82-83 did you swap descriptions of mean overshoot and variance shrinkage?
>
> Thank you for your attention to detail. Yes this was indeed a typo and we have changed it now.
>
> We would like to know whether all of reviewer’s questions and concerns has have been alleviated; if any remain, we would be happy to provide further clarifications/answers.  If the reviewer finds that these clarifications resolve the issues raised, we hope that they will be willing to reconsider their score.
>
> ---
>
> [1] Biroli, G., et al. Dynamical regimes of diffusion models
>
> [2] Ventura et al. Manifolds, Random Matrices and Spectral Gaps: The geometric phases of generative diffusion
>
> [3] Achilli et al. Memorization and Generalization in Generative Diffusion under the Manifold Hypothesis
>
> [4] Bae et al. Diffusion reconstruction for the diluted Ising model
>
> [5] George et al. Analysis of Diffusion Models for Manifold Data
>
> [6] d’Ascoli,et al. Double trouble in double descent: Bias and variance in the lazy regime
>
> [7] Pope P, Zhu C, Abdelkader A, Goldblum M, Goldstein T. The intrinsic dimension of images and its impact on learning.
>
> [8] Facco E, d’Errico M, Rodriguez A, Laio A. Estimating the intrinsic dimension of datasets by a minimal neighborhood information.

---

> > ### Comment · Reviewer_QJyR · 2025-11-25
> >
> > Thank you for the responses. I think the addition of the hidden-manifolds analysis, noting that your theory explains the convergence of conditional to unconditional at low noise, and the additional lower-dimensional ImageNet experiments strengthen the paper. I will keep my positive score.

---

### Official Review · Reviewer_hSQr · 2025-10-30

**Soundness:** 3
**Presentation:** 3
**Contribution:** 3
**Rating:** 6
**Confidence:** 4

**Summary:**

The paper analyzes Classifier-Free Guidance (CFG) for diffusion/flow generative models. Using a high-dimensional dynamical-regime framework, it proves that in sufficiently large dimensions CFG reproduces the correct conditional distribution and accelerates convergence before “speciation” (transition between Regime I and II). In finite dimensions, it formally characterizes mean overshoot and variance shrinkage and attributes them to a more confining effective potential. Motivated by this, it introduces a nonlinear “power-law” CFG that scales the score difference by $|\cdot|^\alpha$, which provably mitigates both artifacts while preserving the high-dimensional guarantee. Experiments span Gaussian-mixture simulations and ImageNet/T2IM models (DiT, EDM2, MMDiT), showing improved FID/recall and robustness to $\omega$.

**Strengths:**

1. The paper's main theoretical punch is the "blessing-of-dimensionality" result. The authors formally prove for the first time that CFG can generate the exact target distribution under appropriate conditions (infinite dimensions).This is a huge deal because it formalizes why this heuristic is so stable and effective in high-dimensional domains like images. (Eq. (4)–(6); Sec. 4.1; Appx. B–D).

2.  The authors' theory also provides a clear explanation for well-known flaws in CFGs. They show that they are mathematically necessary consequences of finite dimensionality and directly relate them to the curvature of the effective potential (Section 4.2; Equation (35)).

3.The authors propose a theoretically derived solution, the power-law CFG, which is very simple. It requires only the addition of a `$|\cdot|^\alpha$` term, and it also has theoretically driven reassurances (Equation (7)).

**Weaknesses:**

1. Theoretical assumptions regarding accurate scores; practical mismatch issues remain unresolved.
The paper obtains true scores under high-dimensional guarantees and finite-dimensional analysis assumptions. The paper acknowledges that it cannot explain why (linear) CFGs perform well with imperfect estimators (Section 3.2, last paragraph; Conclusions/Limitations). Direct robustness analysis of the scores to the error/noise of nonlinear rules (e.g., perturbation bounds) is also lacking.

2. The scope of the study focuses on Gaussian mixture structures.
The core derivation of this paper relies on mixture models with isotropic covariances and separable scaling (Section 4; Appendices B-D); extensions to non-GMM real data are controversial but unproven. While Appendix C discusses multi-component and non-centered mixture models, empirical validation of multi-class schemes focuses primarily on standard vision datasets rather than controlled non-isotropic cases.


3. While $\alpha$ is considered robust ($\alpha \approx 0.9$), the method introduces a second hyperparameter; more extensive ablations (dataset, sampling step size, $omega$ range) are shown in Appendix G, but some edge cases (e.g., extremely small $omega$ or very low step size) require minimal exploration. (Figure 4; Appendix G).

4.  I note that you cite an extension of flow matching with velocity prediction (Appendix G.2.1 notes), but the main theoretical section does not provide a complete derivation.

**Questions:**

refer to weakness

---

> ### Author Response · Authors · 2025-11-19
> **Reply (1/2)**
>
> We sincerely thank the reviewer for their constructive feedback and kind words. We were particularly encouraged by the appreciation of our theoretical contributions as a "huge deal."
>
> Below, we address the weaknesses and questions.
>
>
> > The scope of the study focuses on Gaussian mixture structures. The core derivation of this paper relies on mixture models with isotropic covariances and separable scaling (Section 4; Appendices B-D); extensions to non-GMM real data are controversial but unproven. While Appendix C discusses multi-component and non-centered mixture models, empirical validation of multi-class schemes focuses primarily on standard vision datasets rather than controlled non-isotropic cases.
>
> We thank the reviewer for this suggestion. In the revised supplementary material Section G.6 (highlighted in blue), we have included four additional experiments to compare the behavior of our theoretical framework in numerical simulation environments where we:
>
> 1. Increase the number of Gaussian mixture components.
> 2. Shift the means of the Gaussian components (non-centered).
> 3. Alter the variances of the Gaussian mixture components.
>
> These additional experiments (please see Figures 17-20) provide further insight into how the norms of score differences evolve over time and how they are affected by distribution shifts. Overall, the results align with our theoretical findings: (1) across all experiments, CFG causes mean overshoot and variance shrinkage, (2)as the dimension increases, the effects of mean overshoot and variance shrinkage disappear, and (3) non-linear power-law law guidance helps mitigate these unwanted effects in lower dimensions.
>
> > While $\alpha$ is considered robust ($\alpha\approx0.9$), the method introduces a second hyperparameter; more extensive ablations (dataset, sampling step size, $\alpha$  range) are shown in Appendix G, but some edge cases (e.g., extremely small $\omega$ or very low step size) require minimal exploration. (Figure 4; Appendix G).
>
> Thank you for this interesting suggestion.
>
> Per your suggestion, we have performed two additional sets of edge-case experiments:
> 1. Performance evaluation on very low-dimensional image datasets.
> 2. Evaluation of extremely small $\omega$.
>
> We have performed both evaluations by evaluating a class-conditional U-ViT-like model on ImageNet-1K 32x32, ImageNet-1K 64x64, and CIFAR10 (32x32). Experimental details are provided in our response (see Response (2/2)) and the supplementary material (section G.5, outlined in blue). Before proceeding, regarding very low step sizes, we note that in all reported experiments, we increased the number of sampling steps until performance saturated. For example, for the class-conditional DiT (Table 5 in the Appendix), the number of sampling steps was set to maximum of 250, as increasing it further yielded no significant improvements.
>
> ## Performance evaluation on very low-dimensional image datasets.
> We are grateful for the reviewer's suggestion, which proved highly beneficial. While the FID for high-dimensional experiments improved by roughly 8% across text-to-image and class-conditional, **the mean FID improvement for the low-dimensional experiments was 15%, almost double to the improvement of the high-dimensional ones**. Thanks to the suggestion, we believe this further strengthened motivation for nonlinear CFG.
>
> ### CIFAR-10 Results
>
> | Version       | Num. steps | **α** | **ω** | **FID ↓** | **Precision ↑** | **Recall ↑** |
> |---------------|------------|-------|-------|-----------|------------------|--------------|
> | **Lin. CFG**  | 100  | 0   | 1.55 | 3.50 | 0.6569 | 0.5932 |
> |               | 250  | 0   | 1.40 | 2.77 | 0.6938 | 0.6115 |
> |               | 500  | 0   | 1.35 | 2.03 | 0.6894 | 0.6262 |
> |               | 1000 | 0   | 1.45 | 1.80 | 0.7071 | 0.6186 |
> | **Nonlin. CFG** | 100  | 0.9 | 3.90 | 2.94 | 0.6653 | 0.6111 |
> |                 | 250  | 0.9 | 4.35 | 2.16 | 0.6688 | 0.6298 |
> |                 | 500  | 0.9 | 4.35 | 1.62 | 0.6688 | 0.6298 |
> |                 | 1000 | 0.9 | 4.75 | 1.48 | 0.7228 | 0.6506 |
>
> ### IMNET-32 Results
>
> | Version       | Num. steps | **α** | **ω** | **FID ↓** | **Precision ↑** | **Recall ↑** |
> |---------------|------------|-------|-------|-----------|------------------|--------------|
> | **Lin. CFG**  | 25  | 0   | 1.55 | 5.81 | 0.3842 | 0.4632 |
> |               | 50  | 0   | 1.40 | 4.01 | 0.4161 | 0.4750 |
> |               | 75  | 0   | 1.60 | 3.81 | 0.4120 | 0.4724 |
> |               | 100 | 0   | 1.65 | 3.35 | 0.4306 | 0.4967 |
> | **Nonlin. CFG** | 25  | 0.9 | 3.40 | 4.97 | 0.4022 | 0.4564 |
> |                 | 50  | 0.9 | 3.35 | 3.90 | 0.4241 | 0.4672 |
> |                 | 75  | 0.9 | 3.40 | 3.33 | 0.4431 | 0.5050 |
> |                 | 100 | 0.9 | 3.85 | 2.88 | 0.4539 | 0.5263 |
>
> Detailed results for ImageNet-64x64 can be found in the supplementary material (and in our responses to reviewers canA and QJyR).
>
> (results continued in Reply (2/2))

---

> ### Author Response · Authors · 2025-11-19
> **Reply (2/2)**
>
> ## Evaluation of extremely small $\omega$.
>
> Using the same U-ViT models, we have evaluated and compared unconditional and conditional models to classifier free guidance with $\omega=1+\epsilon$, where $\epsilon\in [1e^{-2},1e^{-3},1e^{-4}]$ and for nonlinear guidance, $\alpha=0.9$ (with experimental details given in the supplementary material):
>
>
> ### Summary FID Comparison
>
> | Method            | CIFAR-10 | IMNET-32 | IMNET-64 |
> |-------------------|----------|----------|----------|
> | Uncond            | 3.1427   | 3.8913   | 5.9304   |
> | Cond.              | 2.4012   | 3.1673   | 4.2776   |
> | Lin. CFG: $\omega= 1.+1e^{-2}$            | 2.4008   | 3.1766   | 4.2950   |
> | Lin. CFG: $\omega= 1.+1e^{-3}$            | 2.4049   | 3.1682   | 4.2907   |
> | Lin. CFG: $\omega = 1.+1e^{-4}$            | 2.4124   | 3.1715   | 4.3094   |
> | Nonlin. CFG: $\omega= 1.+1e^{-2}$       | 2.3907  | 3.1909   | 4.2739   |
> | Nonlin. CFG: $\omega = 1.+1e^{-3}$       | 2.4283   | 3.1890   | 4.2828   |
> | Nonlin. CFG: $\omega = 1.+1e^{-4}$       | 2.4138   | 3.1850   | 4.2851   |
>
>
> We observed that for $\epsilon\leq1e^{-2}$, the performance of non-linear CFG, standard CFG and class-conditioning was roughly the same. For $\epsilon>1e-1$, nonlinear guidance with $\alpha=0.9$ required stronger guidance strength than linear CFG, consistent across all experiments (both class-conditioning and text-to-image).
>
> > I note that you cite an extension of flow matching with velocity prediction (Appendix G.2.1 notes), but the main theoretical section does not provide a complete derivation.
>
> We thank the reviewer for spotting this. We have added a paragraph to the main paper describing (see L350-354 outlined in blue): ‘For velocity flow matching, the power-law formula equals $\phi_t(s)=\omega b_t^\alpha s^\alpha$, where $b_t=(\frac{1-t}{t})^\alpha$, with full derivation given in Appendix G.2.1.
>
>
> > Theoretical assumptions regarding accurate scores; practical mismatch issues remain unresolved. The paper obtains true scores under high-dimensional guarantees and finite-dimensional analysis assumptions. The paper acknowledges that it cannot explain why (linear) CFGs perform well with imperfect estimators (Section 3.2, last paragraph; Conclusions/Limitations). Direct robustness analysis of the scores to the error/noise of nonlinear rules (e.g., perturbation bounds) is also lacking.
>
> We fully agree with the reviewer and have explicitly listed this assumption as a key limitation of our work. Incorporating imperfect score estimators into the theoretical framework is a challenging but vital direction that has been rarely addressed in the literature in the large dimensional limit. We view this as a natural (challenging) next step for theoretical research in this area—not only to guide the creation of new effective guidance mechanisms (as in our work) but also to mathematically explain the empirical utility of CFG despite imperfect scores.
>
> From a theoretical perspective, analyzing the error of score estimation and its propagation to the generated distribution requires a different analytical framework from the one we developed in this work. One would need (1) to use a model for the score (2) to define specific cost functions to investigate how estimation error influences distances between the generated and target probability measures (e.g., Wasserstein distance or KL divergence), (3) solving the optimization dynamics associated to this cost function. Conducting such an analysis in the high-dimensional limit, where both the number of data and their dimension is large, is a highly interesting but also quite challenging and technically distinct project. Consequently, we believe this falls outside the scope of the current work.
>
>
> We would appreciate it if the reviewer lets us know if any remaining concerns of questions remain - we remain at their disposal. We would be happy to answer any further questions. If the reviewer finds that these clarifications resolve the issues raised, we hope that they will be willing to reconsider their score.

---

> ### Comment · Reviewer_hSQr · 2025-11-27
> **Comments**
>
> Thanks for the detailed response. Most of my concerns are addressed, and I will keep my positive score.

---

### Official Review · Reviewer_DodZ · 2025-10-31

**Soundness:** 2
**Presentation:** 3
**Contribution:** 2
**Rating:** 4
**Confidence:** 3

**Summary:**

This paper provides a theoretical analysis of how CFG affects the generation of mixtures of Gaussians via diffusion models. The authors arrive at the result that in the scenario of exact score estimation the effect of CFG on the generated distribution tends to zero as the number of data dimensions tends to infinity. Conversely, they find that in finite/small dimensional cases CFG causes the generated means to shift and the variances to shrink. The authors propose to adjust CFG according to the norm of the score difference in order to mitigate the above effects and evaluation on various image generation models/datasets.

**Strengths:**

- The theoretical contributions advance the understanding of CFG.
- The presentation of the paper is clean and professional. Theoretical results are presented in a structured and understandable manner.
- The scale and variety of the experiments are appropriate.

**Weaknesses:**

The paper load for ICLR this year has been large, and so I have not been able to spend as much time as I would like on reviewing. I encourage the authors to correct any errors/misunderstandings I may have with regards to the paper.


1. **weak motivation for proposed approach**
    1. The authors motivate their Power-Law CFG as a method to reduce mean overshoot and variance shrinkage (implying generation is in the *finite dim* regime), but then justify their approach by saying it retains the *high dim* properties of CFG that they've derived. It seems like they are suggesting that their approach won't "break" the nice high-dimensional properties of CFG, but then are focused on addressing a problem that only exists in low/finite dim regime. I find this link between the theoretical and practical part of the paper quite weak.
    1. It is unclear to me how the proposed approach reduces the mean overshoot and variance shrinkage, as the authors don't provide an explanation in the main body of the paper.
    1. From what I can tell, any heuristic scaling of CFG [1] (e.g. according to timestep) will also retain the nice theoretical properties, which hardly makes power-law scaling unique. Besides, practically speaking it seems like the proposed approach behaves similarly to existing heuristics, gradually increasing CFG as generation progresses [1].
    1. The intuitive description of power-law scaling is misleading. First of all the assertion that score difference large -> reliable, small -> unreliable is not clearly explained or motivated. Secondly, $x_t=\alpha_t x_0 + \sigma_t \epsilon$, $S_\theta = -\epsilon_\theta/\sigma_t$, that is to say from a noise prediction perspective the magnitude of the score estimate depends on the noising schedule. As noise level $\sigma_t$ decreases over the course of generation this will have a direct impact on the power-law scaling. Importantly, the $\sigma$ scaling has nothing to do with the actual difference in spatial content predicted by the conditional and unconditional models and is instead dependent on the diffusion time. In fact, the authors acknowledge that applying their scaling directly to flow velocity results in different behaviour, but do not offer much additional insight.


2. **Tepid experimental results**
    1. Although improvements over vanilla CFG seem notable, improvements over other improved CFG methods are much more marginal. Moreover, due to the aforementioned weak motivation there isn't really a clear explanation for why to prefer power-law scaling to competing approaches.

3. **Lack of intuition to support theory (minor)**
   1. For a general reader interested in using/improving CFG, I feel like this paper would benefit from more intuitive explanations to help understanding. Personally, I still have not been able to build an intuition for why the effect of CFG disappears when dimensionality tends to infinity, making it harder to appreciate the contribution. (I have some vague ideas of distributions tending to hyperspherical shells, leading to different behaviour, but nothing concrete.)

[1] Wang et al, Analysis of Classifier-Free Guidance Weight Schedulers, TMLR 2024.

**Questions:**

See above

My main issues with the paper are (according to my current understanding)
1. The proposed approach is weakly linked with the theoretical contribution.
2. The concrete form of the power-law CFG is not well motivated.

I'm open to increasing my score if the authors can help me understand better the above issues. I have also not read the appendices in detail.

Additionally, I would like to clarify, is the score difference a vector norm or an elementwise difference?

---

> ### Author Response · Authors · 2025-11-19
> **Response (1/3)**
>
> We thank the reviewer for their positive assessment of our work, specifically highlighting our contribution in advancing the understanding of CFG and the scale of our experiments. We address the reviewer's questions and concerns below.
>
> ### W1 On weak motivation for proposed approach
>
> > 1. The authors motivate their Power-Law CFG as a method to reduce mean overshoot and variance shrinkage, but then justify their approach by saying it retains the high dim properties of CFG that they've derived. .. they are suggesting that their approach won't "break" the nice high-dim. properties of CFG, but then are focused on addressing a problem that only exists in low/finite dim regime.
>
> We apologize if the narrative flow was unclear. The logical development of our method actually proceeded in the inverse order of the reviewer's interpretation. We have revised the main paper to reflect this timeline more clearly:
>
> 1. We first establish that, contrary to previous works [1,2], CFG can generate the correct target distribution (provided the dimension is sufficiently large or infinite).
> 2. We then connect these findings to prior work showing "mean overshoot" and "variance shrinkage" arise when the dimension decreases (the finite regime).
> 3. Finally, we demonstrated that these finite-dimensional artifacts can be easily mitigated via nonlinear guidances. We propose a simple method that performs well.
>
> Our approach is based on the fact that the dimension of the data is large but finite; in practice the data usually lies on a manifold with whose dimension is lower than the ambient space, but still much larger than 1 (see e.g., [3,4]). In consequence, we use the infinite dimensional limit as a starting point and then consider the perturbative correction around it. As this intrinsic dimension $d$ is still rather large and the perturbative correction is of small order ($1/d$), allowing to conclude that even for finite-dimensional data, our framework will hold. This provides a theoretical principle to study large but finite dimensional data, ultimately leading to guidelines for practice.
>
> The Power-Law CFG was proposed as the simplest, most straightforward example of nonlinear guidance. In the Appendix, we show that popular methods (e.g. limited guidance and guidance schedulers) all fall under our theoretical framework. We also analyze another nonlinear method theoretically; however, we focused the main text on Power-Law because it is simple, does not introduce additional hyperparameters (as we fix $\alpha=0.9$), and yields significant empirical improvements.
>
> Our core motivation was not to claim Power-Law is the only solution, but to argue via theoretical analysis that mean overshoot and variance shrinkage can be easily mitigated by a class of nonlinear functions, of which Power-Law is a representative example. We hope this clarifies the link between the theory and the practical method.
>
> We appreciate if the reviewer could let us know if this clarifies our motivation for them.
>
> > 2. It is unclear to me how the proposed approach reduces the mean overshoot and variance shrinkage, as the authors don't provide an explanation in the main body of the paper.
>
> We have added a summary of our findings from Appendix E to the main text (see L342-350 in blue) to clarify this. To summarize: by examining CFG in finite dimensions, we identified a key term, $B(q) := d e^{-(t_f-t)}/\Gamma u(q)$ (see Appendix lines 1500-1510 for detailed explanation of each term), which drives mean overshoot and variance shrinkage. This term is tightly linked to the Hessian (curvature) of the potential. By analyzing the effect of nonlinear guidance on curvature, we observe that it directly modifies $B(q)$ by raising it to the power $\alpha$, where we show that $0<B(q)<1$ almost always. Specifically, we show in Appendix E.1 that:
>
> - **Mean Overshoot**: larger B(q) means larger overshoot. Therefore, the parameter $\alpha$ directly controls the overshoot via $B(q)^\alpha$. Specifically, $\alpha > 0$ suppresses/diminishes the overshoot, whereas $\alpha < 0$ increases it.
> - **Variance Shrinkage**: larger B(q) means larger variance shrinkage. Luckily, the parameter $\alpha$ also controls the shrinkage directly via $B(q)^\alpha$. Specifically, $\alpha > 0$ reduces the curvature increase caused by CFG (resulting in a less confining potential and thus decreasing variance shrinkage), while $\alpha < 0$ amplifies it.
>
> We thank the reviewer for pointing this out as we believe it is important for future readers.

---

> ### Author Response · Authors · 2025-11-19
> **Response (2/3)**
>
> > 3. Any heuristic scaling of CFG [1] will retain theoretical properties, which hardly makes power-law scaling unique. It seems like the proposed approach behaves similarly to existing heuristics [1].
>
> The reviewer is correct. The only requirement for a guidance method to fall within our theoretical framework (see L364 in the main paper) is satisfied by the majority of popular guidance techniques. This requirement, stated as $\lim_{s \to 0} s  \phi_t(s) = 0$, ensures guidance vanishes smoothly when cond. and uncond. scores coincide. This allows us to demonstrate two key points:
>
> 1. As the dimension increases, all these guidance methods will generate a correct target distribution.
> 2. These existing guidances can further benefit from adding Power-Law scaling on top of them (as observed in our experiments).
>
> We believe the generality of our theoretical framework is one of its primary strengths. As noted above, we chose Power-Law for its analytic simplicity, which allowed us to prove it reduces variance shrinkage and mean overshoot. It is by no means unique. As mentioned in the paper, we believe there exist many other nonlinear guidances—potentially with the guidance function optimized as a hyperparameter—that could yield even better performance.
>
> > 4. The intuitive description of power-law scaling is misleading, the assertion ... is not clearly explained or motivated. The  scaling has nothing to do with the actual difference in spatial content.. The authors acknowledge that applying their scaling directly to flow velocity results in different behaviour, but do not offer much additional insight.
>
> Regarding the velocity flow matching, we did not intend to imply that nonlinear guidance results in fundamentally different behavior. In velocity-based flows, one instead has the velocity difference $|v_t(x,c)-v_t(s)|$. We simply wished to clarify that, in order be fully consistent with our theoretical proposition, we scale the velocity difference to determine the score difference $s$ rather than applying the $v^\alpha$ directly. By re-scaling to determine the score differences and keeping in line with our manuscript notation, the Power-Law formula is then expressed as $\phi_t(s)=(\frac{1-t}{t})^\alpha s^\alpha$, rather than $\phi_t(s)=s^\alpha$. However, even using $\phi_t(s)=s^\alpha$ (which is a valid guidance within our framework) yields very similar results and behavior. We have rephrased the main text to make this distinction clear.
>
> Regarding the intuitive description, we accept this is a valid critique. Our main theoretical point is that for very small and very large noise levels (or times), the score difference approaches zero (we show this theoretically for standard CFG). In practice, however, neural networks make approximation errors. One interpretation of why $\alpha > 0$ significantly outperforms $\alpha < 0$ across all experiments is that $\alpha > 0$ effectively dampens the score difference when that difference is small, and enlarges it when the difference is large, dampening the practical approximation errors. This observation holds true regardless of the specific $\sigma_t$ scheduler used (note that DiT, EDM, and our text-to-image models all use different schedulers, yet nonlinear CFG improves performance across all of them).
>
> We hope we correctly interpreted the reviewer’s question, and if not, we would appreciate it if the reviewer could let us know.
>
> ### W2 On experimental results
> > More marginal improvements over other improved CFG methods and weak motivation.
>
> The reviewer raises a fair point. We would like to highlight three specific advantages of our approach:
>
> 1. **Ease of parameter selection**: As shown in the right panel of Figure 4, with $\alpha \approx 0.9$, the "basin" of $\omega$ parameters that yield superior FID is significantly wider. This makes the search for an optimal $\omega$ much cheaper and less sensitive compared to standard CFG (where $\alpha=0$).
> 2. **No additional hyperparameter cost**: Unlike limited guidance (which adds two parameters) or schedulers (which add one or more), our method yields consistent performance improvements with a fixed $\alpha=0.9$.
> 3. **Generality**: As discussed in our previous response, we do not claim Power-Law is the optimal nonlinear guidance, but rather a proof of concept that "nonlinear guidances exist that mitigate mean overshoot and variance shrinkage." We proposed Power-Law for its simplicity, its lack of hyperparameter overhead, and as a guiding principle for future investigation into even more effective nonlinear guidances.

---

> ### Author Response · Authors · 2025-11-19
> **Response (3/3)**
>
> ## W3 On intuition to support theory
> > The paper would benefit from intuitive explanations to help understanding.
>
> We thank the reviewer for this feedback. The intuition can be described as follows: as the dimension grows, the "attraction force" for each class grows exponentially. One can visualize a well-separated class acting as a magnet that pulls diffusion trajectories towards itself (the attraction force depending on the mean and variance magnitude). During the backward process this force becomes very strong when the system starts to approach one class. Unless the magnitude of the CFG guidance scale also grows exponentially with the dimension (which is not the case in practice), these natural attraction forces completely dominate the CFG term.
>
> Mathematically, our framework can be viewed as a larger dimension concentration result: one can predict with probability one what is the class at which a given backward trajectory ends, given its position at (backward) time $t>t_S$ (where $t_S$ is the speciation time of Biroli 2024). In consequence, for all times above the speciation time, conditioning has no effect on the score and the CFG term vanishes.  For the reviewer’s convenience, we have added a sketch illustrating how these regimes arise to the supplementary material (please see Figure 6). We hope this comparison provides a concrete intuition.
>
> > Additionally, I would like to clarify, is the score difference a vector norm or an elementwise difference?
>
> It is a vector norm. We have added this clarification in the paper (see L333).
>
> We hope that these explanations have sufficiently addressed the reviewer's concerns and that our revisions have clarified the narrative of the paper. We would be happy to answer any further questions. If the reviewer finds that these clarifications resolve the issues raised, we hope that they will be willing to reconsider their score.
>
> [1] Chidambaram, Muthu, et al. "What does guidance do? a fine-grained analysis in a simple setting." arXiv preprint arXiv:2409.13074 (2024).
>
> [2] Wu, Yuchen, et al. "Theoretical insights for diffusion guidance: A case study for gaussian mixture models." arXiv preprint arXiv:2403.01639 (2024).
>
> [3] Pope P, Zhu C, Abdelkader A, Goldblum M, Goldstein T. The intrinsic dimension of images and its impact on learning. arXiv preprint arXiv:2104.08894. 2021 Apr 18.
>
> [4] Facco E, d’Errico M, Rodriguez A, Laio A. Estimating the intrinsic dimension of datasets by a minimal neighborhood information. Scientific reports. 2017 Sep 22;7(1):12140.

---

> > ### Comment · Reviewer_DodZ · 2025-11-19
> >
> > Thanks for the long and detailed response to my review (as well as for the other reviewers). I'm a little busy over the next few days, unfortunately, but I hope to get back to the authors by the end of Friday.

---

> > > ### Comment · Reviewer_DodZ · 2025-11-22
> > >
> > > The authors' response has been very helpful, and the edits to the manuscript have improved it. I have a few follow-up questions/suggestions for the authors.
> > >
> > > 1. The intuitive explanation you've provided makes it much easier for me to understand your theoretical result. I would appreciate if you could include such a natural explanation in the main body if possible. As an aside, it has prompted another thought which is that it is possible to view text-to-image training as training on a mixture of delta functions where each delta is a zero-variance Gaussian (if each prompt/image pair is unique). Besides, as soon as noise is added to any degree, a mixture of deltas becomes a mixture of Gaussians. Perhaps the authors' analysis could be linked to T2I training generally, or prompt-image memorisation, where the model has learnt to map to a delta given a memorised prompt. (Just food for thought for additional discussion or future work.)
> > > 2. I now understand the "narrative"/motivation of the paper more clearly, and moving a summary of the theoretical analysis of the power law CFG to the main body is also helpful. However, I am still a little iffy on the expression for power law CFG. Now that it has been clarified that it contains a vector norm, I believe the authors need to discuss how its practical effect will vary with image resolution. Intrinsic/manifold dimensionality of images (according to my intuition) won't change much between a 32x32x4 latent and a 512x512x3 pixel-space image, however, the vector norm I would expect to be wildly different -- as such I would expect the power-law scaling to potentially not be robust if ported to a high-resolution pixel-space diffusion model. Would the authors suggest a resolution-based adjustment? With regards to the noising schedule score scaling, it seems like the authors have misunderstood my previous comment. What I meant to express is that a large part of the magnitude of the score difference arises directly from the diffusion-time-based scaling $\sigma_t$ rather than directional/spatial differences between the conditional and unconditional score. That is to say, two completely opposite direction cond/uncond scores may have a smaller difference for $t\approx 1$ compared to much more aligned scores for smaller $t$. Could the authors clarify whether this "built-in" scaling is important or not? Would replacing the difference vector norm with a cosine distance improve or harm performance?
> > > 3. I think the paper would benefit from an intuitive link (in the introduction) between the abstract ideas of mean-overshoot and variance shrinkage and concrete image-generation concepts that CFG is known to hurt, such as diversity and saturation/contrast. This would help better connect everything together.

---

> > > > ### Author Response · Authors · 2025-11-25
> > > > **Response (2/2)**
> > > >
> > > > > With regards to the noising schedule score scaling, it seems like the authors have misunderstood my previous comment. What I meant to express is that a large part of the magnitude of the score difference arises directly from the diffusion-time-based scaling  rather than directional/spatial differences between the conditional and unconditional score. That is to say, two completely opposite direction cond/uncond scores may have a smaller difference for $t\approx1$ compared to much more aligned scores for smaller $t$. Could the authors clarify whether this "built-in" scaling is important or not?
> > > >
> > > > Thank you for the clarification. Theoretically, the score difference decays exponentially fast for very small and large $t$; in sufficiently high dimensions, this specific scaling is less critical. However, in finite dimensions, one can theoretically define non-linear guidance of the form $\phi_t(s, \sigma_t)=s^\alpha f(\sigma_t)$, where $f(\sigma_t)$ is derived specifically for a particular scheduler choice $\sigma_t$ to further reduce overshoot and variance shrinkage. Such schedule-specific guidance indeed fit within our framework, however we prioritized the standard Power-Law formulation for its simplicity and robust performance across different schedulers without requiring complex derivation.
> > > >
> > > > >  Would replacing the difference vector norm with a cosine distance improve or harm performance?
> > > >
> > > > An interesting suggestion! Using cosine distance is theoretically justified within our framework. However as it does not involve any hyperparameters, we thought it would be fair to also try out raising the cosine difference to the power $\alpha$ and tune for $\alpha$. We therefore we compared "Cosine CFG" and "Cosine CFG w. $\alpha$" against standard and Power-Law CFG using EDM-2/S on ImageNet (512x512 latent):
> > > >
> > > >
> > > > ### EDM-2/S ImageNet-1K 512x512 (latent space)
> > > >
> > > >
> > > > | Version        | FID ↓ | FID_DINO ↓ |
> > > > |----------------|--------|-------------|
> > > > | Standard CFG  | 2.29 | 54.76 |
> > > > | Cosine CFG    | 2.26 | 54.28 |
> > > > | Cosine CFG w. $\alpha$    | 2.21 | 54.09 |
> > > > | Nonlinear CFG | 1.93 | 52.77 |
> > > >
> > > > While Cosine CFG slightly outperforms Standard CFG and Cosine CFG w. $\alpha$ further improves it, they do not match the performance of Power-Law CFG. We hope that this motivates the reviewer (as well as future readers) to come up and test other non-linear guidances in order to further improve performance.
> > > >
> > > > > I think the paper would benefit from an intuitive link (in the introduction) between the abstract ideas of mean-overshoot and variance shrinkage and concrete image-generation concepts that CFG is known to hurt, such as diversity and saturation/contrast. This would help better connect everything together.
> > > >
> > > > We have added this intuitive link in the introduction (L83-L87, highlighted in green), explicitly connecting our theoretical framework (mean-overshoot and variance shrinkage) to practical observations regarding saturation, contrast, and diversity reduction. We believe this bridges the gap between theory and practice for a broader audience.
> > > >
> > > > We hope that these new explanations/manuscript additions, along with the additional pixel-space experiments and the Cosine CFG comparison, have successfully addressed your remaining concerns. If there are any outstanding questions or suggestions, please let us know.
> > > >
> > > >
> > > > [1] Gu, J., Zhai, S., Zhang, Y., Susskind, J. and Jaitly, N., 2023. Matryoshka diffusion models. arXiv preprint arXiv:2310.15111.
> > > >
> > > > [2] Chen, S., Ge, C., Zhang, S., Sun, P. and Luo, P., 2025. PixelFlow: Pixel-Space Generative Models with Flow. arXiv preprint arXiv:2504.07963.

---

> > > > > ### Comment · Reviewer_DodZ · 2025-11-26
> > > > >
> > > > > Thanks for following up and updating the manuscript. With regards to argument of euclidean difference vs cosine similarity, I think it is still important to include a note on this in the paper when introducing the method. For the readers' benefit, I think they should be aware that the scaling due to score difference is not just due to the unconditional and conditional scores being in different directions, but also due to the score magnitudes as well (e.g. for the cosine case, epsilon difference and score difference lead to equivalent adjustments, but for Euclidean distance they are different.) This can be done with a footnote pointing to a paragraph in the appendix.
> > > > >
> > > > > After the above is added I will be happy to raise my score to a 6. I do not feel comfortable giving a score of 8, purely because I do not have the background/expertise in verifying the correctness of the authors' theoretical contribution, which is a large part of the paper's contribution. I hope the authors and AC understand this choice.

---

> > > > > > ### Author Response · Authors · 2025-11-26
> > > > > >
> > > > > > We thank the reviewer for the follow-up and the constructive suggestion to clarify the scaling property of the Euclidean difference and the argument between Euclidean difference and Cosine similarity in the main text.
> > > > > >
> > > > > > We agree that readers should be aware that the power-law scaling is driven by both the direction and the magnitude of the score difference. In response, we have added a dedicated paragraph in the main text (highlighted in green, Lines 394–400) to explicitly discuss this distinction and how the guidance parameter $\omega$ acts as a renormalization factor across different resolutions.
> > > > > >
> > > > > > The added text is as follows:
> > > > > >
> > > > > > **Magnitude vs. Directional Scaling.** Our Power-Law formulation uses the Euclidean norm $\|S_t(\vec{x},c) - S_t(\vec{x})\|$ which captures both score *direction* and *magnitude*, unlike, e.g., cosine metrics that would isolate alignment. While the Euclidean norm is resolution-sensitive, luckily, the guidance parameter $\omega$ acts as a renormalization factor, which ultimately removes the need for an additional explicit resolution-based adjustments. As detailed in App. G.7, our analyses showed that this combined signal yields superior performance compared to purely directional metrics, though alternative cosine-based methods remain a promising avenue for future work.
> > > > > >
> > > > > > Furthermore, to ensure the experimental evidence supporting these claims is available to the reader, we have included a new subsection in Appendix G.7 (Lines 2155–2194). This section contains all the analyses we conducted in light of your suggestions; including the pixel-space robustness tests (including Matryoshka and PixelFlow) and the comparative results with "Cosine CFG" and "Cosine CFG w. $\alpha$" on EDM-2/S ImageNet1K.
> > > > > >
> > > > > > All new additions to the text can be found in \green{green}. We extend our gratitude to the reviewer for their continued engagement and fruitful discussion.

---

> ### Author Response · Authors · 2025-11-25
> **Response (1/2)**
>
> We thank the reviewer for their continued engagement and detailed feedback.
>
> > The intuitive explanation you've provided makes it much easier for me to understand your theoretical result. I would appreciate if you could include such a natural explanation in the main body if possible.
>
> We are glad the intuitive explanation proved helpful. As per your suggestion, we incorporated it into the main body, alongside a figure depicting the backward dynamics and attraction forces (please see lines 118-131, highlighted in green).
>
> > As an aside, it has prompted another thought which is that it is possible to view text-to-image training as training on a mixture of delta functions where each delta is a zero-variance Gaussian (if each prompt/image pair is unique). Besides, as soon as noise is added to any degree, a mixture of deltas becomes a mixture of Gaussians. Perhaps the authors' analysis could be linked to T2I training generally, or prompt-image memorisation, where the model has learnt to map to a delta given a memorised prompt. (Just food for thought for additional discussion or future work.)
>
> We agree with the reviewer’s insight. If the mapping is unique, training can indeed be viewed as a mixture of zero-variance Gaussians (deltas). Conversely, if the mapping is not unique, learning can be viewed as a mixture of Gaussians (one Gaussian corresponding to all possible images for a single prompt). We believe this strengthens the link between our work and general T2I training - we thank the reviewer for their suggestion. We have added a discussion on this point to the main text (L273-279, highlighted in green).
>
> > I believe the authors need to discuss how Power-Law’s practical effect will vary with image resolution. Intrinsic/manifold dimensionality of images (according to my intuition) won't change much between a 32x32x4 latent and a 512x512x3 pixel-space image, however, the vector norm I would expect to be wildly different -- as such I would expect the power-law scaling to potentially not be robust if ported to a high-resolution pixel-space diffusion model. Would the authors suggest a resolution-based adjustment?
>
> The reviewer raises a good point. For power law CFG, the overall scale matters since not all terms scale in the same way (for linear CFG the scale of $S_t$ does not matter  since all terms are linear in $S_t$). Theoretically, although the overall shape of the score difference curve will stay the same (for very small and large times the difference will tend to zero with a hump-shaped behavior in the middle), the overall scale does indeed change.  For power law CFG, choosing the optimal omega will renormalise the difference of the score in that regime. We agree that this corresponds to a resolution based adjustment.
>
>
> To go beyond the qualitative arguments above, we also provide some numerical tests as practical justification. Therefore, we have evaluated whether power-law CFG would improve performance of standard CFG in two pixel-space methods based on the Matroyshka-diffusion model [1] and PixelFlow [2] flow-matching model. The results can be found in the table below.
>
>
>
>
> ### Matryoshka diffusion ImageNet-1K 256x256 (pixel space)
>
>
> | Version        | FID ↓ | Precision ↑ | Recall ↑ |
> |----------------|-------|--------------|----------|
> | Standard CFG  | 3.51 | 0.81 | 0.54 |
> | Nonlinear CFG (fixed $\alpha=0.9$) | 3.39 | 0.82 | 0.55 |
> | Nonlinear CFG (tuned $\alpha=0.65$) | 3.17 | 0.83 | 0.56 |
>
>
>
>
> ### PixelFlow flow matching ImageNet-1K 256x256 (pixel space)
>
>
> | Version        | FID ↓ | Precision ↑ | Recall ↑ |
> |----------------|-------|--------------|----------|
> | Standard CFG | 2.43 | 0.83 | 0.56 |
> | Nonlinear CFG (fixed $\alpha=0.9$) | 2.29 | 0.83 | 0.58 |
> | Nonlinear CFG (fixed $\alpha=0.45$) | 2.12 | 0.85 | 0.59 |
>
>
> We would like to note that, while fixing $\alpha=0.9$ yields improvements, pixel-space models benefit more from tuning both $\alpha$ and $\omega$ compared to latent-space models, confirming the reviewer's intuition regarding possible different behavior in norm differences between pixel- and latent-space. We have added these experiments (Table 2) and a discussion on latent-space vs pixel-space robustness to the paper (L516-521, highlighted in green).

---

### Official Review · Reviewer_canA · 2025-11-02

**Soundness:** 3
**Presentation:** 2
**Contribution:** 3
**Rating:** 4
**Confidence:** 3

**Summary:**

The paper observes the problem of mean overshooting and variance shrinkage resulted by classifier-free gudiance during Diffusion Sampling process. The main results are drawn from the connection of dynamical regimes and classifier-free guidance. Main observations are conducted on  the mixture of Gaussian (2 modes) with the same weight and variance. This gives us the division of CFG into three distinct steps (1) guidance before speciation (2) align before exiting regime I (3) follow unguided path. However, when the dimensionality is limited, the observation (2) is no longer correct and over push the conditional information. To solve this problem in lower-dimension setting, the authors propose power-law guidance where it will enhance the guidance when the guidance signal is strong, and purging the guidance when the guidance is weak. The results show the significant improvement

**Strengths:**

1. The paper is well-written
2. The theoretical part is well-written
3. The experiments are comprehensive and up-to-dated.
4. The improvement is quite significant
5. The proposed method is simple

**Weaknesses:**

The main drawback of the paper lies in its over-complication. In my opinion, simplicity should be viewed as a strength rather than a weakness. The proposed method is inherently simple, and the results are largely empirical. Therefore, I strongly suggest simplifying the presentation and avoiding unnecessary theoretical analysis that does not contribute to the core message.

1. **Overly Strong Assumptions:**
   The assumptions made in the paper are excessively restrictive and unrealistic.
   - The use of a mixture of two Gaussians is not representative of most real-world datasets.
   - The assumption that both Gaussians share the same variance is also impractical and lacks justification.
   - Furthermore, assuming the means to be +m and –m is arbitrary and overly specific.
   In contrast, *Biroli (2024)* assumes only the existence of two classes without making any assumptions about their distributions, means, or variances. The authors may want to consider relaxing or revising these assumptions to enhance realism and generalizability.

2. **Unnecessary First Key Finding:**
   The first key finding adds confusion rather than clarity. It does not appear essential to the overall narrative or conclusions. The paper would benefit from removing this section and instead starting directly from the second finding (e.g., Figures 8 and 12), focusing on the proposed method for addressing low-dimensional cases.

3. **Weak Connection to Previous Findings:**
   The third key finding seems only loosely connected to the earlier results. It is unclear how the concepts of “Regime I” and “Regime II” relate to the presented equation. This section requires clearer logical linkage and explanation.

4. **Experimental Results and Resolution:**
   The experiments report results for resolutions of 512 and 256. According to the paper’s own claims, one might expect the method to perform even better on lower resolutions such as 64×64 or 32×32. If this is indeed the case, the authors should provide empirical evidence to support this claim.

**Questions:**

Please see the weakness. I would encourage the authors to simplify the paper.

---

> ### Author Response · Authors · 2025-11-19
> **Response (1/2)**
>
> We thank the reviewer for their detailed evaluation and for highlighting that our experiments are comprehensive and up-to-date. We are encouraged that the reviewer finds the performance improvement significant and appreciates the simplicity of the method. We address the specific concerns and questions below.
>
> ### W1 Overly strong assumptions
>
> > The use of a mixture of two Gaussians is not representative of most real-world datasets. The assumption that both Gaussians share the same variance is also impractical and lacks justification.
>
> We appreciate the feedback. While we utilized a simple Gaussian mixture model to maintain a clear/simple exposition in the main text, in Appendix C, we provide generalizations of our proofs that extend beyond these simplifying assumptions. Specifically, we consider:
>
> 1. Gaussians centered on different means with different variances.
> 2. Extensions to mixtures involving any finite number of Gaussians.
>
> We also added a new section outlining the approach to extend our framework to data supported on manifolds (following the approaches in [1, 2]).
>
> Therefore, our theoretical framework in fact goes even beyond Biroli (2024), i.e., making even weaker assumptions than in the work by Biroli (2024). Hence, we do extend our approach to more general cases. We agree that outlining the generalizability and the limitations of our analysis is indeed crucial, and we have done it in the revised version (see part outlined in blue starting from L222).
>
> To address the reviewer's concern regarding clarity, we have revised the manuscript to explicitly emphasize the extendability of our proofs. The spirit of our approach is to begin with the simplest nontrivial case—a mixture of 2 Gaussians—to characterize fundamental mechanisms that can guide future work. However, as noted, the natural extension of our work is to consider mixtures of any finite number of Gaussians with differing means and variances.
>
>
>  > Furthermore, assuming the means to be +m and –m is arbitrary and overly specific. In contrast, Biroli (2024) assumes only the existence of two classes without making any assumptions about their distributions, means, or variances. The authors may want to consider relaxing or revising these assumptions to enhance realism and generalizability.
>
> We would like to clarify again that our theoretical framework (within main manuscript) operates under the same assumptions as made by Biroli et al. (2024). In fact, the additional proofs in Appendix C (mentioned above) provide a more general framework, relaxing the constraints on means and variances, as well as the number of gaussian mixture components.
>
>
> ### W2 On the necessity of our first finding
>
> > The first key finding adds confusion rather than clarity. It does not appear essential to the overall narrative or conclusions. The paper would benefit from removing this section and instead starting directly from the second finding (e.g., Figures 8 and 12), focusing on the proposed method for addressing low-dimensional cases.
>
> We respectfully argue for keeping this result in the main paper. We believe this finding is important for two main reasons:
> 1. **Novel finding addressing previous theoretical gaps**: Prior to our work, previous theoretical works [3, 4] suggested CFG would never result in the correct target distribution. Our first finding in fact shows that CFG will generate the correct target distribution when the dimension is sufficiently high. We believe this is a significant theoretical contribution that will be of interest to the community.
> 2. **Foundation for Practical Guidance**: Our proposed nonlinear guidance directly stems from the attempt to prove that CFG can generate the correct distribution. The "blessing of dimensionality" serves as the theoretical starting point that guided us toward our practical method (we generalize CFG in a way that still enjoys the blessing of dimensionality result).
>
> We note that other reviewers (DodZ, hSQr, and QJyR) highlighted this result as important. We do think that for some readers this Key Finding is an important one, and useful for the rest of the paper. We have revised the text to clarify the narrative flow and better explain why this result is the necessary motivator for our practical contributions.

---

> ### Author Response · Authors · 2025-11-19
> **Response contd. (2/2)**
>
> ### W3 On the Connection to Previous Findings:
>
> > The third key finding seems only loosely connected to the earlier results. It is unclear how the concepts of “Regime I” and “Regime II” relate to the presented equation. This section requires clearer logical linkage and explanation.
>
>
> We have rephrased the main message to ensure a more linear narrative. The connection between the three results is summarized as follows:
> 1. We show that if the dimension is high enough, CFG generates the correct target distribution.
> 2. We then show as the dimension decreases, the mean overshoot and variance shrinkage effects arise.
> 3. We finally prove these effects can be mitigated: many guidances exist that improve mean overshoot/variance shrinkage over standard CFG (of which we give a simple example, the power-law CFG).
>
> We have rephrased our third finding from “A simple nonlinear guidance mitigates these effects” to “There exist many simple guidances that effectively mitigate overshoot and variance shrinkage.” We hope this new derivation: showing CFG can generate the correct distribution -> linking it to practical limitation in stand. CFG -> providing a practical solution using nonlin. CFG, addresses the reviewer’s concern.
>
> ### W4 On Experimental Results and Resolution:
>
> > The experiments report results for resolutions of 512 and 256. According to the paper’s own claims, one might expect the method to perform even better on lower resolutions such as 64×64 or 32×32. If this is indeed the case, the authors should provide empirical evidence to support this claim.
>
> We thank the reviewer for their suggestion, which proved highly beneficial. To verify this, we performed additional experiments on CIFAR-10 (pixel space, 32x32) and ImageNet (pixel space, both 32x32 and 64x64). While the FID for high-dimensional experiments improved by roughly 8% across text-to-image and class-conditional, **the mean FID improvement for the low-dim. experiments was 15%, almost twice as large improvement as for the high-dim. datasets**. Thanks to the suggestion, we believe this further strengthened motivation for nonlinear CFG.
>
> We trained a class-cond. U-ViT-like architecture and evaluated it using standard and nonlin. CFG. We observed that the improvement in FID is indeed even stronger at these lower resolutions. The results are detailed below:
>
> ### CIFAR-10 Results
>
> | Version| Num. steps | **α** | **ω** | **FID ↓** | **Precision ↑** | **Recall ↑** |
> |--|--|--|--|--|--|--|
> | **Lin. CFG**  | 100  | 0   | 1.55 | 3.50 | 0.6569 | 0.5932 |
> ||250|0|1.40|2.77|0.6938 | 0.6115 |
> ||500| 0 | 1.35 | 2.03 | 0.6894 | 0.6262 |
> || 1000 | 0   | 1.45 | 1.80 | 0.7071 | 0.6186 |
> | **Nonlin. CFG** | 100  | 0.9 | 3.90 | 2.94 | 0.6653 | 0.6111 |
> || 250  | 0.9 | 4.35 | 2.16 | 0.6688 | 0.6298 |
> || 500  | 0.9 | 4.35 | 1.62 | 0.6688 | 0.6298 |
> || 1000 | 0.9 | 4.75 | 1.48 | 0.7228 | 0.6506 |
>
> ### IMNET-32 Results
>
> | Version       | Num. steps | **α** | **ω** | **FID ↓** | **Precision ↑** | **Recall ↑** |
> |--|--|--|--|---|--|--|
> |**Lin.CFG**|25|0|1.55|5.81|0.3842|0.4632|
> ||50|0|1.40|4.01|0.4161|0.4750|
> ||75|0|1.60|3.81|0.4120|0.4724|
> ||100|0|1.65|3.35|0.4306|0.4967|
> |**Nonlin. CFG**|25|0.9|3.40|4.97|0.4022|0.4564|
> ||50|0.9|3.35|3.90|0.4241|0.4672|
> ||75|0.9|3.40|3.33|0.4431|0.5050|
> ||100|0.9|3.85|2.88|0.4539|0.5263|
>
> ### IMNET-64 Results
>
> | Version|Num. steps|**α**|**ω**|**FID↓** | **Precision ↑** | **Recall ↑** |
> |--|--|--|--|--|--|--|
> |**Lin.CFG**|25|0|1.60|5.32|0.3996|0.4845|
> ||50|0|1.80|4.48|0.4484|0.4962|
> ||75|0|1.70|4.37|0.4576|0.5137|
> ||100|0|1.75|4.34|0.4511|0.5267|
> |**Nonlin. CFG**|25|0.9|3.80|4.76|0.4212|0.4718|
> ||50|0.9|4.00|4.02|0.4656|0.5183|
> ||75|0.9|4.20|3.75|0.4734|0.5512|
> ||100|0.9|4.15|3.71|0.4883|0.5563|
>
> Full experimental details have been added to the supplementary material (section G.5, outlined in blue) in the revised manuscript. We believe these results significantly strengthen our claims.
>
> ---
>
> We thank the reviewer again for their time and constructive criticism. We hope these clarifications have improved the presentation of our theoretical contributions, and that the additional experiments have satisfactorily addressed the reviewer's concerns.
>
> If there are any further experimental evaluations the reviewer believes would be valuable, or if any theoretical implications remain unclear, we would happily address them. If our responses have addressed the reviewer's concerns, we hope the reviewer will be willing to reconsider their score.
>
> [1] Achilli et al. Memorization and Generalization in Generative Diffusion under the Manifold Hypothesis
>
> [2] George et al. Analysis of Diffusion Models for Manifold Data
>
> [3] Chidambaram, Muthu, et al. "What does guidance do? a fine-grained analysis in a simple setting." arXiv preprint arXiv:2409.13074 (2024).
>
> [4] Wu, Yuchen, et al. "Theoretical insights for diffusion guidance: A case study for gaussian mixture models." arXiv preprint arXiv:2403.01639 (2024).

---

> ### Author Response · Authors · 2025-11-27
>
> Dear Reviewer canA,
>
> We are following up to see if you have had a chance to review our response. We specifically addressed your concerns regarding **framework extendability** and **theoretical novelty**, and provided **new experiments** demonstrating that nonlinear guidance performs even better in lower dimensions. We look forward to hearing if these changes satisfy your concerns.

---

### Author Response · Authors · 2025-12-02

We thank the reviewers for their constructive feedback. Based on their suggestions, we have revised the manuscript to:

1. **Generalize the Framework**: We clarified that our theory applies to generic mixtures of Gaussians (varying means/variances) and included a discussion on extensions to manifold data.

2. **Clarify Theoretical Contributions**: We refined the narrative to establish our main three findings:
    * CFG generates the correct target distribution in the infinite-dimensional limit.
    * Finite dimensions introduce "mean overshoot" and "variance shrinkage."
    * These artifacts are effectively mitigated by our proposed non-linear guidance.

3. **Expand Empirical Validation**: We added new experiments demonstrating:
    * Validation on general Gaussian mixtures (multiple components, different means/variancees).
    * Superior performance of non-linear guidance in low-dimensional latent-spaces and high-dimensional pixel-space settings.
    * Results using cosine-type guidances.
    * Extensive ablations on edge cases (e.g., very small $\omega$).

We appreciate the thorough discussion and thank the reviewers that reacted to our response for acknowledging that their concerns have been addressed.

---

### Meta-Review · Area_Chair_y2DQ · 2026-01-08

**Summary:**

The reviewers raised several concerns that informed the initial scores:

The theoretical analysis initially focused on a simple two-component Gaussian mixture with equal variance, raising questions about generalizability to real-world data.

The connection between the theoretical results (infinite-dimensional limit) and the proposed practical method (nonlinear guidance) was perceived as unclear or tenuous.

Reviewers requested experiments on lower-resolution datasets and edge cases to better demonstrate the method’s robustness and scope.

**Reviewer Concerns:**

Addressed concerns:

The authors clarified that their framework extends to generic Gaussian mixtures (multiple components, different means/variances) and provided proofs in the appendix. They also added a discussion on extensions to manifold data.

The rebuttal and revised manuscript better articulate the logical progression from theoretical insights (CFG correctness in high dimensions) to finite-dimensional artifacts (mean overshoot/variance shrinkage) and finally to the proposed nonlinear guidance as a mitigation strategy.

Significant new experiments were added to demonstrate improvements in lower-dimensional settings.

Outstanding concerns:

The theoretical analysis assumes perfect score estimation. While the authors acknowledge this limitation and discuss it in the conclusion, a formal analysis of robustness to imperfect estimators remains an open challenge.

**Reviewer Scores:**

Reviewer canA (initial score: 4): The reviewer’s concerns about assumptions and experimental validation were directly addressed. Given the constructive engagement and added experiments, the reviewer likely would have raised their score to 6.

Reviewer DodZ (initial score: 4): The reviewer explicitly stated they would raise their score to 6 after the authors addressed their remaining points (adding a note on Euclidean vs. cosine scaling).

Reviewer hSQr (initial score: 6): The reviewer maintained their positive score, indicating that most concerns were addressed.

Reviewer QJyR (initial score: 6): The reviewer also maintained their positive score, acknowledging that the additions strengthened the paper.

---

### Decision · Program_Chairs · 2026-01-26

Accept (Poster)